JCB Journal of Cell Biology

# Coordination of actin plus-end dynamics by IQGAP1, formin, and capping protein

Morgan L. Pimm[1], Brian K. Haarer[1], Alexander D. Nobles[1], Laura M. Haney[1], Alexandra G. Marcin[1], Marcela Alcaide Eligio[1], and Jessica L. Henty-Ridilla[1,2]

**Cell processes require precise regulation of actin polymerization that is mediated by plus-end regulatory proteins. Detailed mechanisms that explain plus-end dynamics involve regulators with opposing roles, including factors that enhance assembly, e.g., the formin mDia1, and others that stop growth (capping protein, CP). We explore IQGAP1's roles in regulating actin filament plus-ends and the consequences of perturbing its activity in cells. We confirm that IQGAP1 pauses elongation and interacts with plus ends through two residues (C756 and C781). We directly visualize the dynamic interplay between IQGAP1 and mDia1, revealing that IQGAP1 displaces the formin to influence actin assembly. Using four-color TIRF, we show that IQGAP1's displacement activity extends to formin-CP "decision complexes," promoting end-binding protein turnover at plus-ends. Loss of IQGAP1 or its plus-end activities disrupts morphology and migration, emphasizing its essential role. These results reveal a new role for IQGAP1 in promoting protein turnover on filament ends and provide new insights into how plus-end actin assembly is regulated in cells.**

## Introduction

Actin filament assembly at the leading edge is highly regulated to produce filaments of specific length and structure to power diverse cell processes. Short filaments present in lamellipodia are regulated by high-affinity interactions with plus ends (historically referred to as barbed ends) and capping protein (CP), which blocks filament polymerization (Fujiwara et al., 2014; Funk et al., 2021; Wear et al., 2003; Goode et al., 2023; Towsif and Shekhar, 2023, *Preprint*; Alimov et al., 2023). In contrast, long unbranched filaments present in filopodia or stress fibers are produced by plus end binding formin proteins, like mDia1 (Funk et al., 2021; Goode and Eck, 2007; Kovar et al., 2006; Rottner et al., 2017; Chesarone et al., 2010; Breitsprecher and Goode, 2013; Zweifel et al., 2021). Formin forms complexes with additional proteins (e.g., APC, CLIP-170, or spire) to further enhance actin assembly (Breitsprecher et al., 2012; Henty-Ridilla et al., 2016; Bradley et al., 2020; Montaville et al., 2014; Bosch et al., 2007; Wirshing et al., 2023; Ulrichs et al., 2023). Similarly, higher-order complexes can also be formed with CP (e.g., twinfilin or CARMIL) to limit filament assembly (Hakala et al., 2021; Johnston et al., 2018; Mwangangi et al., 2021; Stark et al., 2017; Wirshing et al., 2023; Ulrichs et al., 2023). CP and mDia1 also form "decision complexes" that pause filament assembly, until either protein leaves the plus end, reinitiating growth if CP dissociates first or extending the

pause in growth if formin departs first (Bombardier et al., 2015; Shekhar et al., 2015; Maufront et al., 2023). Thus, many (often seemingly opposing) plus-end regulatory proteins work together to balance actin dynamics in cells, although the specific mechanisms that detail these interactions remain unclear.

IQ-motif containing GTPase activating protein 1 (IQGAP1) is a conserved 189 kDa scaffolding protein that coordinates actin and microtubule dynamics, cell signaling pathways, and other essential cell processes (Brown and Sacks, 2006; Hedman et al., 2015; Shannon, 2012; White et al., 2012; Brandt et al., 2007; Cao et al., 2015; Thines et al., 2023). IQGAP1 influences actin filaments in two ways: filament bundling via an N-terminal calponin homology domain (CHD) and transient suppression of plus end growth via residues located in its C-terminus (744–1,657) (Hoeprich et al., 2022; Liu et al., 2016; Ren et al., 2005; Pelikan-Conchaudron et al., 2011; Bashour et al., 1997). Notably, IQGAP1 is also a ligand of formins (mDia1 and INF2) (Brandt et al., 2007; Bartolini et al., 2016; Chen et al., 2020). Whether IQGAP1–formin activities influence the assembly of individual actin filaments or multi-component plus-end regulatory systems like the formin-CP "decision complex" is not known.

Here, we identify residues in IQGAP1 that mediate interactions with actin filament plus ends. We use four-color TIRF

---

[1]Department of Biochemistry and Molecular Biology, SUNY Upstate Medical University, Syracuse, NY, USA;   [2]Department of Neuroscience and Physiology, SUNY Upstate Medical University, Syracuse, NY, USA.

Correspondence to Jessica L. Henty-Ridilla: ridillaj@upstate.edu.



microscopy monitoring each molecular player to show that IQGAP1 is not a transient capping protein but rather an end-protein displacement factor that removes the formin mDia1, CP, or stalled decision complexes from plus-ends. The loss of these activities perturbs cell shape, cytoskeletal arrays, and migration. Thus, IQGAP1 promotes a more frequent exchange of proteins present on plus ends to regulate filament assembly.

## Results

### IQGAP1 bundles and temporarily pauses actin filament elongation at the plus end

To explore the effects of IQGAP1 on actin filament assembly, we purified the 189-kDa full-length protein (FL-IQGAP1; Fig. 1 A) via 6×His affinity and gel filtration (Fig. 1 B). We directly assessed its effects on actin filament assembly using time-lapse total internal reflection fluorescence (TIRF) microscopy assays over a range of IQGAP1 concentrations (Fig. 1 C and Video 1). Reactions containing 1 μM actin polymerized as expected with filaments encompassing the field of view (FOV) within 600 s (Fig. 1 C). However, actin filaments in reactions containing any nanomolar concentration of IQGAP1 were noticeably sparse, and reactions contained several thick filament bundles (Fig. 1 C). Fewer actin filaments in IQGAP1-containing TIRF reactions may arise from several different scenarios including a reduction in the number of filaments being nucleated, changes to the filament elongation rate, filament capping events, or the coalescence of filaments into bundles. To distinguish between these mechanisms, we examined individual actin filaments present in FOVs more closely (Fig. 1, D–G). We first counted the number of filaments present in TIRF FOVs 200 s after polymerization was initiated in the absence (i.e., control: actin alone) or the presence of different concentrations of IQGAP1 (Fig. 1 F). The mean number of filaments varied between 35.3 (125 nM IQGAP1) and 60.7 (control). However, we did not observe a statistically significant change from the actin-alone control for any concentration of IQGAP1 tested (Fig. 1 F; P = 0.7926). Next, we measured the length (μm) of individual actin filaments over time to calculate the mean elongation rate of actin filaments present in TIRF reactions performed over a range of IQGAP1 concentrations. In contrast to the nucleation parameter, all reactions containing IQGAP1 significantly slowed the mean rate of actin filament elongation, from 10.2 ± 0.2 (SE) subunits s$^{-1}$ μM$^{-1}$ to 6.9 ± 0.3 (SE) subunits s$^{-1}$ μM$^{-1}$ (Fig. 1 G; P < 0.0001). Reduced mean rates of elongation could arise from processively slowed filament assembly, abrupt capping events that block filament growth, or other mechanisms that may transiently pause filament assembly. We specifically hypothesized that the reduction in rates was caused by a previously identified plus-end capping activity (Hoeprich et al., 2022; Pelikan-Conchaudron et al., 2011).

To distinguish between these mechanisms, we examined the elongation rate data more closely (Fig. 1 G). Using montages of individual filaments (Fig. 1 D), kymographs (Fig. 1 E), and length over time plots (Fig. 1 H), we noticed filaments polymerized in the presence of IQGAP1 often displayed distinct pauses to their elongation rate (Fig. 1, D, E, and H; and Video 2). To quantify these effects further, we measured the frequency and duration

of pauses to actin assembly. Indeed, actin filaments from reactions containing IQGAP1 suffered pauses to elongation (sometimes multiple; 438 pauses were recorded from 375 total actin filaments). On average, the duration of IQGAP1-mediated pauses was 20.6 ± 1.9 (SE) s, regardless of concentration (Fig. 1 I). Thus, IQGAP1 performs two actin-related activities: (1) it bundles actin filaments (Fig. 1 C; (Hoeprich et al., 2022; Bashour et al., 1997; Mateer et al., 2002; Samson et al., 2017; Fukata et al., 1997), and (2) it reduces overall actin filament assembly by transiently pausing elongation at the filament plus-end (Fig. 1, D–I).

### Two cysteine residues are essential for IQGAP1's plus-end functions

Previous studies attribute IQGAP1's actin filament side-binding and bundling activities to the calponin homology domain (CHD) located in the first 160 residues (Fig. 2 A) (Hoeprich et al., 2022; Fukata et al., 1997; Ho et al., 1999). The residues associated with IQGAP1's "transient capping" or plus-end pausing activity are less specific and thought to be located in the C-terminal half of the protein (residues 745–1,502) and likely require protein homodimerization to function (Hoeprich et al., 2022; Pelikan-Conchaudron et al., 2011). To further deduce the residues required for interacting with actin filament plus-ends and ultimately the residues that pause filament growth, we performed an extensive truncation analysis of IQGAP1, purifying 16 versions of the protein to assess the role of its known features and compare it with previous studies (Fig. 2 A; and Fig. S1, A and B). To assess the capacity for pausing actin filament elongation, we performed TIRF microscopy with polymerizing actin filaments and 75 nM of each protein (Fig. 2, A and B; Fig. S1 C, and Video 3). This concentration was sufficient (k$_D$ = 25–35 nM [Hoeprich et al., 2022; Pelikan-Conchaudron et al., 2011]) for identifying IQGAP1-mediated pauses to actin filament elongation in TIRF assays, in kymographs made from individual filaments (Fig. 2, B and C; and Fig. S1 C), and for calculating mean filament elongation rates comparing various conditions (Fig. 1 G, Fig. 2 D, and Video 3). N-terminal fragments of IQGAP1 lacking IQ motifs, GRD, and LBR regions (i.e., 1–159, 1–216, and 1–744) were unable to bundle or pause actin filament elongation (Fig. 2 D and Video 3; P ≥ 0.1818, compared with actin alone control), consistent with previous studies (Hoeprich et al., 2022; Pelikan-Conchaudron et al., 2011). In contrast, actin filaments present in reactions containing any modified IQGAP1 protein with an intact dimerization domain and IQ motif–containing region exhibited enough pauses to filament elongation to significantly reduce the average elongation rate compared with controls lacking IQGAP1 (P < 0.0001; Fig. 2 D and Video 3). These IQGAP1 proteins did not bundle actin filaments and presumably do not bind filament sides as they each lack the required CHD domain. This analysis effectively narrowed plus-end pausing activity to 280 residues (amino acids 745–1,024) that contain the four IQ motifs and dimerization region.

While the goal of our truncation analysis was to identify the residues responsible for plus-end activities, we were concerned that the CHD domain may confound our analyses by providing an abundant source of IQGAP1 binding sites at filament sides.

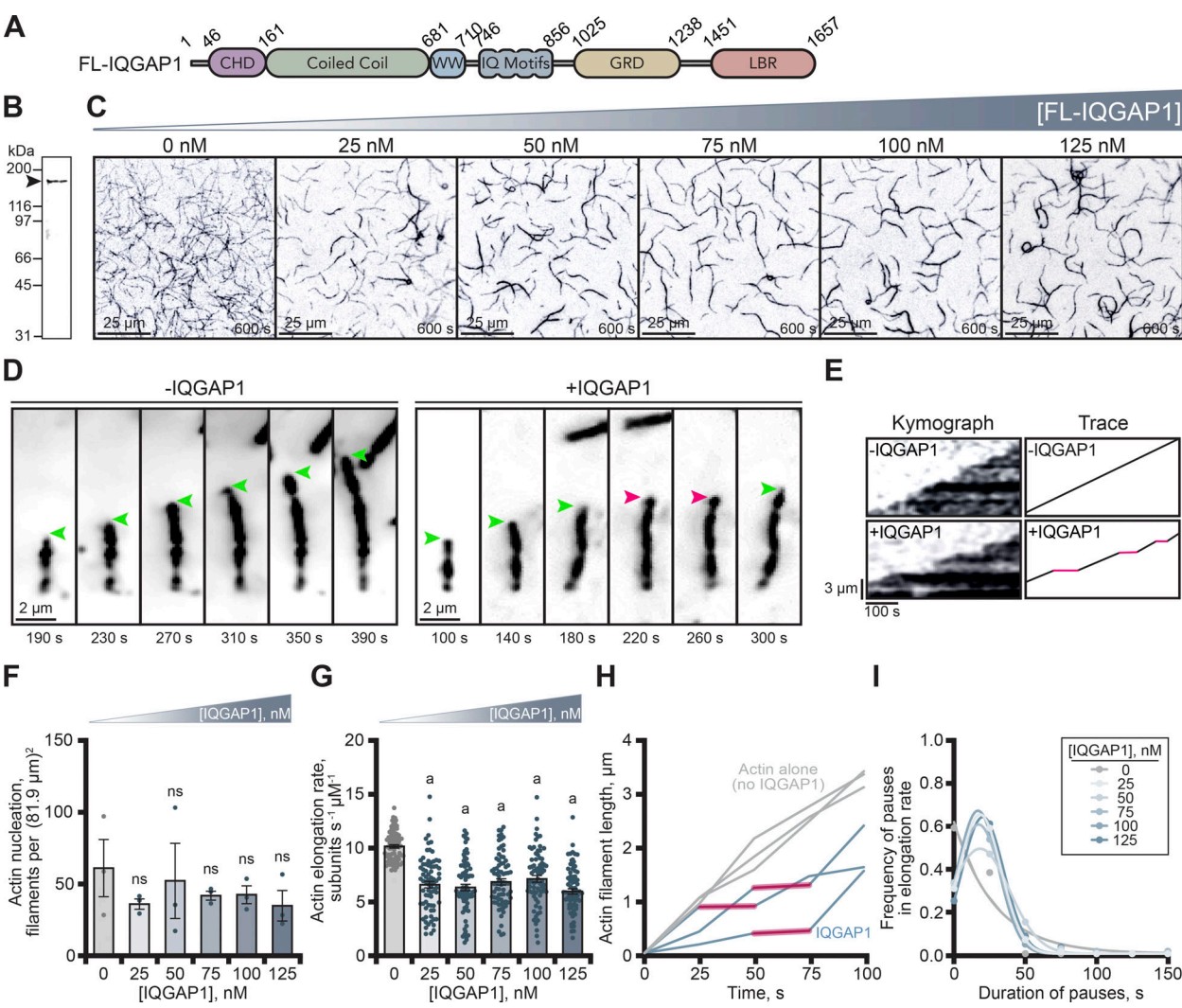

Figure 1. **IQGAP1 reduces the mean actin filament elongation rate in vitro. (A)** Schematic of IQGAP1 domains. Abbreviations: CHD, calponin homology domain; WW, WW domain; GRD, GAP-related domain; LBR, ligand binding region. **(B)** SDS-PAGE gel of purified IQGAP1. **(C)** Images from TIRF assays containing 1 µM actin monomers (20% Oregon Green [OG]-label) and noted concentrations of IQGAP1. Scale, 25 µm. **(D)** Image montages displaying the polymerization of single actin filaments in the absence or presence of 75 nM IQGAP1. Arrows mark actively growing ends (green) or IQGAP1-mediated pauses in actin filament elongation (pink). Scale, 2 µm. **(E)** Kymographs and traces of elongating actin filaments in the absence or presence of 75 nM IQGAP1. Red lines indicate pauses in elongation. Scale: length, 3 µm; time, 100 s. **(F)** Mean actin filament nucleation at noted concentrations of IQGAP1. Dots represent filament counts 200 s after initiation of reactions in C from $n = 3$ fields of view. **(G)** Mean actin filament elongation rates from TIRF reactions in C. Dots represent elongation rates of individual actin filaments ($n = 75$ filaments per condition; pooled from 3 independent trials). Error bars in F and G, SE. Statistics, ANOVA: (a) $P \leq 0.05$ compared with control (0 nM IQGAP1); ns, $P \geq 0.05$ control. **(H)** Representative actin filament length-over-time plots indicate that filaments polymerized in the presence of 75 nM IQGAP1 (teal) have pauses in filament elongation (red shading), whereas filaments polymerized without IQGAP1 (gray) do not. **(I)** Frequency distribution plots indicate the average duration (20.6 s) of IQGAP1-mediated pauses in actin filament elongation. Pauses were calculated from elongation rates measured in G ($n = 31–70$ pauses [331 total] measured from $n = 75$ filaments per condition). The $R^2$ values for Gaussians ranged between 0.99 and 1.00, whereas the $R^2$ values for non-pausing conditions fit using a non-linear fit ranged between 0.94 and 0.99. Source data are available for this figure: SourceData F1.

Thus, to separate IQGAP1's bundling and plus-end pausing activities, we generated IQGAP1(160-end), which lacks the CHD but contains the dimerization region, to test whether IQGAP1's plus-end pauses were further enhanced, extended, or otherwise different from the full-length protein (Fig. 2, A–D, Fig. S1, A–E, and Video 3). Unsurprisingly, actin filaments from reactions containing the 160-end protein appeared less bundled than reactions containing full-length IQGAP1 (Fig. 2 B and Video 3). Filaments appeared shorter in these reactions and elongated at 6.96 ± 0.23 subunits s⁻¹ µM⁻¹, i.e., significantly slower than

control reactions lacking IQGAP1 (P = 0.0091; Fig. 2 D), but significantly faster than reactions with the full-length protein (P < 0.0001). Additional analysis of actin filament elongation rates and pause durations revealed that 160-end does pause actin filament elongation (Fig. 2, A–C and Fig. S1, D–F); however, these pauses are significantly shorter than reactions containing full-length IQGAP1 (P = 0.0205), lasting an average of 9.3 s ± 2.4 (SE) (Fig. 2 C; and Fig. S1, D and E). These results demonstrate that IQGAP1(160-end) can pause filament growth and reduce filament bundling. Unfortunately, this observation did not aid in

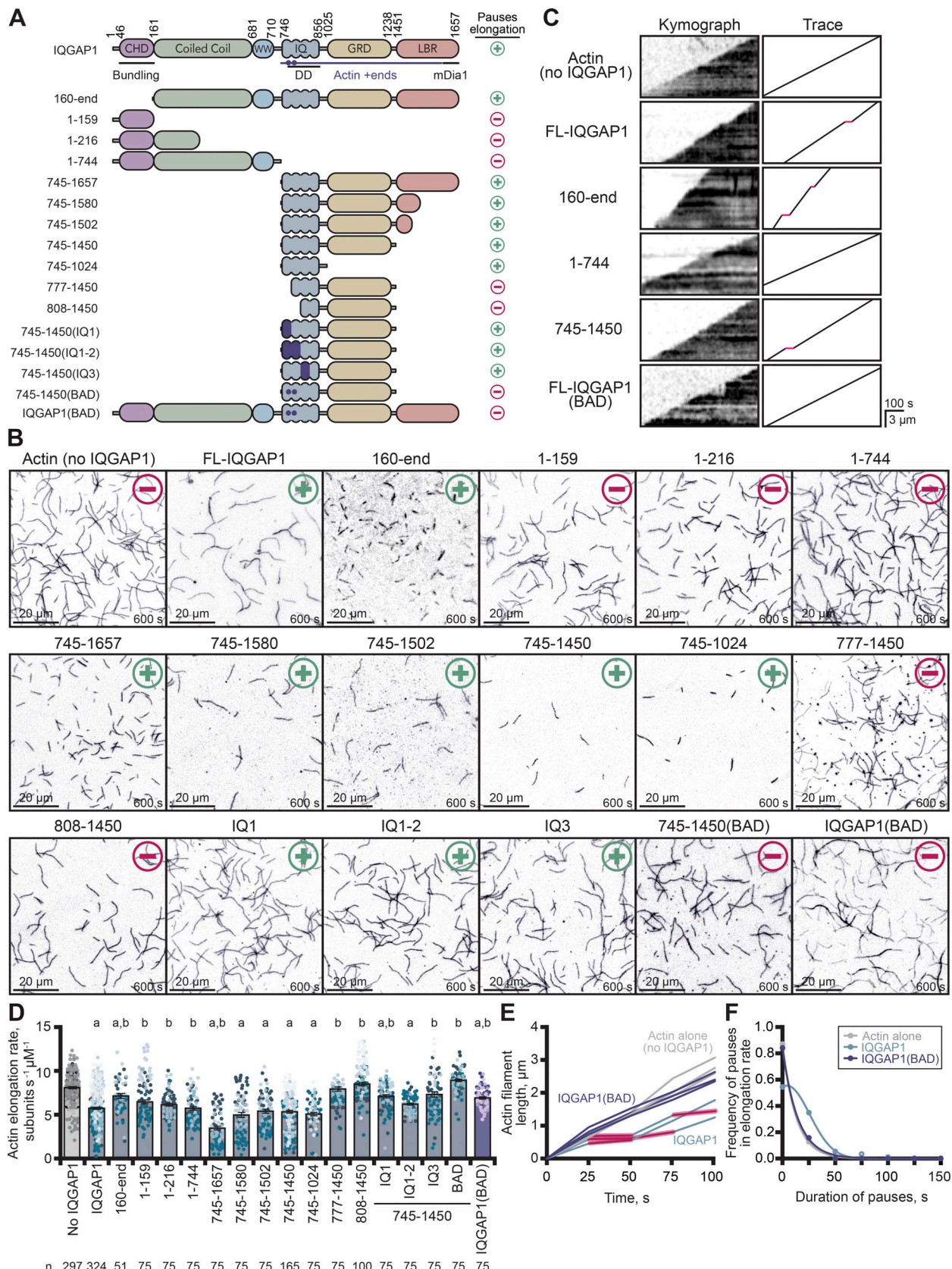

Figure 2. **IQGAP1(BAD) does not pause elongation or transiently cap actin filaments. (A)** IQGAP1 constructs that pause (+) or fail to pause (−) actin filament elongation. DD, dimerization domain. Purple shading, location of disrupting mutations in IQ motifs. Purple dots, two residues necessary for plus-end activities. **(B)** Representative images of actin filaments from TIRF reactions containing 1 µM actin monomers (20% OG-label) and 75 nM of each indicated

IQGAP1 protein. BAD, Barbed-end Association Deficient. Scale, 20 μm. **(C)** Kymographs and traces of elongating actin filaments in the presence of key mutants from B. Red lines in traces indicate pauses to filament elongation. Scale: length, 3 μm; time, 100 s. **(D)** Mean actin filament elongation rates from TIRF reactions in B. Dots represent rates of individual actin filaments ($n$ = 51–324 filaments per condition with exact values noted per condition). Dot shading indicates experimental replicates ($n \geq 3$ per condition). Error bars, SE. Statistics, ANOVA: (a) $P \leq 0.05$ compared with control (0 nM IQGAP1); (b) $P \leq 0.05$ compared with actin and 75 nM IQGAP1. **(E)** Filament length-over-time plots and **(F)** Frequency distribution plots displaying the duration of IQGAP1-mediated pauses in filament elongation ($n$ = 159/324 pauses/filaments for IQGAP1; $n$ = 12/75 pauses/filaments for IQGAP1(BAD) in B).

extending the length or detectability of IQGAP1 pauses but is consistent with the notion that the high-affinity CHD-side binding interactions contribute to slow off-rate of IQGAP1 from filament sides ($K_{off}$ = 0.0010 s$^{-1}$ [Hoeprich et al., 2022]).

We continued to narrow our focus on residues 745–1,024, the minimal region necessary for filament pausing, comprised of the four IQ motifs and IQGAP1 dimerization region. However, this protein was prone to degradation and did not bind the 6×His affinity column (Fig. S1 B). Therefore, we used the stable and highly pure IQGAP1(745–1,450) (Fig. S1 B) and site-directed mutagenesis to dissect the contribution of each IQ motif and the only two cysteine residues (i.e., C756 and C781) present in this region. Surprisingly, actin filaments present in TIRF reactions containing purified proteins with disrupting mutations in IQ-motif 1, IQ-motif 1 and 2, or IQ-motif 3 behaved similarly to the full-length protein (Fig. 2, A, B, and D; and Fig. S1). Each reduced the mean elongation rate of filaments significantly compared with filaments in reactions lacking IQGAP1 ($P \leq 0.0022$). Notably, each IQ-motif mutant contained C756 and C781 and still displayed plus-end activities (Fig. 2, B and C; Fig. S1, C–F, and Video 3). Both cysteine residues lie in the calmodulin-binding regions and are directly adjacent to residues involved in salt-bridge formation (Zhang et al., 2019). Thus, we substituted these residues for alanine in IQGAP1(745–1,450), purified the protein, and tested its actin filament pausing activity in TIRF assays (Fig. 2 and Fig. S1). Actin filaments polymerized in the presence of IQGAP1 containing the two alanine substitutions elongated consistently, without pauses, at a rate of 8.9 ± 0.1 (SE). This was not significantly different from the rate of actin alone of 8.1 ± 0.1 (SE) ($P$ = 0.0892) (Fig. 2 D), but significantly faster than reactions that contained the full-length IQGAP1 ($P$ < 0.0001). To further confirm that the mutation of these residues resulted in a barbed-end association deficient (BAD) IQGAP1, we substituted C756 and C781 for alanine in the sequence of full-length IQGAP1. We purified the protein (Fig. S1, A and B) and monitored its effect on actin filament assembly (Fig. 2 B). Indeed, actin filaments polymerized in the presence of full-length IQGAP1(BAD) elongated consistently at a mean rate of 6.99 ± 0.13 (SE) compared with 5.79 ± 0.12 (SE) for the unmutated protein (Fig. 2 D). This rate was significantly faster than reactions containing the unaltered IQGAP1 protein ($P$ = 0.0002). Further analysis of kymographs (Fig. 2 C and Fig. S1 C), representative filament length-over-time traces (Fig. 2 E), and the frequency distribution of pause durations (Fig. 2 F and Fig. S1, D–F) confirmed that filaments in these reactions display uninterrupted growth, while also retaining the ability to bundle actin filaments (Fig. 2 B). Henceforth, we refer to IQGAP1 harboring the cysteine mutations as IQGAP1(BAD) proteins.

## IQGAP1(BAD) is a dimer that does not pause filament elongation or localize to plus ends

IQGAP1(BAD) appeared to lack plus-end pausing activity (Fig. 2 and Fig. S1). However, single-wavelength TIRF microscopy assays where only actin assembly is monitored do not directly rule out competing interpretations, including failed dimerization. Thus, to further explore these ideas, we generated, purified, and fluorescently labeled several SNAP-tagged IQGAP1 proteins, including FL-IQGAP1, FL-IQGAP1(BAD), and the CHD-absent IQGAP1(160-end) (Fig. S2, A and B). We first tested the activity of the two full-length SNAP-tagged proteins head-to-head with the untagged versions in pyrene fluorescence assays containing preformed actin filament seeds (Fig. 3 A). Both SNAP-IQGAP1 and the untagged version blocked some end-based elongation, albeit to a much lesser extent than the hallmark capping factor, heterodimeric CP (Wear et al., 2003) (Fig. 3 A). In contrast, IQGAP1(BAD) (SNAP-tagged or tag-free) did not block end-based elongation, with bulk assembly reaching similar levels as reactions lacking IQGAP1 (Fig. 3 A). This demonstrates that the activities of SNAP-IQGAP1 and SNAP-IQGAP1(BAD) are comparable with the untagged proteins and further confirms IQGAP1's end-based pausing activity via a complementary approach to TIRF microscopy assays (Figs. 1 and 2; and Fig. S1). Previous determinations by analytical ultracentrifugation and step-photobleaching suggest that IQGAP1 exists as a dimer (Fukata et al., 1997; Hoeprich et al., 2022). Similarly, we used step-photobleaching to determine the oligomeric state of each SNAP-tagged protein and to assess if mutations present in IQGAP1(BAD) negatively impact its oligomeric state (Fig. 3, B–D). The distribution of observed step-photobleaching events for molecules of 488-SNAP-IQGAP1 were mostly two steps (Fig. 3, B and C) and were most consistent with the mathematical prediction for it to exist as a dimer. 488-SNAP-IQGAP1(BAD) had a similar percent label, most observations bleached in two steps, and overall observations were consistent with the prediction it was also a dimer. Therefore, differences in plus-end activities were likely not due to changes in the protein's oligomeric state (Fig. 3, B–D). Most molecules of 488-SNAP-IQGAP1(160-end) bleached in one step or two steps (Fig. 3, B–D). Unfortunately, we are unable to conclusively determine the oligomeric state of 488-SNAP-IQGAP1(160-end) due to its low labeling efficiency.

We utilized two-color TIRF microscopy assays to see if we could visualize labeled IQGAP1 on the ends or sides of actin filaments. We hypothesized that 488-SNAP-IQGAP1 (488-IQGAP1) would be present on filament sides and plus-ends, and that plus-end association might coincide with pauses in filament elongation. Indeed, 488-IQGAP1 was present on filament ends in two-color TIRF reactions (Fig. 3 E), and pauses in filament

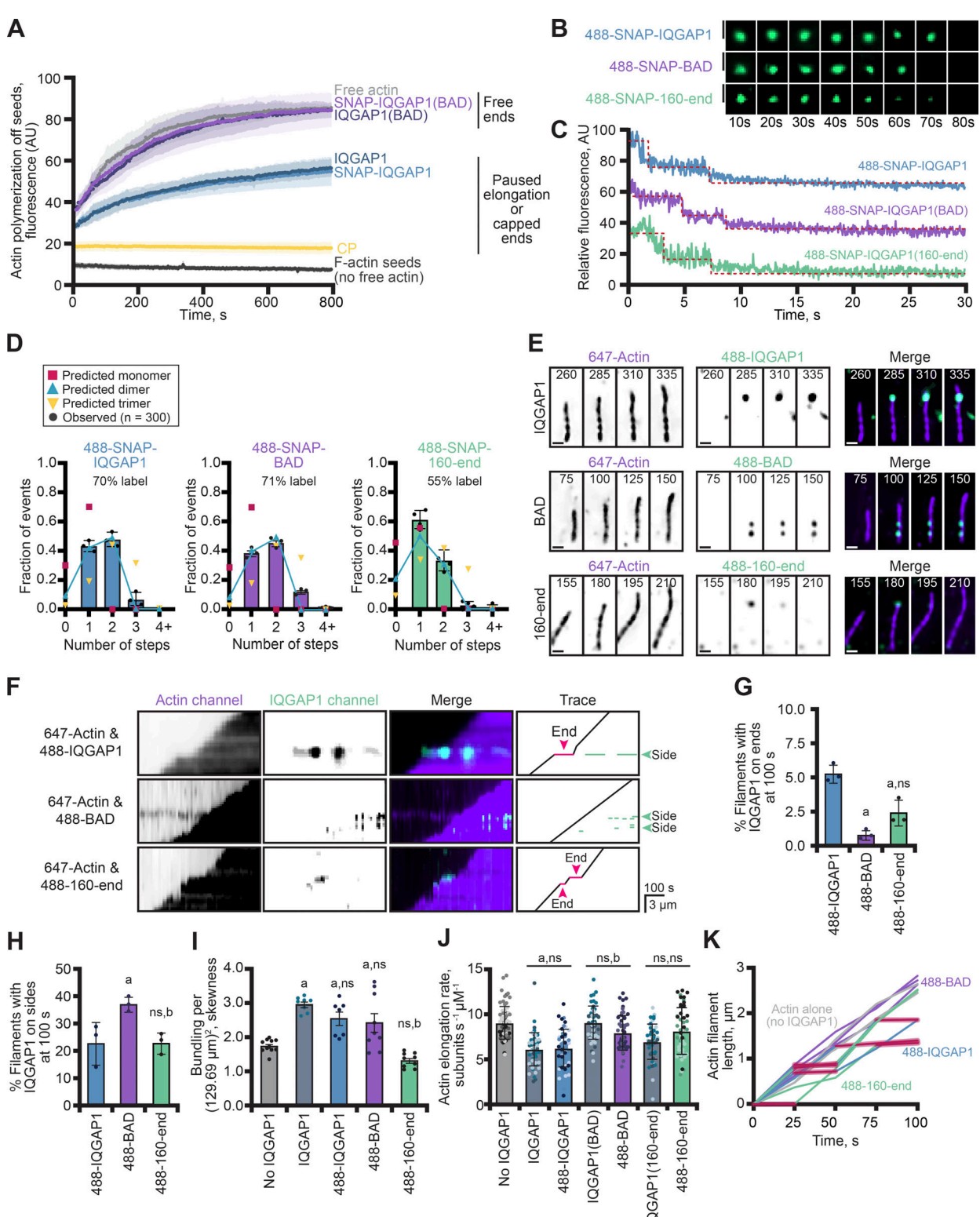

**Figure 3. IQGAP1(BAD) is a dimer that bundles actin but does not influence the rate of filament elongation. (A)** IQGAP1(BAD) does not interfere with plus-end elongation in seeded actin assembly assays. Assays contain pre-polymerized (unlabeled) actin seeds, 0.5 µM actin monomers (5% pyrene-labeled), and 75 nM of each indicated IQGAP1 protein. Reactions with 10 nM Capping Protein (CP) or unlabeled seeds (alone) in the absence of IQGAP1 were used as polymerization negative controls. Values were averaged from n = 3 ± SD (shaded). **(B)** Single molecules of 488-labeled SNAP-IQGAP1, SNAP-IQGAP1(BAD), and SNAP-IQGAP1(160-end) subjected to step-photobleaching analysis. **(C)** Fluorescence intensity profiles of step photobleaching events for 1 nM SNAP-IQGAP1 proteins from reactions as in B. Red lines emphasize individual photobleaching steps. Scale, 1 µm. **(D)** Predictions and analysis of the oligomeric state of 488-SNAP-IQGAP1 proteins from photobleaching reactions in B and C (n = 300 molecules per protein, pooled from three replicates). **(E)** Representative two-channel montages depicting polymerization of single actin filaments (10% Alexa 647-label) with 75 nM of each 488-SNAP-labeled IQGAP1 over 75 s. Scale, 1 µm.

**(F)** Kymographs and traces of elongating actin filaments as in E. Red lines and arrows indicate IQGAP1-induced pauses to filament elongation. Green lines and arrows denote IQGAP1-actin filament side-binding interactions. Scale: length, 3 μm; time, 100 s. **(G and H)** Percentage of actin filaments with 488-SNAP-IQGAP1 molecules on plus ends (G) or sides at 100 s in a single field of view (H) from reactions in E ($n$ = 3 FOVs, dots). All error bars, SE unless otherwise noted. Statistics in G and H, ANOVA: (a) $P ≤ 0.05$ compared with 488-SNAP-IQGAP1; (b) $P ≤ 0.05$ compared with 488-SNAP-IQGAP1(BAD); ns, $P ≥ 0.05$ compared with 488-SNAP-IQGAP1. **(I)** Actin filament bundling (skewness) quantified at 90 s from TIRF reactions in E ($n$ = 6–10 FOVs, dots). **(J)** Mean actin filament elongation rates from reactions as in E. Dots represent rates of individual actin filaments ($n$ = 17 filaments from each of 3 independent replicates [different shades] per condition, $n$ = 51 filaments measured in total per condition). Error bars, SD. Statistics in I and J, ANOVA: (a) $P ≤ 0.05$ compared with actin alone (no IQGAP1); (b) $P ≤ 0.05$ compared with untagged IQGAP1; ns, $P ≥ 0.05$. In all instances, the SNAP-tagged protein was not significantly different compared with the untagged protein. **(K)** Representative actin filament length-over-time plots for 75 nM 488-SNAP-IQGAP1 proteins. Both SNAP-IQGAP1 (blue) and SNAP-IQGAP1(160-end) (green) have noticeable pauses in filament elongation (red shading), whereas filaments polymerized with SNAP-IQGAP1(BAD) (purple) do not.

elongation could be seen in representative kymographs of filaments (Fig. 3 F). Sometimes these pauses ended with molecules of IQGAP1 dissociating from the end (Fig. 3 F), while at other times the molecules may have been repositioned from plus-ends to filament sides (Fig. 3, E and F; and Video 4). At 100 s, 5.3% ± 0.4 (SE) of all filaments present in TIRF fields of view (FOV) had 488-IQGAP1 on plus-ends and 22.8% ± 4.6 (SE) of filaments had molecules on filament sides (Fig. 3, G and H). Not surprisingly, more bundled actin filaments were present in reactions that contained 488-IQGAP1 compared with actin alone controls ($P < 0.0001$), and the extent of bundling in these reactions was not significantly different than reactions performed with the untagged protein ($P = 0.4135$) (Fig. 3 I). As a final measure of quality control between untagged- and 488-IQGAP1, we measured the elongation rate of actin filaments present in two-color TIRF microscopy assays (Fig. 3, E, J, and K). Unsurprisingly, the presence of 75 nM IQGAP1 significantly slowed the mean elongation rate of polymerizing actin filaments in this experiment from 9.04 ± 0.22 (SE) to 6.08 ± 0.26 (SE) subunits s$^{-1}$ μM$^{-1}$ ($P < 0.0001$) (Fig. 3, J and K). The same concentration of 488-IQGAP1 behaved in a manner not significantly different from the untagged version ($P > 0.9999$) and significantly slowed the mean rate of actin filament elongation to 6.21 ± 0.29 subunits s$^{-1}$ μM$^{-1}$ (SE) ($P < 0.0001$) (Fig. 3, J and K). These experiments demonstrate that 488-IQGAP1 behaves identically to the untagged protein in several actin assembly assays.

Single-color TIRF assays suggest that IQGAP1(BAD) may not bind or perform plus-end activities but its side-binding interactions may remain intact. Conversely, IQGAP1(160-end) was not able to bind filament sides, and by freeing up potential binding sites, it may interact more robustly at plus-ends than full-length IQGAP1. With SNAP-labeled versions of these proteins in hand, we next assessed the localization and functionality of 488-SNAP-IQGAP1(BAD) (488-BAD) and 488-SNAP-IQGAP1(160-end) (488-160-end) in two-color TIRF microscopy assays (Fig. 3, E and F; and Video 4). As expected, 488-BAD does not localize to filament plus ends as well as 488-IQGAP1 (0.8% ± 0.2; $P = 0.0055$; Fig. 3, E–G) but does robustly bind to filament sides, labeling 37.3% of all filaments observed at 100 s, which was significantly more than 488-IQGAP1 ($P = 0.0368$; Fig. 3 H). The presence of 488-IQGAP1(BAD) on filament sides significantly promoted actin filament bundling compared with controls lacking IQGAP1 ($P = 0.0059$), though bundling levels were not significantly elevated comparing FOVs generated with the untagged IQGAP1 and IQGAP1(BAD) proteins ($P = 0.1309$) (Fig. 3

I). As expected, 488-IQGAP1(BAD) did not pause mean actin filament elongation significantly different from the untagged version ($P = 0.0759$) or controls lacking the protein ($P = 0.0538$) (Fig. 3, J and K), and these values were significantly faster than the unmutated 488-SNAP-IQGAP1 protein ($P = 0.0005$). We also tested 488-160-end, which did not fully behave as expected. It localized to the plus end (Fig. 3, E–G), although significantly less than the full-length protein ($P = 0.0055$). Despite lacking the CHD domain, single-molecules of 488-160-end were present on the sides of 22% of filaments present in FOVs (Fig. 3 H), although there was no significant amount of bundling measured by the skewness parameter as compared with actin-alone controls ($P = 0.2557$) (Fig. 3 I). Finally, the mean rate of actin filament elongation for 488-160-end was not significantly different than actin alone control ($P = 0.1886$), despite some observations of filament pausing events (Fig. 3 K) and previous observations of significantly elevated mean elongation rates compared with the full-length protein ($P < 0.0001$; Fig. 2 D). We did not use 488-160-end in additional assays due to this weaker pausing activity. In sum, these experiments demonstrate that 488-BAD behaves like the untagged protein, does not localize to or pause plus-ends, and still retains side-binding and filament bundling activities.

**IQGAP1 can displace mDia1 from actin filament plus ends**

Purified IQGAP1 directly activates the formin mDia1 by binding to its Diaphanous Inhibitory Domain (DID) to relieve auto-inhibition (Fig. 4 A [Brandt et al., 2007; Boyer et al., 2011; Bartolini et al., 2016; Chen et al., 2020; Wallar et al., 2006]). Compelling biochemical evidence detailing the contribution of either IQGAP1 or mDia1 to actin assembly suggests that these proteins may function as agonists of each other. IQGAP1 slows the mean rate of filament elongation (Figs. 1, 2, and 3 [Hoeprich et al., 2022; Pelikan-Conchaudron et al., 2011]), whereas mDia1, in the presence of profilin-1 (PFN1), drastically accelerates filament nucleation and elongation (Kovar et al., 2006; Courtemanche, 2018; Breitsprecher and Goode, 2013; Zweifel and Courtemanche, 2020). These observations motivated us to explore whether IQGAP1 and mDia1 could bind the same filament plus end and how they might synergize to mediate actin assembly. Would a plus-end associate complex of mDia1 and IQGAP1 lead to fast filament assembly, pause filament growth, or something unexpected and emergent?

Many studies assessing formin-based actin assembly use a constitutively active version (i.e., mDia1(FH1-C)), which lacks the IQGAP1-binding site (Fig. 4 A) (Brandt et al., 2007; Boyer et al., 2011; Bartolini et al., 2016; Chen et al., 2020). We also

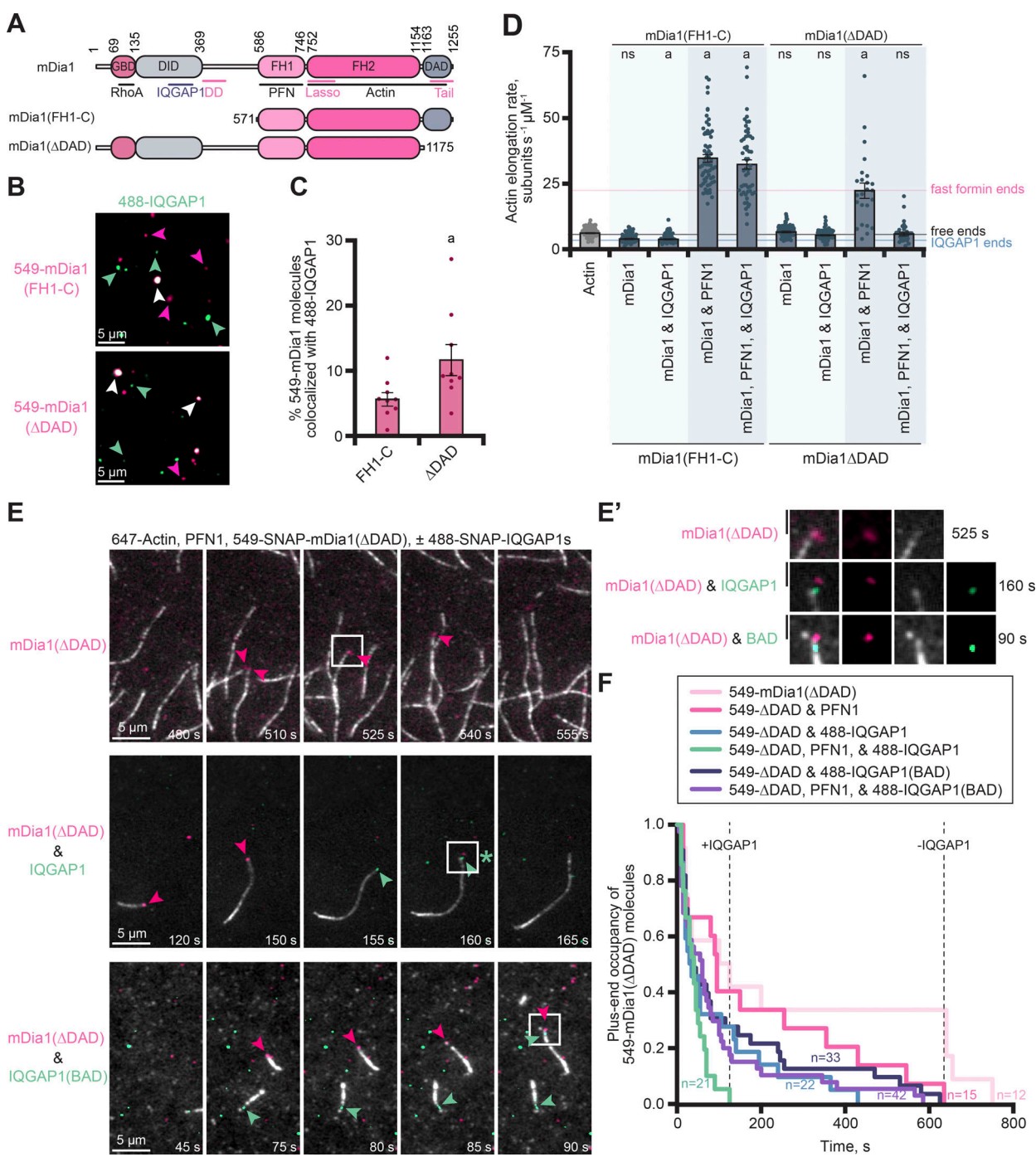

Figure 4. **IQGAP1 and mDia1 can co-occupy plus ends and binding reduces formin-based elongation by displacing mDia1. (A)** Schematic of formin (mDia1) constructs that bind (ΔDAD) or do not bind (FH1-C) to IQGAP1. Abbreviations: GBD, GTPase-binding domain; DID, Diaphanous inhibitory domain; FH1, formin homology 1 domain; FH2, formin homology 2 domain; DAD, Diaphanous autoregulatory domain. DD, dimerization domain. **(B)** Images from single-molecule TIRF of 1 nM 549-mDia1 constructs with 1 nM 488-IQGAP1. Arrows highlight examples of individual molecules of SNAP-IQGAP1 (green) or SNAP-mDia1 (pink) and colocalization of both proteins (white). Scale, 5 μm. **(C)** Quantification of colocalized mDia1-IQGAP1 molecules from reactions in B. Error bars, SE. Dots are percentages calculated for individual FOVs (n = 9 FOVs total, pooled from three replicates). Statistics, Student's *t* test (two-tailed): (a) P ≤ 0.05 compared with mDia1(FH1-C). **(D)** Mean actin filament elongation rates from TIRF reactions containing 1 μM actin (10% Alexa 488 label), 5 μM profilin-1 (PFN1), 10 nM mDia1(FH1-C or ΔDAD), and 75 nM IQGAP1, as indicated. Reactions performed at least three times for each condition. Measurements from n = 24–105 filaments (dots). Horizontal lines indicate the mean elongation rate of filaments elongating alone (free ends; black), the mean filament elongation rate in the presence of IQGAP1 (blue) or the mean elongation rates stimulated by formin (e.g., ΔDAD growth with PFN1; pink). Error bars, SE. Statistics, ANOVA: (a) P ≤ 0.05 compared with control (actin alone); ns, P ≥ 0.05 compared with (actin alone). **(E)** Time-lapse montages from three-color TIRF reactions containing 1 μM actin (10% Alexa 647 label; gray) polymerizing in the presence of 5 μM PFN1, 1 nM 549-SNAP-mDia1(ΔDAD) (pink), and 1 nM 488-SNAP-IQGAP1 or 488-SNAP-IQGAP1(BAD) (green). Scale, 5 μm. Arrows indicate mDia1 (pink) and IQGAP1 (green) localization. **(E')** Insets (boxes) from E show a zoomed in view of single molecules on or near filament ends. The asterisk in E marks an instance of likely IQGAP1-mediated formin displacement from the plus end. Scale, 1.5 μm.

**(F)** Survival plots of the plus-end occupancy of 549-SNAP-mDia1(ΔDAD) molecules in the presence and absence of 488-labeled IQGAP1 proteins from reactions as in E. SNAP-mDia1 persists on ends for a longer time in reactions without IQGAP1 than in reactions with IQGAP1, as marked by dotted lines (n = 12–42 molecules per condition, as stated, from n = 3 independent reactions).

generated mDia1(ΔDAD) which retains the IQGAP1 binding site and could be constitutively active for actin filament assembly because it lacks the diaphanous autoinhibitory domain (DAD) (Fig. 4 A and Fig. S3 A). We purified and directly compared the actin assembly capacity of untagged and 549-SNAP-tagged versions of these formins in bulk pyrene fluorescence and TIRF microscopy assays. Regardless of construct or tag, each formin promoted actin filament assembly to similar levels, which could be further stimulated in the presence of PFN1 (Fig. S3, B and C). These observations were further confirmed by measuring the mean elongation rate of actin filaments present in single-color TIRF microscopy assays (Fig. S3 D). As expected, formin-based filament elongation was approximately fivefold faster in the presence of PFN1, from 7.6 ± 0.4 (SE) to 52.4 ± 1.5 (SE) subunits s$^{-1}$ μM$^{-1}$ for mDia1(FH1-C) (P < 0.0001), and from 9.4 ± 0.3 (SE) to 58.3 ± 1.9 (SE) subunits s$^{-1}$ μM$^{-1}$ for mDia1(ΔDAD) (P < 0.0001) (Fig. S3 D). With functional tagged and untagged formins in hand, we used two-color TIRF microscopy to assess the binding capacity of each formin for 488-IQGAP1 (Fig. 4, B and C) or 488-IQGAP1(BAD) (Fig. S3, E and F). As expected, both 488-IQGAP1 proteins show significantly greater association with 549-mDia1(ΔDAD), which contains the IQGAP1 binding site, than with molecules of 549-mDia1(FH1-C) (P = 0.0324 and P = 0.0109 for 488-IQGAP1 or 488-IQGAP1(BAD), respectively).

We next performed TIRF microscopy assays to evaluate whether the activity of IQGAP1 or mDia1 prevailed at actin filament plus ends (Fig. 4 D). Each untagged formin construct performed as expected, significantly accelerating mean actin filament elongation from 6.5 ± 0.2 (SE) subunits s$^{-1}$ μM$^{-1}$ to 35.1 ± 1.4 (SE) subunits s$^{-1}$ μM$^{-1}$ for mDia1(FH1-C) (P < 0.0001) or 22.7 ± 2.9 (SE) subunits s$^{-1}$ μM$^{-1}$ for mDia1(ΔDAD) (P < 0.0001) (Fig. 4 D). The addition of IQGAP1 to reactions containing mDia1(FH1-C) and PFN1 did not significantly change the rate of actin assembly. However, the mean elongation rate of actin filaments from reactions performed with IQGAP1, mDia1(ΔDAD), and PFN1 was significantly reduced to 6.1 ± 0.7 (SE) subunits s$^{-1}$ μM$^{-1}$ (P < 0.0001), indicating this observation is reliant on a direct interaction between mDia1 and IQGAP1 (Fig. 4, A and D). This rate is not significantly different than reactions containing actin alone (P > 0.9999) or additional controls containing actin, IQGAP1, and PFN1 (P = 0.0057), which elongated at 4.8 ± 0.2 (SE) subunits s$^{-1}$ μM$^{-1}$ (Fig. 4 D). These results presented several exciting questions: were the filaments elongating at a rate consistent with IQGAP1 dictating plus-end behaviors or is the rate reflective of actin elongation in the absence of end-binding proteins?

Thus, we used multiwavelength TIRF microscopy to directly visualize the impact of 488-IQGAP1 on 549-mDia1(ΔDAD) filament assembly at plus-ends (Fig. 4 E). As expected in the absence of 488-IQGAP1, molecules of 549-SNAP-mDia1(ΔDAD) tracked the growing plus ends of actin filaments (Fig. 4, E and E′; and Video 5). We visualized instances where the apparent colocalization of IQGAP1 on an end directly preceded the loss of 549-SNAP-mDia1(ΔDAD) (Fig. 4, E and E′; and Video 5). We tracked and quantified the duration of the plus end occupancy of 549-SNAP-mDia1(ΔDAD) molecules in various actin assembly conditions in the absence or presence of 488-IQGAP1 (Fig. 4 F). Molecules of 549-mDia1(ΔDAD) are less processive and more frequently displaced from plus ends in reactions that contain 488-IQGAP1, particularly under conditions that promote fast-formin assembly (i.e., in the presence of PFN1), where the maximum duration of fast-formin growth declined from 635 to 125 s when IQGAP1 was included (Fig. 4 F). This effect did not occur in reactions using 549-SNAP-mDia1(FH1-C), which does not bind to IQGAP1 (Fig. S3, G and H; and Video 6). Further, the maximum occupancy of formin was only reduced by 50 s in the presence of molecules lacking plus-end pausing activities (i.e., 488-IQGAP1(BAD)) (Fig. 4 F and Video 5). Taken together, these observations suggest that IQGAP1 acts as a displacement factor for formin, not as an elongation-pausing protein, and this activity relies on the direct interaction of IQGAP1 and mDia1.

## IQGAP1 promotes the dynamic exchange of end-binding proteins

IQGAP1 displaces mDia1 from filament ends (Fig. 4, D–F). However, this specific interaction is one of many other competing regulators vying to manage the dynamics occurring at plus ends. The factor(s) that "win" this fast-paced game each produces vastly different consequences for filament length, filament stability, mechanisms of turnover, and overall array architecture (Courtemanche, 2018; Romet-Lemonne and Jégou, 2021; Shekhar et al., 2016). One notable example is the epic "tug-of-war" between mDia1 and the canonical capping factor CP where the polymerization fate of individual actin filaments is resolved in "decision complexes." The formin "wins" when filament growth resumes (CP dissociates), whereas formin "loses" when it is evicted from the plus end by the beta-tentacle of CP resulting in no additional growth (Bombardier et al., 2015; Funk et al., 2021; Hoeprich et al., 2022; Shekhar et al., 2015; Maufront et al., 2023). Given the direct relationship between IQGAP1 and mDia1, we were curious whether the presence of IQGAP1 tipped the balance in favor of formin or CP and whether IQGAP1 could displace CP or mDia1-CP decision complexes.

We first tested the hypothesis that IQGAP1 could influence the plus end dynamics regulated by the mDia1-CP "decision complex" using pyrene fluorescence assays that contained preformed actin filament (F-actin) seeds and PFN1 (Fig. 5, A–C). The formin mDia1(ΔDAD) promoted actin assembly via efficient elongation of the preformed F-actin seeds and reactions containing both formin and IQGAP1 were comparable or slightly reduced from these values (Fig. 5 A). Little actin assembly occurred in reactions containing CP or CP and IQGAP1, and these values were reduced compared with the IQGAP1 control (Fig. 5 B). Reactions probing the activity of the decision complex were

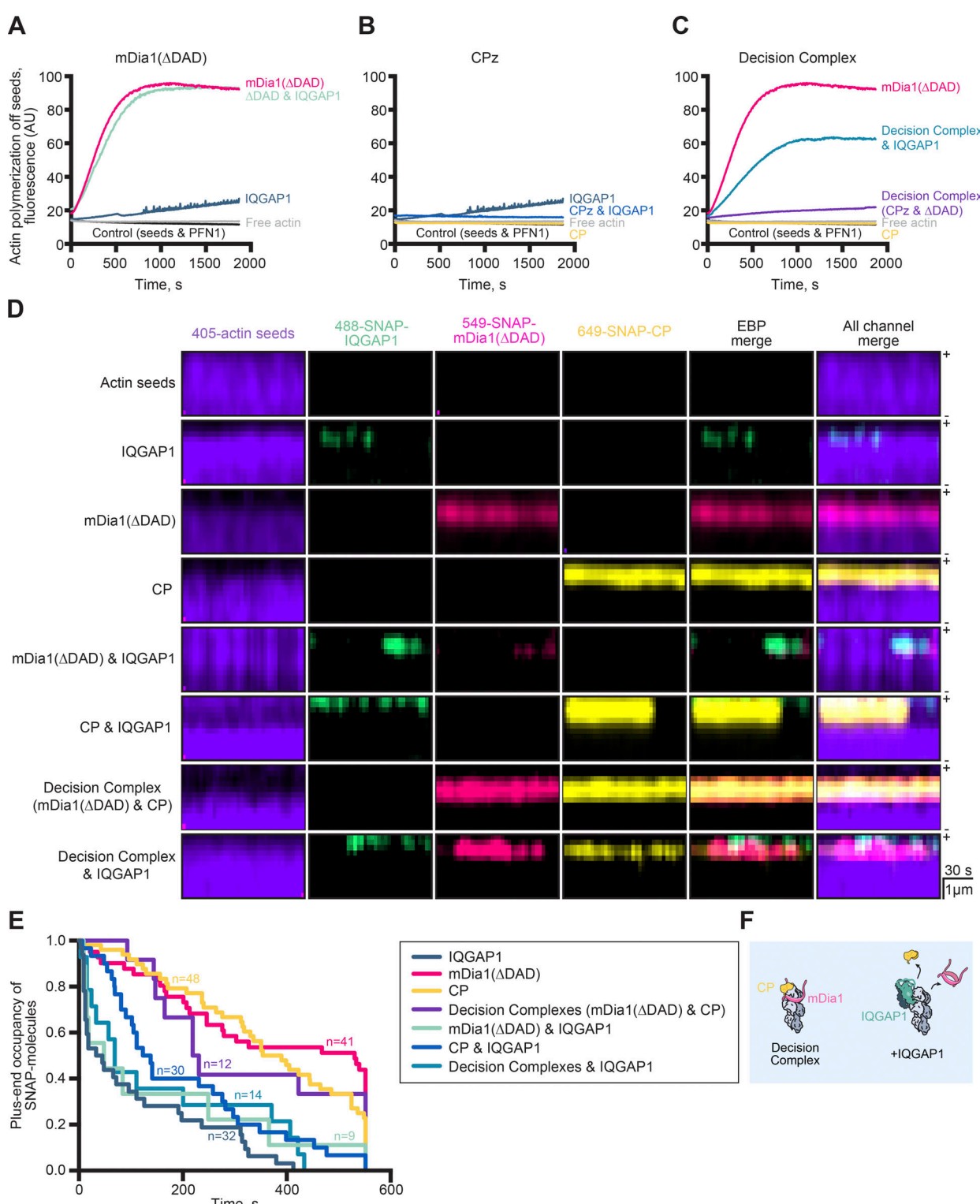

Figure 5. **IQGAP1 promotes the dynamic exchange of actin filament end-binding proteins. (A–C)** A representative seeded actin assembly assay showing the influence of IQGAP1 on actin assembly in the presence of (A) formin (mDia1(ΔDAD)) (B) Capping Protein (CP), and (C) formin-CP "decision complexes." Assays contain prepolymerized (unlabeled) actin seeds, 0.5 µM actin monomers (5% pyrene-labeled), 5 µM PFN1 and 75 nM IQGAP1, 2 nM mDia1(ΔDAD), or 10 nM CP, as noted. Curves in A–C were plotted for clarity and were generated from the same read on a plate reader (n = 3 total were performed). **(D)** Kymographs from four-color TIRF movies of stabilized actin filament seeds show the status of end binding proteins (EBPs) on actin filament plus ends. Individual channels, EBP merge (without actin seed), and the merge of all four wavelengths are shown. Reactions contain actin filament seeds (10% Dylight 405 label stabilized with 132 nM Alexa 405 phalloidin) and various combinations of 10 nM 549-SNAP-mDia1(ΔDAD), 10 nM 649-SNAP-Capping Protein (CP), and 10 nM 488-SNAP-IQGAP1. Scale: length, 1 µm; time, 30 s. **(E)** Survival plots of the plus-end occupancy of indicated proteins from reactions as in D (n = 10–48 molecules per condition, as stated, from n = 3 independent reactions). **(F)** Summary of IQGAP1 displacement activities at filament plus ends.

consistent with previous reports, with bulk fluorescence intermediate to that of mDia1 or CP alone (Fig. 5 C). There was a noticeable increase in actin filament polymerization when IQGAP1 was added to these reactions (Fig. 5 C). Taken together, these results may indicate IQGAP1 is displacing formin and decision complexes from plus ends.

To directly visualize actin filament plus-ends with each of these proteins (i.e., IQGAP1, mDia1, and CP), we performed four-color single-molecule TIRF microscopy. Although the 405-labeled actin elongated at rates comparable to other actin probes (Fig. S2, C and D), it was prone to rapid photobleaching, and we were unable to visualize 405-filaments in the presence of formin and profilin. Consequently, we used actin filament seeds stabilized with Alexa 405 phalloidin and kymographs to visualize the association and disassociation of combinations of 488-SNAP-IQGAP1, 549-SNAP-mDia1(ΔDAD), and 649-SNAP-CP molecules on plus ends (Fig. 5 D). Molecules of 549-mDia1(ΔDAD), 649-SNAP-CP, and decision complexes each bound plus ends stably (i.e., for minutes), whereas molecules of 488-SNAP-IQGAP1 bound plus ends transiently (Fig. 5 D). However, combinations of SNAP-labeled proteins with IQGAP1 resulted in less stable end interactions and often displacement of all end binding proteins (Fig. 5 D and Video 7). This was further analyzed using survival plots to quantify the length of end association (Fig. 5 E). Thus, IQGAP1 is an end displacement factor that may regulate actin dynamics by promoting the exchange of diverse plus-end regulators (Fig. 5 F).

## IQGAP1-mediated actin regulation contributes to normal cell activities

The plus ends of actin filaments are critical for many cellular features including cell morphology and migration (Shekhar et al., 2016; Svitkina, 2018), and past studies have noted differences in these processes in NIH-3T3 cells upon reduction of IQGAP1 levels (Sharma and Henderson, 2007; Arora et al., 2020; Choi et al., 2013; Brandt et al., 2007). Thus, to explore these concepts in a more physiological setting, we purchased and screened mouse NIH-3T3 fibroblasts lacking IQGAP1 (Fig. S4, A–F). We used a combination of reverse genetics and light microscopy to assess IQGAP1's plus end activities on actin dynamics in cells. First, two clonal knockout lines were identified via Western blot (Fig. S4 A) and transfected with mock treatments (i.e., transfection reagents but no plasmid), plasmids harboring human IQGAP1 (untagged or SNAP-tagged), or human IQGAP1(BAD) (Fig. S4, B–F). Notably, by immunofluorescence or live-cell screening, mean transfection efficiencies were >77% (Fig. S4, C–F). We used cell morphology and cell migration assays to assess IQGAP1's role in cellular actin because both processes require functional actin dynamics and can be resolved with our microscope capabilities (Fig. 6). We measured circularity to assess whether the loss of IQGAP1 or the expression of various IQGAP1 plasmids in the knockout lines influenced cell morphology (Fig. 6, A and B). Cells expressing endogenous levels of mouse IQGAP1 were relatively circular with an average circularity measurement of 0.69 ± 0.01 (SE), whereas IQGAP1 knockout (mock-treated) cells were the least circular with a mean measurement of 0.56 ± 0.02 (SE) (P < 0.0001). Cells

expressing untagged human IQGAP1 or SNAP-IQGAP1 plasmids were not significantly different from each other (P = 0.7721). However, these cells were significantly less circular than cells expressing endogenous mouse IQGAP1 (P = 0.0011 and P = 0.0028, respectively), and significantly more circular than knockout cells (mock) (P = 0.0364 and P = 0.0151, respectively). IQGAP1 knockout cells expressing SNAP-IQGAP1(BAD) seemed to have a more protrusive phenotype with significantly less circularity compared with endogenous (P < 0.0001), IQGAP1 (P = 0.0280), and SNAP-IQGAP1 (P = 0.0122) controls, but not the knockout control (P = 0.7737). Thus, IQGAP1 and its plus-end activities contribute to the morphology of cells.

To explore these changes more closely and to standardize the different treatments, we plated cells on crossbow-shaped fibronectin micropatterns to further assess their morphology (Fig. S4, G and H) and to visualize the subcellular cytoskeletal arrays (Fig. 6; and Fig. S4, I and J). Cells expressing endogenous IQGAP1 had significantly less pixel area (i.e., morphology) than mock-treated knockout cells (P = 0.0219) but were not significantly different than knockout cells expressing the tag-free (P = 0.8527) or SNAP-tagged IQGAP1 on plasmids (P = 0.7328) (Fig. S4, G and H). Knockout cells expressing SNAP-IQGAP1(BAD) were not significantly different from the knockout (P = 0.8527) but, consistent with circularity measurements above, covered more area than endogenous (P = 0.0011), tag-free IQGAP1 (P = 0.0031), or SNAP-IQGAP1 (P = 0.0219) (Fig. S4, G and H). We extended this analysis to actin filament arrays by measuring the total fluorescence of actin arrays stained by phalloidin, which was significantly reduced in IQGAP1 knockout cells (P = 0.0043) (Fig. 6, C and D) and rescued by human IQGAP1 on plasmids (P = 0.9809 and P = 0.9963 for IQGAP1 and SNAP-IQGAP1, respectively). Actin filament arrays in SNAP-IQGAP1(BAD) cells were not different from knockout cells (P = 0.7414) but were significantly less abundant than IQGAP1 (P = 0.0146) or SNAP-IQGAP1 controls (P = 0.0209) (Fig. 6, C and D). Microtubule arrays were significantly reduced with the loss of IQGAP1 (P < 0.0001) (Fig. S4, I and J). Compared with endogenous, this phenotype could be rescued by IQGAP1 (P = 0.9878) or SNAP-IQGAP1 (P = 0.3381), and even the SNAP-IQGAP1(BAD) plasmid (P = 0.1534) (Fig. S4, I and J). Thus, the effects of the two cysteine mutations in IQGAP1(BAD) are specific to the regulation of actin dynamics and do not extend to microtubules. Taken together, this demonstrates that regulation of actin-filament plus ends by IQGAP1 shapes the overall architecture of actin filaments and ultimately the higher-order morphology of cells through its plus-end displacement activity.

To assess the effect of IQGAP1 or IQGAP1(BAD) on cell migration, we used assays measuring wound closure from near confluent dishes of NIH-3T3 cells expressing endogenous mouse IQGAP1, mock-treated knockout, and knockout cells expressing human IQGAP1 (no tag and SNAP-IQGAP1) or SNAP-IQGAP1(BAD) on plasmids (Fig. 6, E and F). The mean closure percentage of endogenous, IQGAP1, or SNAP-IQGAP1 was not significantly different from each other at 12 h after the wound event. However, IQGAP1 knockout (mock) wounds did not close efficiently, displaying significantly less closure than endogenous (P = 0.0229), IQGAP1 (P = 0.0319), or SNAP-IQGAP1 (P =

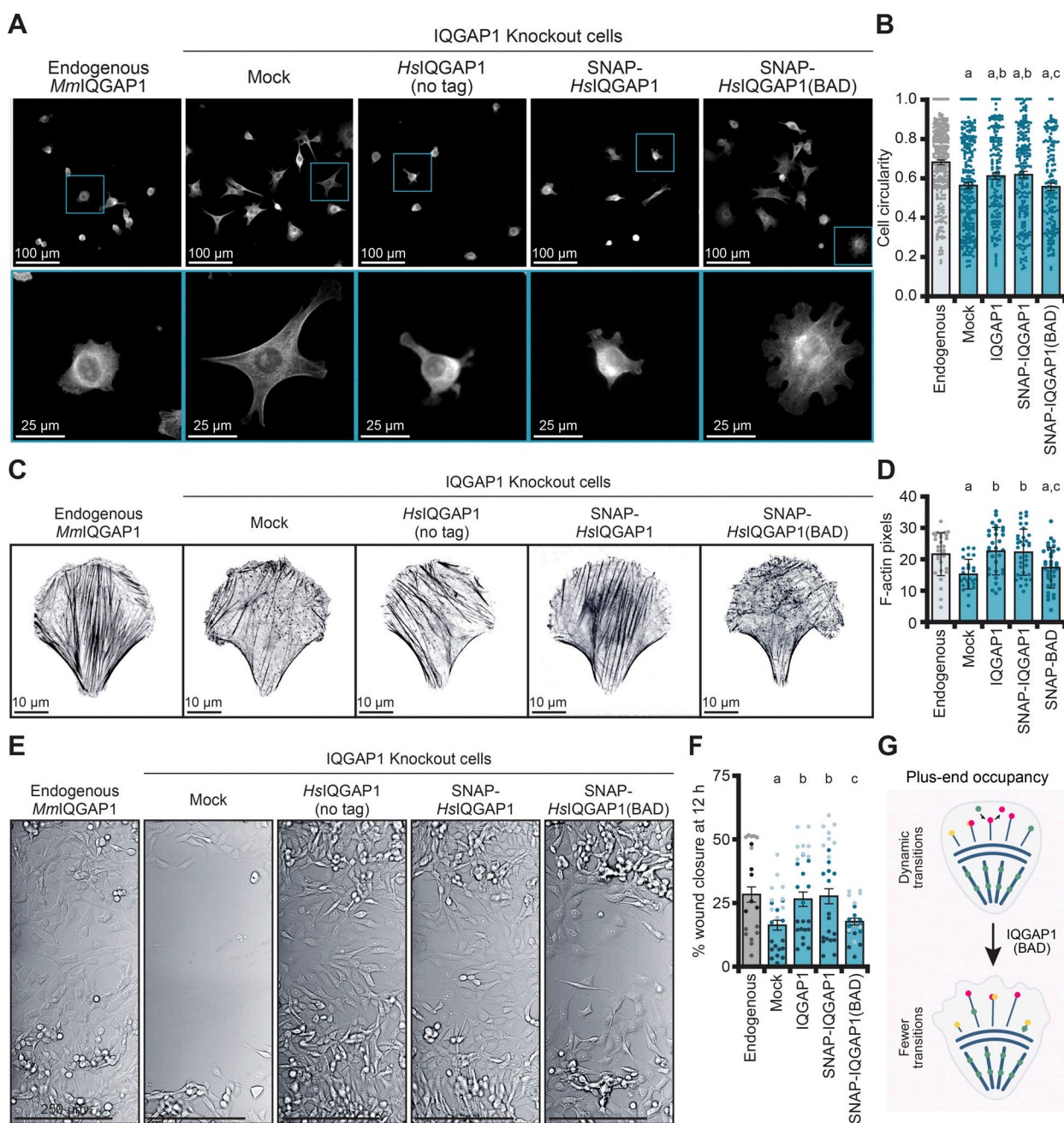

Figure 6. **Actin filament dynamics regulated by IQGAP1 influence the morphology and migration of NIH-3T3 cells. (A)** Representative images of phalloidin-stained actin filaments from NIH-3T3 (expressing endogenous *Mm*IQGAP1) compared with IQGAP1 knockout cells lacking (mock) or expressing *Hs*IQGAP1 (untagged and SNAP-tagged) and SNAP-*Hs*IQGAP1(BAD). Zoomed insets (bottom panels) highlight single cells. Scale: top, 100 µm; bottom, 25 µm. **(B)** Cell circularity measurements from cells in A (n = 151–261 cells, pooled from 3 different coverslips). Error bars, SE. Statistics, ANOVA: (a) P ≤ 0.05 compared with endogenous; (b) P ≤ 0.05 compared with mock (IQGAP1 knockout cells); (c) P ≤ 0.05 compared with IQGAP1 knockout cells expressing a SNAP-IQGAP1 plasmid; ns, P ≥ 0.05 compared with IQGAP1 knockout cells expressing an untagged IQGAP1 plasmid. **(C)** Cells as in A, plated on micropatterns. Scale, 10 µm. **(D)** Quantification of total actin signal from cells in C (n = 29–35 cells per condition). Error bars, SD. Statistics, ANOVA: comparisons as in B. **(E)** Mock and IQGAP1(BAD) cells do not migrate as effectively as cells expressing IQGAP1 in wound healing assays. Representative images from the same trial are shown 12 h after wounding event. Scale, 250 µm. **(F)** Quantification of wound healing assays in E. Dots represent normalized closure values for FOVs along the wound (n = 7–11 FOVs) from each condition, with shading grouped by n = 3 independent replicates. Statistics, ANOVA with comparisons as in B. **(G)** Model of IQGAP1 plus-end activities in cells. Disruption of IQGAP1 (green) results in less turnover of actin filament end binding proteins including formin (pink) and capping protein (yellow). Fewer IQGAP1-mediated transitions promote aberrant cell morphology and actin structures as filaments suffer prolonged capping or growth events.

0.0124) controls (Fig. 6, E and F). Knockout cells expressing SNAP-IQGAP1(BAD) had significantly less closure than SNAP-IQGAP1 cells (P = 0.0478) and did not display significantly more closure than mock-treated cells (P = 0.9919) (Fig. 6, E and F). This demonstrates that normal plus-end IQGAP1 function is necessary for cell movements. Further differences shown by IQGAP1(BAD) indicate that IQGAP1-mediated actin filament plus-end displacement regulates actin dynamics in essential cell processes (Fig. 6 G).

## Discussion

Actin polymerization is regulated by vastly different and often opposing classes of plus-end binding proteins and protein complexes that stimulate, arrest, or pause filament growth. This feature has remarkable consequences for cellular processes and behaviors, as plus-end protein processivity dictates the physical properties and structural dimensions of cellular actin arrays (Funk et al., 2021; Breitsprecher and Goode, 2013; Henty-Ridilla et al., 2016; Wirshing et al., 2023; Ulrichs et al., 2023; Bombardier et al., 2015; Shekhar et al., 2015; Hoeprich et al., 2022; Efimova et al., 2020). Filament assembly is further complicated by disassembly factors, end-blocking proteins, and proteins that limit the availability of actin monomers (Pollard, 2016; Pimm et al., 2020; Skruber et al., 2018). How this actin assembly paradox is resolved in cells remains unclear. Here, we examined the role of IQGAP1 in actin filament assembly and identified two amino acids that are necessary for plus end–related activities. Our four-color TIRF microscopy with purified proteins suggested that IQGAP1 is a displacement factor able to promote a more rapid exchange of formin (mDia1), CP, and formin-CP "decision complexes." This feature may be useful in promoting the transition of various proteins present on actin filament ends or for switching between periods of filament growth or disassembly in cells (Fig. 6 G). This idea is reinforced by our data in cells, where the loss of IQGAP1 interactions with actin filament plus-ends via IQGAP1(BAD) resulted in a significant departure from the normal architecture of actin filament arrays (Fig. 6 C), cell morphology (Fig. 6, A and B; and Fig. S4, G and H), and cell migration (Fig. 6, E and F).

Here, we used multicolor TIRF microscopy assays to help unravel the complex interactions of proteins at actin filament plus-ends. TIRF microscopy assays can be challenging and are not always directly comparable across experiments due to the different assembly dynamics of different labels on actin (Fig. S2, C and D) (Kovar et al., 2006; Amann and Pollard, 2001; Malm, 1984), actin concentrations, filament tethering styles (i.e., biotin-streptavidin, NEM-myosin, spectrin seeds, poly-L-lysine, etc), or experimental setups (i.e., open flow or constant-flow) (Hoeprich et al., 2022; Henty-Ridilla, 2022; Shekhar, 2017; Jégou et al., 2011). Here, we use an "open flow" based system with biotin–streptavidin linkages (roughly 1.3 linkages per 1 μm filament) to tether actin filaments within the TIRF imaging plane. The advantages include analysis from whole fields of view, small (<100 μl) reaction volumes, and filaments are not under any known pulling forces that influence actin and protein dynamics (Cao et al., 2018; Jégou et al., 2013; Hayakawa

et al., 2011; Sun et al., 2020; Winkelman et al., 2020; Sun and Alushin, 2023). However, single actin filaments imaged in our system are not conducive to high-throughput kymograph analysis as in a constant flow system and are currently obtained through time-consuming measurements by hand. Additionally, though not measured in association with actin filaments, unbound/inactive molecules can contribute noise to the image background. Even these caveats do not detract from the power of "seeing" the direct confirmation of protein localization or activity of a purified protein. For example, multiple interpretations can be gleaned from the seeded pyrene actin assembly assays (Fig. 5, A–C), including that IQGAP1 blocked rather than displaced proteins on filament ends. Employing multicolor (two, three, and four-color) TIRF with this orthogonal method gave us more information about the bigger picture of plus-end assembly—the mechanism was not end-capping but rather end displacement–based.

Is IQGAP1 truly bound at or near actin filament plus ends? While a focus of this work was dissecting IQGAP1's end-binding role, IQGAP1 also uses CHD-mediated side-binding activity to bundle actin filaments (Palani et al., 2021; Pelikan-Conchaudron et al., 2011; Hoeprich et al., 2022; Mateer et al., 2004; Wang et al., 2023). A similar question was posed for mDia1-CP decision complexes (Bombardier et al., 2015; Shekhar et al., 2015) and will likely require similar high-resolution studies for unambiguous confirmation (Maufront et al., 2023). The exact mechanism of IQGAP1-mediated pausing or displacement is unclear. Could the mechanism be as simple as steric hindrance of end-binding proteins by dimers of IQGAP1? Do direct interactions with protofilament ends or lateral interactions with terminal subunits mediate the pauses/displacement? Each individual component of mDia1-CP decision complexes is associated with a different subunit at the plus-end (Maufront et al., 2023; Bombardier et al., 2015). When formin "steps," the beta-tentacle of CP can slip into this binding region to displace formin (Kovar et al., 2006; Romero et al., 2004; Bombardier et al., 2015; Maufront et al., 2023). It is possible that IQGAP1 displaces these complexes through several different mechanisms including competing with mDia1 or CP for an actin filament binding site or through interactions with individual components of the decision complex. In our study, plus-end displacement of formins from actively polymerizing actin filaments required direct interactions between IQGAP1 and mDia1 (Fig. 4 D). While formins, particularly mDia1, have increased processivity and affinity for plus-ends in the presence of profilin (Cao et al., 2018; Kovar et al., 2006; Romero et al., 2004), we did not observe significant changes to IQGAP1-mediated end displacement comparing profilin conditions (Fig. 4 D and Fig. S3 D) or using non-polymerizing actin filament seeds (Fig. 5 D). Additional experiments using IQGAP1(BAD) or the side-binding deficient IQGAP1(160-end) further suggest that IQGAP1 physically binding to the plus end plays a role in the displacement mechanism (Fig. 4, E and F). In sum, at this resolution, we are unable to truly discern whether decision complex displacement occurs from a plus-end binding affinity-based mechanism or whether IQGAP1–formin interactions "pull" the complex from the plus end.

Historically, IQGAP1 has been characterized as a transient capping factor (Hoeprich et al., 2022; Pelikan-Conchaudron

et al., 2011). Here, we classify it as a displacement factor. Semantics aside, are these activities relevant in cells? IQGAP1 is localized to sites of meticulous actin filament end regulation, including in filopodia (Jacquemet et al., 2019), along stress fibers (Samson et al., 2017), and at the leading edge (Brandt et al., 2007; Chen et al., 2020). The highest concentration of filament ends exists in the lamellipodium, which has an estimated 500 actin filaments per μm squared (Raz-Ben Aroush et al., 2017; Svitkina et al., 1997; Abraham et al., 1999). When plus-end factors are absent, there would be 1,720–5,000 free ends in an average lamellipodium ($1 \times 10 \times 0.2$ μm) (Raz-Ben Aroush et al., 2017; Svitkina et al., 1997; Abraham et al., 1999). If we consider plus-end factors (and assume they are all active), 1,200 ends will be occupied by CP (1 μM) (Pollard and Borisy, 2003), <60 ends will be occupied by mDia1 dimers (<100 nM) (Chhabra et al., 2009), leaving as few as 460 free ends that could be occupied by 243 dimers of IQGAP1. Considering IQGAP1 side-binding affinity (47 μM) (Mateer et al., 2004) and other regulators that also bind these proteins, like twinfilin (602 molecules in this space) (Johnston et al., 2018) or other formins like INF2 (180 dimers) (Chhabra et al., 2009), there may not be enough free ends to bind all the regulators. However, evidence indicates that multiple regulators co-occupy filament plus-ends (Ulrichs et al., 2023; Funk et al., 2021; Alimov et al., 2023; Towsif and Shekhar, 2023, Preprint; Bombardier et al., 2015; Shekhar et al., 2015; Wirshing et al., 2023), and our work suggests that IQGAP1 joins several factors present there (Brown and Sacks, 2006; Hedman et al., 2015; White et al., 2012). This may explain how the IQGAP1(BAD) substitution mutant displayed significant perturbation to actin-based cell processes. Specifically, cells may not migrate as efficiently because formin-engaged filaments are overgrowing, and CP-subdued ends are being capped for too long. Intriguingly, IQGAP1 activities in cells are further regulated by calmodulin (CaM) and one of the two residues necessary for plus-end activities (C756) is present at the predicted binding site (Pelikan-Conchaudron et al., 2011; Zhang et al., 2019). Unraveling these molecular details provides a foundation for future studies to examine how additional plus-end regulators and IQGAP1 ligands further influence actin filament assembly.

## Materials and methods

### Reagents

All chemicals were obtained from Thermo Fisher Scientific unless otherwise stated. Synthesis of cDNA, plasmid construction, subcloning, site-directed mutagenesis, and plasmid sequencing were performed by Genscript unless otherwise indicated.

### Plasmid construction

The coding sequence for full-length human IQGAP1 (UniProt: P46940) was synthesized with a silent basepair substitution (c4486a) to remove a native KpnI restriction site and cloned into a modified 6×His-SUMO-containing pET23b vector (Pimm et al., 2022), flanked by AgeI and NotI sequences. IQGAP1 truncation mutants were subcloned into the pET23b backbone at the same sites. Substitution mutations were generated in IQGAP1(745–1,450)

or the full-length protein using site-directed mutagenesis to create: IQ1 (A754P, R757A, L760A), IQ1-2 (A754P, R757A, L760A, Q783A, W786A, R787A, K790A), IQ3 (H819A, R822A R826A), and IQGAP1(BAD) (C756A, C781A). Formin constructs, 6×His-SUMO-tagged mDia1(ΔDAD) (amino acids 1–1,175), and 6×His-SUMO-tagged mDia1(FH1-C) (amino acids 571–1,256) were synthesized and cloned into the pET23b vector between the AgeI and NotI sites with a methionine added before the FH1-C insert. A cassette containing a 6×His-SUMO sequence followed by a ULP1 cut site and the coding sequence of a SNAP-tag flanked by NsiI and AgeI was synthesized. This cassette was used to generate the following SNAP-tagged protein constructs: IQGAP1, IQGAP1(BAD), IQGAP1(160-end), mDia1(ΔDAD), and mDia1(FH1-C). Mammalian cell expression plasmids driven by the CMV promoter were generated via subcloning into pcDNA vectors, with untagged IQGAP1 inserted between KpnI and NotI of pcDNA3.1(+) and IQGAP1 or IQGAP1(BAD) inserted between AgeI and NotI of pcDNA5 with the SNAP-tag.

### Protein purification, labeling, and handling

All 6×His-SUMO-tagged IQGAP1 proteins were expressed in Rosetta2(DE3) pRare2 cells (MilliporeSigma) and induced with 0.4 mM Isopropyl β-D-1-thiogalactopyranoside (IPTG) at 18°C for 22 h. Cell pellets were resuspended in buffer (2× PBS [280 mM NaCl, 5 mM KCl, 20 mM sodium phosphate dibasic, 0.35 mM potassium phosphate monobasic, pH 7.4], 500 mM NaCl, 0.1% Triton X-100, 14 mM β-mercaptoethanol [BME], 10 μg/ml DNase I, and 1× protease inhibitor cocktail [0.5 μg/ml leupeptin, 1,000 U/μl aprotinin, 0.5 μg/ml pepstatin A, 0.5 μg/ml chymostatin, 0.5 μg/ml antipain]), and lysed with 1 mg/ml lysozyme for 20 min and probe sonication at 100 mW for 90 s. Clarified lysates were applied to cobalt affinity columns (Cytiva), equilibrated in 1× PBS (pH 7.4), 500 mM NaCl, 0.1% Triton X-100, 14 mM BME, and eluted with a linear imidazole (pH 7.4) gradient (0–300 mM). The 6×His-SUMO-tag was cleaved for 30–60 min at room temperature using 5 μg/ml ULP1 protease and the final protein was gel filtered over a Superose 6 Increase (10/300) column (Cytiva) equilibrated with 1× PBS (pH 8.0), 150 mM NaCl, 14 mM BME. Pooled fractions were concentrated with appropriately sized MWCO centrifugal concentrators (MilliporeSigma), aliquoted, flash frozen, and stored at –80°C until use.

Purification of 6×His-mDia1(FH1-C) or 6×His-mDia1(ΔDAD) was conducted as described above for IQGAP1. Previously published actin-binding proteins (and the related expression constructs) and rabbit muscle actin (RMA; labeled and unlabeled) were purified according to detailed and well-described protocols, referenced as follows: 6×His-SNAP-CP$_{(α1β1)}$ (Bombardier et al., 2015; Johnston et al., 2018; Soeno et al., 1998), PFN1 (Pimm et al., 2022; Liu et al., 2022), unlabeled (Spudich and Watt, 1971; Cooper et al., 1984; Kovar et al., 2003), or labeled RMA (Cys$_{374}$: Oregon Green [OG] 488 iodoacetamide, DyLight-405 maleimide, or N-(1-pyrene) iodoacetamide [pyrene] [Kuhn and Pollard, 2005; Cooper et al., 1983]; all lysine residues via NHS ester: Alexa Fluor 488 or 647 [Hertzog and Carlier, 2005; Pimm et al., 2022]). Biotin–actin was purchased from Cytoskeleton Inc. A brief description of how these well-established

purifications were executed is as follows: RMA was purified from acetone powder previously stored at –80°C. Acetone powder was ground, rehydrated in G-buffer (3 mM Tris pH 8.0, 0.5 mM DTT, 0.2 mM ATP, 0.1 mM CaCl$_2$), and cleared via centrifugation. Actin was polymerized at 4°C overnight and pelleted. The pellet was prepared via dounce homogenization and dialyzed against G-buffer for 2 days, with buffer exchanges every 24 h, cleared via ultracentrifugation, and gel filtered on a 16/60 S200 column (Cytiva). Pyrene-, OG-, and DyLight 405-actin were prepared from RMA pellets (as above), first dialyzed against G-buffer lacking DTT for 4 h, then diluted to 1 mg/ml, polymerized, and labeled with 7–10-fold molar excess dye in 25 mM imidazole (pH 7.5), 100 mM KCl, 0.15 mM ATP, and 2 mM MgCl$_2$ overnight. Actin filaments were pelleted and subjected to the same dialysis and gel filtration treatments as the unlabeled RMA above. Fluorescence RMA labeled on lysine residues was made similarly except the initial dialysis was performed using HEPES-buffered G-buffer (3 mM HEPES [pH 8.2], 0.5 mM DTT, 0.3 mM ATP, 0.1 mM CaCl$_2$) and labeled on actin filaments in 3 mM HEPES (pH 8.2), 0.5 mM DTT, 0.3 mM ATP, 1 mM MgCl$_2$, and 50 mM KCl. Profilin and CP were purified from previously frozen *E. coli* pellets induced and stored as above. Cells from both pellets were lysed via sonication in the presence of lysozyme and protease inhibitors and precleared via centrifugation. Profilin lysates were subjected to ion exchange chromatography via a 5 ml HiTrap column (Cytiva) over a 30 ml 0–500 mM KCl gradient in 50 mM Tris (pH 8.0), 50 mM KCl, and 1 mM DTT. Peak fractions were pooled and gel-filtered on a Superdex 75 Increase (10/300) column (Cytiva) in 50 mM Tris (pH 8.0), 50 mM KCl, and 1 mM DTT. CP lysates were loaded onto a different 5 ml HiTrap column (Cytiva) and subjected to a 45 ml salt gradient (0–500 mM KCl) in 20 mM Tris (pH 8.0). Peak fractions were pooled (and labeled as described for SNAP-IQGAP1, below) and then gel filtered on a Superdex 75 Increase (10/300) column (Cytiva) into 20 mM Tris-HCl, pH 8.0; 50 mM KCl, 1 mM DTT. Peak fractions for profilin or CP were pooled, flash-frozen, and stored at –80°C.

All purified SNAP-tagged proteins were labeled with 10-fold molar excess of SNAP-Surface dyes (New England Biolabs) in a labeling buffer (1× PBS, 150 mM NaCl, 300 mM imidazole [pH 8.0], 0.1% Triton X-100, and 10 mM DTT) for 1 h at room temperature, then the 6×His-SUMO-tag was cleaved, and each protein was gel filtered over the Superose 6 column, as above. Pooled fractions were concentrated, aliquoted, flash-frozen, and stored at –80°C until use.

Protein concentrations and purity were determined by densitometry of Coomassie gels compared with a BSA standard curve. Labeling efficiency of directly labeled and SNAP-tagged proteins was calculated using spectroscopy and the following extinction coefficients: actin (unlabeled)$\varepsilon_{290}$, 25,974 M$^{-1}$ cm$^{-1}$; Oregon Green (OG)$\varepsilon_{496}$, 70,000 M$^{-1}$ cm$^{-1}$; pyrene$\varepsilon_{344}$, 26,000 M$^{-1}$ cm$^{-1}$, DyLight 405$\varepsilon_{400}$, 30,000 M$^{-1}$ cm$^{-1}$; AlexaFluor 488$\varepsilon_{495}$, 71,000 M$^{-1}$ cm$^{-1}$; SNAP-549$\varepsilon_{560}$, 140,300 M$^{-1}$ cm$^{-1}$; AlexaFluor 647$\varepsilon_{650}$, 239,000 M$^{-1}$ cm$^{-1}$, and SNAP-649$\varepsilon_{655}$, 250,000 M$^{-1}$ cm$^{-1}$. Correction factors used: OG, 0.12; AlexaFluor 488, 0.11; SNAP-549, 0.12; AlexaFluor 647, 0.03; and SNAP-649, 0.03. The percent label for each SNAP-tagged protein is as follows: 488-SNAP-IQGAP1: 70%; 488-SNAP-IQGAP1(BAD): 67 or 71%; 488-SNAP-IQGAP1(160-end): 55%; 549-SNAP-mDia1(Δ-DAD): 61, 66.5, or 70%; 549-SNAP-mDia1(FH1-C): 52 or 57%; and 647-SNAP-CP: 82%. Notably, we noticed a small yet significant reduction in the elongation rate of individual actin filaments labeled with Alexa 647 when compared with the more traditional label of actin on cysteine 374 with the Oregon Green label (P = 0.0021) (Fig. S2, C and D).

## Total internal reflection fluorescence (TIRF) microscopy

Slide cleaning, coating, conditioning steps, and the basic imaging setup were previously described (Henty-Ridilla, 2022). Briefly, imaging chambers were assembled from biotin-PEG silane-coated coverslips adhered to custom μ-slide VI bottomless Luer slides (IBIDI). Coverslips (24 × 60 mm, #1.5) were extensively cleaned and sonicated, then coated with 2 mg/ml mPEG-silane (MW 2,000; Laysan Bio Inc.) and 0.04 mg/ml biotin-PEG silane (MW 3,400; Laysan Bio Inc.) in 80% ethanol (pH 2.0), and evaporated in a 70°C incubator for 12–48 h before assembly. Coated coverslips were rinsed thrice with $_{dd}$H$_2$O, and imaging chambers were constructed by affixing coverslips via pieces of 0.12 mm SA-S-Secure Seal double-sided tape (Grace Bio-labs) flanking the long axis of wells. The short edge of each chamber was sealed with 5-min epoxy (Loctite). Imaging chambers were conditioned by flowing 50 μl of the following buffers in the following order: 1% BSA, 0.005 mg/ml streptavidin resuspended HEK buffer (20 mM HEPES [pH 7.5], 1 mM EDTA [pH 8.0], 50 mM KCl), incubated for 30–60 s, 1% BSA to reduce nonspecific binding, 1× TIRF buffer, and then the final reaction. All reactions were conducted at 20°C in TIRF buffer (final: 20 mM imidazole [pH 7.4] 50 mM KCl, 1 mM MgCl$_2$, 1 mM EGTA, 0.2 mM ATP, 10 mM DTT, 40 mM glucose, and 0.25% methylcellulose [4000 cP]), diluted from a 2× stock, with 1 μl of anti-bleach solution (10 mg/ml glucose oxidase and 2 mg/ml catalase), proteins of interest, and appropriate buffer controls for each assessed protein/combination). In addition, 45 nM biotin-actin was included in the actin stock for all TIRF-based assays to loosely anchor filaments to the biotin-streptavidin-coated slide surface. The final concentration of biotin–actin in the 1 μM total actin reaction is 3.6 nM.

Two TIRF microscopes were used in the experiments. Importantly, all figures and data comparing treatments were collected from the same microscope (i.e., differences reported are not due to different setups). Most experiments (and all four-color experiments; exceptions noted, below) were performed on a DMi8 microscope equipped with solid-state 405 nm (50 mW), 488 nm (150 mW), 561 nm (120 mW), and 647 nM (150 mW) lasers and matched filter sets/cubes (GFP-T, Cherry-T, Y5-T, and QWF-T [size P]) using a 100× Plan Apo 1.47 NA oil-immersion TIRF objective (Leica Microsystems). Images were captured in 5-s intervals for 20 min (unless otherwise noted) using LAS X software, and an iXon Life 897 EMCCD camera (Andor), with an (81.9 μm)$^2$ FOV. The percent laser and exposures were consistent for each experiment and typically 10% 405 50 ms, 10% 488 200 ms, or 10% 647 50 ms for actin labels and 10% 488 50 ms, 10% 561 50 ms, or 10% 647 50 ms for various SNAP-labels. Images were processed using Fiji software

(Schindelin et al., 2012) using a 50-pixel rolling-ball radius background subtraction and 1.0-pixel Gaussian blur.

Figure panels Fig. 1, D and E; Fig. 3, B, E, and F; Fig. S3, E, G, and G′; Fig. 4, B, E, and E′; Fig. S4 E; and Fig. 6, A and E were imaged using Ti2 motorized base iLAS TIRF system equipped with solid-state 488 nm (90 mW), 561 nm (70 mW), and 640 nm (65 mW) lasers and matched filter sets/cubes (C-NSTORM ultrahigh signal-to-noise 405/488/561/640 quad filter set, compatible epifluorescence cubes, DIC polarizers), using a 60× Apochromat 1.49 NA TIRF, DIC oil-immersion objective (Nikon Instruments Inc). DIC experiments were executed with an equipped SOLA LED box (Lumencor Inc). Time-lapse images were captured in 5-s intervals for 15 min (unless otherwise noted) using NIS-Elements Advanced Research Package software (with image denoise and deconvolution modules), and a 6.5 µm pixel Prime BSI express sCMOS camera (Photometrics Inc), with a (129.69 µm)$^2$ FOV. The percent laser and exposures were consistent for each experiment and typically 10% 488 100 ms, or 10% 640 100 ms for actin labels and 20% 488 200 ms, 20% 561 100 ms, or 10% 640 100 ms for various SNAP-labels. Images collected from the Nikon TIRF were denoised using the included software. Images were processed as above using Fiji software.

### Measurement of actin filament nucleation, elongation, and pauses to elongation

Actin nucleation was quantified as the total count of actin filaments per TIRF FOV present at 200 s following the addition of actin to start the polymerization reaction.

The mean actin filament elongation rates (subunits s$^{-1}$ µM$^{-1}$) were calculated by measuring the length (µm) of individual actin filaments present in TIRF movies over at least four different time points per filament. The slope (length over time; µm/s) was multiplied by 370 subunits (i.e., the number of subunits present per micron of actin (Pollard et al., 2000) and divided by the concentration (1 µM). Depending on conditions, hundreds of filaments can be present in a typical TIRF movie. Thus, 17 filaments (or all filaments present if fewer than 17; represented as dots in the figures) were measured per FOV, from at least three independent reactions. Elongation rates measured in Fig. S2 C were subject to single-blind analysis (i.e., the measurer did not have knowledge of the treatments being measured).

Pauses to filament elongation and their duration were measured from single-color (actin only) TIRF reactions from actin filament length over time plots. Pauses were measured as instances of stalled filament elongation for at least three consecutive imaging frames. Thus, the minimum pause duration we were able to resolve with the TIRF imaging setup was 15 s. The frequency distributions of pause durations (Fig. 1 I, Fig. 2 F; and Fig. S1, E and F) were displayed as best-fit values of a Gaussian distribution $Y = (A) * \left( e^{(-0.5)*\left(\frac{(x-|x|)}{SD}\right)^2} \right)$, where A is the amplitude of the peak and SD is the standard deviation of measured durations. For actin-alone controls or proteins unable to pause filament elongation, the data were not Gaussian and best modeled as exponential decay $Y = (Y_0 - D_{max}) * (e^{(-k*x)} + D_{max})$, where $D_{max}$ is the plateau and k is the exponential rate constant.

Kymographs of actin filament elongation were made using the KymographBuilder (release 1.2.4) plugin (https://github.com/fiji/KymographBuilder [Mary et al., 2016]) in Fiji software. To generate all kymographs, the line width was set to 3.0. Each kymograph was cropped consistently within the figure panels.

### Seeded and bulk pyrene assembly assays

Seeded pyrene actin filament assembly assays (Fig. 3 A and Fig. 5, A–C) were performed by combining 1 µM unlabeled Mg-ATP actin seeds with 0.5 µM Mg-ATP actin monomers (5% pyrene labeled), proteins or control buffers, and initiation mix (IM; 2 mM MgCl$_2$, 0.5 mM ATP, 50 mM KCl). Actin filament seeds were generated by polymerizing 10 µM actin in IM for 1 h at room temperature. Just prior to use, seeds were sheared via pipetting and then added to non-treated black microplate wells containing reaction components (proteins, buffers, and IM). To simultaneously initiate reactions, actin monomers present in different microplate wells were combined with reaction components with a multichannel pipette. Total fluorescence was monitored at the 365/407 nm spectrums using a plate reader (Tecan Inc.). Values in Fig. 3 A were averaged from $n = 3 \pm SD$ (shaded). Values shown in Fig. 5, A–C are representative, each plotted from the same single run in the plate reader.

Bulk actin filament assembly assays (Fig. S3, B and C) were performed by combining 2 µM Mg-ATP actin (5% pyrene labeled), proteins or control buffers, and IM. Reactions were initiated and monitored as above. Presented values were averaged from $n = 3 \pm SD$ (shaded).

### Step photobleaching predictions and analysis of IQGAP1 oligomeric state

To determine predicted photobleaching steps, the expression $(X + Y)^n$ for $n$ = each hypothetical oligomerization state was expanded, then the expanded polynomial was solved using the calculated labeling efficiency of 488-SNAP-tagged IQGAP1 proteins, with $X$ representing the percent of visible molecules, and $Y$ representing the percent of unlabeled molecules. Calculations were performed to predict monomer, dimer, trimer, and tetramer states using WolframAlpha (https://www.wolframalpha.com/) to see which best modeled the data (Hoeprich et al., 2022; Breitsprecher et al., 2012). The labeling efficiency used for these calculations was as follows: 70% labeled 488-SNAP-IQGAP1, 71% labeled 488-SNAP-IQGAP1(BAD), and 55% labeled 488-SNAP-IQGAP1(160-end) (shown as symbols in Fig. 3 D). The observed number of photobleaching events was measured from TIRF movies generated from reactions containing 1 nM of each protein resuspended in 1× TIRF buffer lacking anti-bleach components (i.e., glucose oxidase or catalase). Samples were flowed into imaging chambers and allowed to settle for 15 min before observation as surface-adsorbed molecules. Samples were imaged on the Nikon TIRF microscope at 99% 488 laser power under continuous acquisition for 2 min. Stepwise reductions in integrated fluorescence (Fig. 3, B and C) of $n \geq 300$ individual molecules per condition were scored by hand and used to generate the histograms in Fig. 3 D from $n = 3$ reactions per condition (dots). These values were compared with

mathematical predictions and are consistent with 488-IQGAP1 and 488-IQGAP1(BAD) molecules existing as dimers (Hoeprich et al., 2022; Liu et al., 2016; Ren et al., 2005). We were unable to conclusively determine the oligomeric state of 488-IQGAP1(160-end) given its lower labeling efficiency.

### Single-molecule colocalization analysis

Quantification of colocalized 488-SNAP-IQGAP1 proteins with 549-SNAP-mDia1 proteins was determined by mixing equal stoichiometries (1 nM) of noted proteins in 1× TIRF buffer. Following 15 min incubation, reactions were flowed into imaging chambers and imaged via Nikon TIRF in 2 s intervals (200 ms exposure for each laser) for 5 min. Individual molecules were detected using frame 30 (60 s into the 5-min imaging period), using the ComDet v.0.5.5 FIJI plugin (https://github.com/ekatrukha/ComDet), with particle size set to five pixels and intensity threshold set to 4. Dots in Fig. 4 C and Fig. S3 F are the percentage of colocalized molecules calculated for entire FOVs ($n$ = 9 FOVs total, pooled from three replicates with $n \geq 150$ molecules present per FOV). Proteins were precleared via centrifugation at 20,000 × $g$ for 5 min before use. 1–5% 549-signal can be attributed to background.

### Actin filament plus-end occupancy of single molecules derived from survival plots

To quantify the single molecules and complexes of IQGAP1 and/or mDia1 on actin filament plus ends, multiwavelength TIRF microscopy (Nikon) was performed with 1 µM actin (10% Alexa-647 label; 3.6 nM biotin-actin) and 1 nM SNAP-tagged 488-IQGAP1 or 549-mDia1 proteins. The occupancy of growing plus-ends with 549-mDia1(ΔDAD) or 549-mDia1(FH1-C) in the presence and absence of IQGAP1 proteins was scored from TIRF movies monitored in 5-s intervals for 15 min from six FOVs from $n$ = 2 different reactions. Occupancy of 549-mDia1 molecules (dwell time) was plotted as the percent occupying ends over time (i.e., each step represents a percentage of molecules disassociating from filament ends).

The plus-end occupancy of IQGAP1, formin, CP, or formin-CP decision complexes was determined using four-color TIRF microscopy (Leica) with 405-actin filament seeds. 1 µM actin (10% 405-label; 3.6 nM biotin-actin) was polymerized in 1× TIRF buffer in TIRF chambers. After 2.5 min, filament polymerization was stopped and the free monomers were removed by washing the chamber and remaining biotin-actin seeds thrice with 1× TIRF buffer. Reactions containing noted SNAP-labeled molecules or formin-CP decision complexes and 132 nM 405-Alexa phalloidin were preincubated for 5 min on ice and then flowed into the imaging chamber and monitored every 3.2 s for 10 min for $n$ = 3 different reactions. Occupancy of various molecules (dwell time) was plotted as the percent occupying ends over time.

### Cell growth, screening, and transfection

A pooled CRISPR knockout line and comparable wildtype NIH-3T3 cells expressing endogenous IQGAP1 were purchased from Synthego. Cells were grown in DMEM (Gibco) supplemented with 10% FBS (Genesee Scientific), 200 mM L-glutamine (Gibco), and 45 U/ml penicillin-streptomycin (Gibco). All cell

lines were screened for mycoplasma at regular intervals by screening fixed cells for irregular DAPI stain. Clonal lines were produced via dilution by plating 0.5 cells per 48-well plate (Genesee Scientific) and visual confirmation of single-cell deposition. Wells containing cells were grown in conditioned media, grown to ~70% confluency, and screened for protein levels via Western blots with monoclonal mouse anti-IQGAP1 (1:1,000; catalog #610612; BD Biosciences), polyclonal rabbit α-tubulin (1:2,500; catalog #ab18251; Abcam Inc.), and appropriate secondary antibodies (1:5,000 IR-dye-680 Goat anti-Mouse IgG; catalog #NC0252290; Thermo Fisher Scientific for IQGAP1 and 1:5,000 IR-dye-800 Donkey anti-Rabbit IgG; catalog #NC9523609; Thermo Fisher Scientific for α-tubulin; LI-COR Biotechnology). Blots were imaged with a LI-COR Odyssey $F_c$ imaging system (LI-COR Biotechnology). For experiments with transfected cells, NIH-3T3 cells were placed in 1 ml of Optimem (Gibco) and Lipofectamine 3000, 0.5–2.0 µg (noted by experiment) of plasmid DNA, and 10 µl of P3000 reagent. After 2 h, cells were diluted with 1 ml of DMEM supplemented with 20% FBS and 200 mM L-glutamine and then incubated at 37°C for use in experiments. To estimate transfection efficiency by blot (Fig. S4 B), cells were transfected with 0.5 µg plasmids and probed as above.

### Cell fixation and staining

Cells were washed thrice in 1× PBS, washed into 0.3% glutaraldehyde and 0.25% Triton X-100 diluted in 1× PBS, and then fixed in 2% glutaraldehyde for 8 min. Autofluorescence was quenched with fresh 0.1%(wt/vol) sodium borohydride in 1× PBS for 7 min at room temperature. Coverslips were washed twice and blocked with PBST (1× PBS and 0.1% Tween-20) supplemented with 1% BSA for 1 h at room temperature, washed thrice as above, and incubated with primary antibodies (described below) for 1 h at room temperature. Cells were washed thrice with PBST and incubated for 1 h with secondary antibodies (described below) and AlexaFluor-568 phalloidin (1:500; Life Technologies), and/or 1 µM SiR-647-SNAP-dye (New England Biolabs) or 100 nM DAPI (Life Technologies). Coverslips were washed in 1× PBS and mounted in Aquamount (Andwin Scientific).

Coverslips were stained using three different conditions: (1) To assess antibody specificity and SNAP-labeling ligand background (Fig. S4, C and D): 42,000 IQGAP1 knockout cells were plated on round 12-mm coverslips and transfected with mock (reagents but no DNA) or 2 µg SNAP-IQGAP1 plasmid for 48 h, then fixed and probed for IQGAP1 (primary: 1:250, catalog #610612; BD Biosciences; secondary: 1:1,000, AlexaFluor-488 donkey anti-mouse, catalog #A21202; Life Technologies), and stained with 568-phalllodin, SiR-647-SNAP-dye, and DAPI. (2) To evaluate circularity and transfection efficiency of multiple plasmids (Fig. 6, A and B; and Fig. S4, E and F): 42,000 3T3 cells or IQGAP1 knockout cells were plated on round 12-mm coverslips and transfected with mock or 2.0 µg noted IQGAP1 and GFP-actin (Addgene: 31502 [Watanabe and Mitchison, 2002]) plasmids for 24 h. Cells were fixed and probed for tubulin (primary: 1:250, catalog #DM1A; MilliporeSigma; secondary: 1:1,000, Alexa-647 donkey anti-mouse, catalog #A31571; Life Technologies), and stained with 568-phalllodin and DAPI. (3) To

standardize cell shape and evaluate actin and microtubule arrays (Fig. 6, C and D; and Fig. S4, G–J): 100,000 3T3 cells or IQGAP1 knockout cells were plated on CW-M fibronectin coverslips (CYTOO Inc), aspirated after 30 min to remove unattached cells, and transfected with mock or 2 µg noted IQGAP1 plasmids for 24 h. Cells were fixed and probed for tubulin (primary: 1:250; catalog #ab18251; Abcam; secondary: 1:1,000, AlexaFluor-488 donkey anti-rabbit, catalog #A21206; Life Technologies), and stained with 568-phalloidin, SiR-647-SNAP-dye, and DAPI. Only cells expressing either GFP-actin or SNAP-IQGAP1 were used for circularity or morphology analyses.

## Cell imaging and measurements of circularity, morphology, and cytoskeletal architecture

For cells imaged to assess transfection efficiency and SNAP-tag function (Fig. S4, C and D), 33 fields of view (FOVs) sampling the entirety of three independently prepared coverslips per condition were collected as 10-µm thick (0.5 µm step size) Z-stacks using an inverted Nikon Ti2-E spinning disk confocal microscope (SoRa; Nikon Instruments) equipped with 100 mW 405 nm, 100 mW 488 nm, 100 mW 561 nm, and 75 mW 640 nm wavelength lasers, Plan Apochromat 20× 0.75 NA air and Plan Apochromat 60× 1.4 NA oil immersion objectives, a CSU-W1 imaging head (Yokogawa Instruments), a SoRa disk (Nikon Instruments), and a Prime BSI sCMOS camera (Teledyne Photometrics). All cells were collected under the same laser power and exposures, which were 9.8% 405, 50 ms; 21.7% 488, 200 ms; 18.4% 561, 200 ms; 50% 640, 100 ms. The Nikon software denoise module was applied to each Z-stack. The entire Z-stack was projected as a maximum intensity image using Fiji software and all cells ($n$ = 1,872–3,747 cells) were scored for 488-IQGAP1 secondary antibody or 647-SiR-ligand signals (including the necessary controls of knockout cells that do not have IQGAP1 and 647-ligand in cells lacking appropriate SNAP-tag constructs).

To assess plasmid transfection efficiency (Fig. S4, E and F), 20 single Z-plane FOVs were collected from areas spanning the entire area of three independently prepared coverslips per condition. Coverslips were imaged on the Nikon TIRF system described above using the Plan Apo 20× 0.75. NA air objective, DIC-appropriate filters (100 ms white light exposure), and the following laser settings and exposures: 20% 488, 200 ms; 10% 561, 50 ms; and 5% 647, 100 ms. Transfection efficiency was determined as the ratio of cells with GFP-actin signal divided by total cells present visualized by DIC. These values were calculated for each FOV and plotted as the dots shown in Fig. S4 F ($n$ = 60 FOVs, pooled from $n$ = 3 coverslips), with the histogram representing mean values ± SE.

Circularity measurements (Fig. 6 B) were made from the single Z-plane FOVs collected and used for analysis in Fig. S4, E and F. All cells imaged that were not overlapping with other cells or cut off by the edge of the FOV were cropped using a rectangle placed as close to the cell edges as possible, auto threshold (Huang) was set, and then circularity was measured from the "analyze particles" command in Fiji software with output "excluding any holes or edges" selected. Each of these individual cell values was plotted as dots ($n$ = 151–261), with the histogram representing mean values ± SE.

For micropattern-based analyses (Fig. 6, C and D; and Fig. S4, G–J), cells from coverslips were selected (single-blind) and then 10 µm Z-stacks covering the entire cell were imaged in 0.15-µm steps using the inverted Nikon Ti2-E SoRa described above but using a Plan Apo 60× 1.4 NA oil-immersion objective. All images were collected under the same laser power and exposures, which were 9.8% 405, 50 ms; 21.7% 488, 200 ms; 18.4% 561, 200 ms; and 50% 640, 100 ms. The Nikon software denoise module and 40 iterations of Richardson-Lucy deconvolution were applied to each Z-stack. Z-slices equivalent to the bottom 1.5 µm were projected as maximum intensity projections and used for quantitative analyses. For morphology (Fig. S4, G and H), maximum intensity projections of the phalloidin (F-actin) channel were converted to 8-bit grayscale, binarized with a threshold set to maximize the entire cell perimeter, and then the total signal (IntDen) present was recorded using Fiji software. These values were then divided by the total number of available pixels for each image. For actin filament (Fig. 6, C and D) and microtubule architecture (Fig. S4, I and J), the same strategy was applied on the same cells; however, the threshold was set to values that captured available filament signal without saturation. Each of these ratios represents the signal for individual cells ($n$ = 27–35 cells). Ratios per cell are presented as dots in Fig. 6 D and Fig. S4, H and J, and the histogram is the mean ± SD.

For wound-healing assays, 100,000 cells were plated in six-well plates and transfected with 10 µl lipofectamine (2 µg of GFP-actin [mock] and IQGAP1 plasmids, as noted) for 12 h. Cells were replated following treatment with 100 µl 0.25% trypsin for 1 min, diluted with 1 ml of fresh media, and collected via centrifugation at 3,000 × $g$ for 3 min. Cells were gently resuspended in 200 µl of fresh growth media and transferred to each chamber (100 µl per well) of a polymer-coated 2-well µ-dish (IBIDI). After 3 h incubation, the chamber inserts were removed (time zero) and the wound surface was washed thrice with warm DMEM (buffered with 10 mM HEPES [pH 7.4]). Cells were visualized with DIC microscopy using the Nikon TIRF system described above, a Plan Apo 20× 0.75 NA air objective, and three to four stitched together FOVs spanning the wound gap. At least three FOVs were collected along the wound area per condition and then plates were washed in standard media and returned to the incubation chamber between time points. The temperature was maintained during imaging with a heated stage insert (OKO labs). The image area occupied by cells at $T_0$ and $T_{12}$ was determined by converting the DIC image signal to binary and counting the total pixel area equivalent to the cell coverage, divided by the total FOV area. The wound area was normalized by subtracting the $T_0$ coverage from the $T_{12}$ coverage. Thus, the percent closure is the ratio of cell occupancy at 12 h to occupancy at time zero.

## Data analysis, statistics, and presentation

GraphPad Prism 10 (version 10.2.0; GraphPad Software) was used to plot all data and perform statistical tests. The sample size, number of replicates, and statistical tests used for each experiment are in each figure legend. Individual data points (displayed as dots) are shown in each figure; histograms represent means (unless noted otherwise). Bars are standard error

of mean (SE) or standard deviation (SD) as noted in each figure legend. All statistical tests use the significance threshold of $\alpha = 0.05$ to designate significant differences. Student's $t$ test was used for two-group comparisons (Fig. 4 D and Fig. S3 F). All other tests were all-comparison one-way ANOVA performed with Tukey post-hoc analysis. Unless noted, the data distribution was assumed to be normal but not formally tested. Figures were made in Adobe Illustrator 2023 (version 27.4.1; Adobe).

### Online supplemental material

Fig. S1 (related to Fig. 2) provides schematics, SDS-PAGE gels, and example kymographs and quantification of pauses to elongation for untagged IQGAP1 proteins. Fig. S2 (related to Fig. 3) provides SDS-PAGE gels of SNAP-tagged IQGAP1 proteins and provides a comparison of the different elongation rates of each fluorescently labeled actin used in this study. Fig. S3 (related to Fig. 4) contains bulk polymerization assays and the elongation rates of actin filaments from TIRF reactions comparing mDia1(FH1-C) and mDia1($\Delta$DAD) under different conditions. Additional panels detail multicolor TIRF of 488-IQGAP1, 549-mDia1(FH1-C), and 647-actin. Fig. S4 (related to Fig. 6) provides validation of IQGAP1 knockout lines and transfection efficiency of each IQGAP1 construct in NIH-3T3 cells and details the effects of IQGAP1 constructs on cell morphology and microtubule arrays. Video 1 (related to Fig. 1 C) details actin polymerization in the presence of different IQGAP1 concentrations. Video 2 (related to Fig. 1 D) is an example of an IQGAP1-mediated pause to actin filament growth. Video 3 (related to Fig. 2 B) shows TIRF movies of actin filaments polymerizing in the presence of purified IQGAP1 proteins. Video 4 (related to Fig. 3 E) shows a comparison of 488-SNAP-labeled IQGAP1 proteins with polymerizing 647-actin via two-color TIRF microscopy. Video 5 (related to Fig. 4, E and E') is an example of 488-IQGAP1 displacing 549-mDia1($\Delta$DAD) from the plus end (647-actin) obtained via three-color TIRF microscopy. Video 6 (related to Fig. S3, G and G') shows a multicolor TIRF video of 488-IQGAP1, 549-mDia1(FH1-C), 647-actin. Video 7 (related to Fig. 5 D) details the effect of 488-IQGAP1 on 549-mDia1(DAD)-647-CP decision complexes with 405-actin using four-color TIRF microscopy.

### Data availability

Datasets for each figure have been uploaded and deposited, available here https://doi.org/10.5281/zenodo.10895906.

## Acknowledgments

We are grateful to Marc Ridilla for the helpful comments on this manuscript. We also thank J.P.B Pascal for showing us the art of "The Way."

This work was supported by the National Institutes of Health (GM133485) to J.L. Henty-Ridilla.

Author contributions: M.L. Pimm: Conceptualization, Formal analysis, Investigation, Methodology, Validation, Visualization, Writing - original draft, Writing - review & editing, B.K. Haarer: Formal analysis, Investigation, Methodology, Resources, Validation, Writing - review & editing, A.D. Nobles: Formal analysis, Investigation, Writing - review & editing, L.M. Haney: Investigation, Writing - review & editing, A.G. Marcin: Formal analysis, Investigation, Writing - review & editing, M. Alcaide Eligio: Investigation, J.L. Henty-Ridilla: Conceptualization, Data curation, Formal analysis, Funding acquisition, Investigation, Methodology, Project administration, Resources, Supervision, Validation, Visualization, Writing - review & editing.

Disclosures: The authors declare no competing interests exist.

Submitted: 16 May 2023

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

## Supplemental material

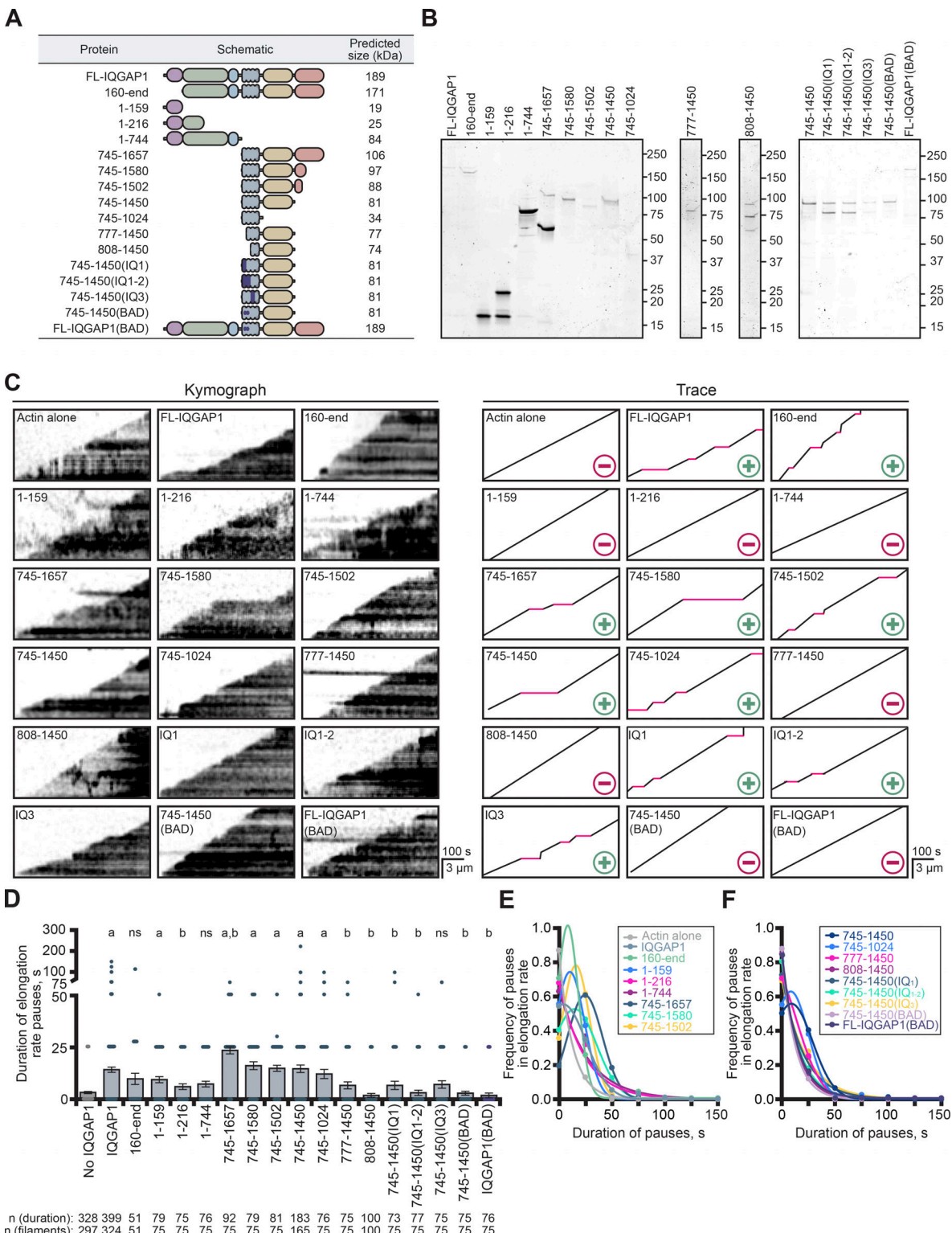

Figure S1. **Protein purity and evaluation of the regions necessary for IQGAP1's plus-end activity. (A)** Schematic of IQGAP1 constructs and predicted molecular weights. **(B)** SDS-PAGE gels of IQGAP1 proteins used in this work. Incomplete cleavage of the SUMO tag results in a ~12 kDa shift. **(C)** Kymographs and traces of elongating actin filaments in the presence of all IQGAP1 proteins listed in A, generated from TIRF images. Red lines in traces indicate pauses in filament elongation. +, IQGAP1 protein pauses filament elongation. −, IQGAP1 protein does not pause filament elongation. Scale for kymographs and traces: length, 3 μm; time, 100 s. **(D)** Duration of individual pauses measured to generate plots in E and F (n [duration] = 75–399 derived from n [filaments] = 51–324 as noted, from at least three technical replicates). Error bars, SE. Statistics, ANOVA: (a) P ≤ 0.05 compared with control (0 nM IQGAP1); (b) P ≤ 0.05 compared with 75 nM FL-IQGAP1 reactions; ns, P ≥ 0.05 to control. **(E and F)** Frequency distribution plots displaying the duration of IQGAP1-mediated pauses in filament elongation for all IQGAP1 conditions tested. Values for the following conditions are repeated from Fig. 2 F for comparative purposes: No IQGAP1(actin alone), IQGAP1, and FL-IQGAP1(BAD). Source data are available for this figure: SourceData FS1.

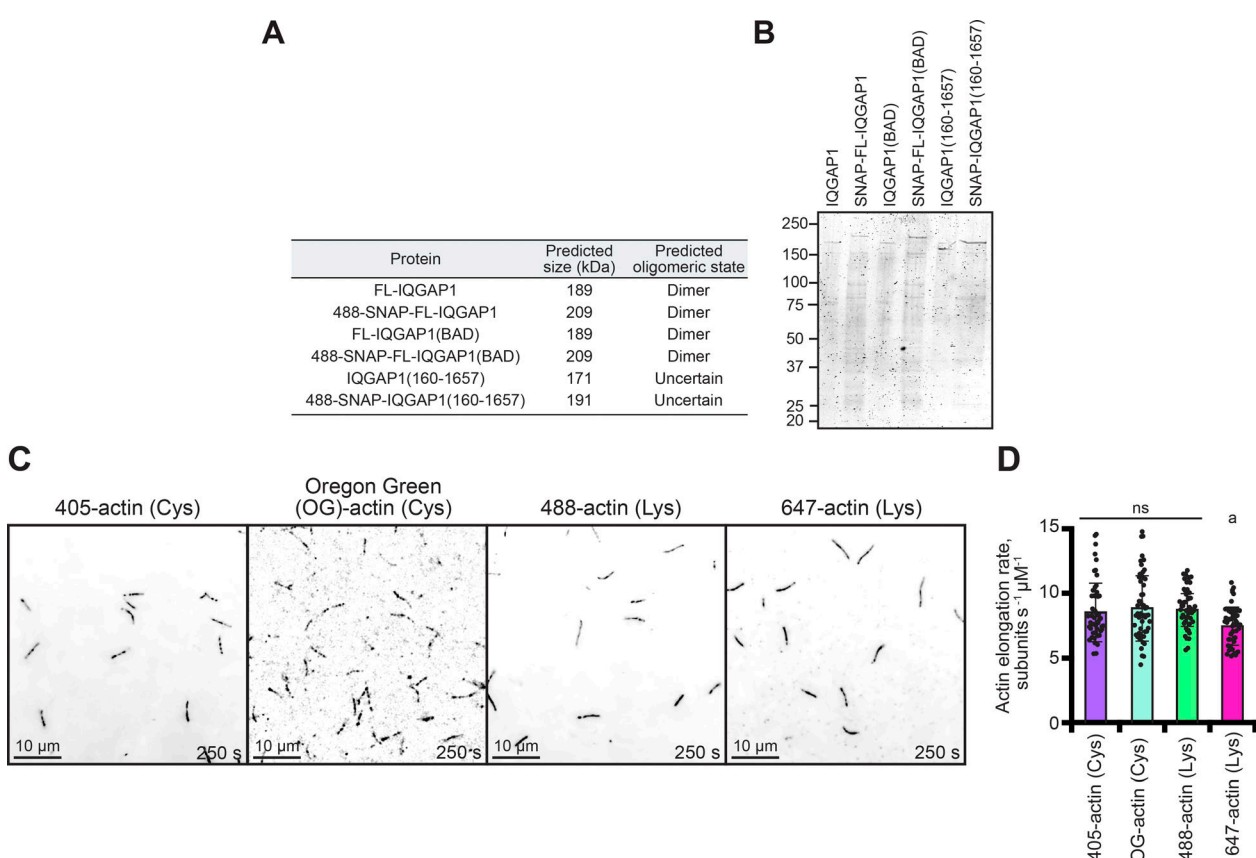

Figure S2.   **Purity of SNAP-tagged IQGAP1 proteins and comparison of the labeled actins used in TIRF assays. (A)** Table of predicted molecular weight and oligomeric state of untagged and SNAP-tagged IQGAP1 proteins. **(B)** SDS-PAGE gels of untagged and SNAP-tagged IQGAP1 proteins in A. **(C)** Images from TIRF reactions comparing the actin polymerization of different fluorescent labeling strategies. Reactions contain 1 μM actin labeled with either 10% Dylight 405, 10% Oregon Green (OG), 10% Alexa 488, or 10% Alexa 647 dyes. Dylight 405- and OG-actin were labeled on cysteine (Cys) 374, whereas 488- and 647-actin were labeled on accessible lysine (Lys) residues of polymerized actin. Scale, 10 μm. **(D)** Mean actin filament elongation rates from reactions in C ($n$ = 17 filaments per reaction, 51 total per condition [dots]; from three different trials). Error bars, SD. Statistics, ANOVA: (a) P ≤ 0.05 compared with all other conditions; ns, P ≥ 0.05 compared with each condition under the line. Source data are available for this figure: SourceData FS2.

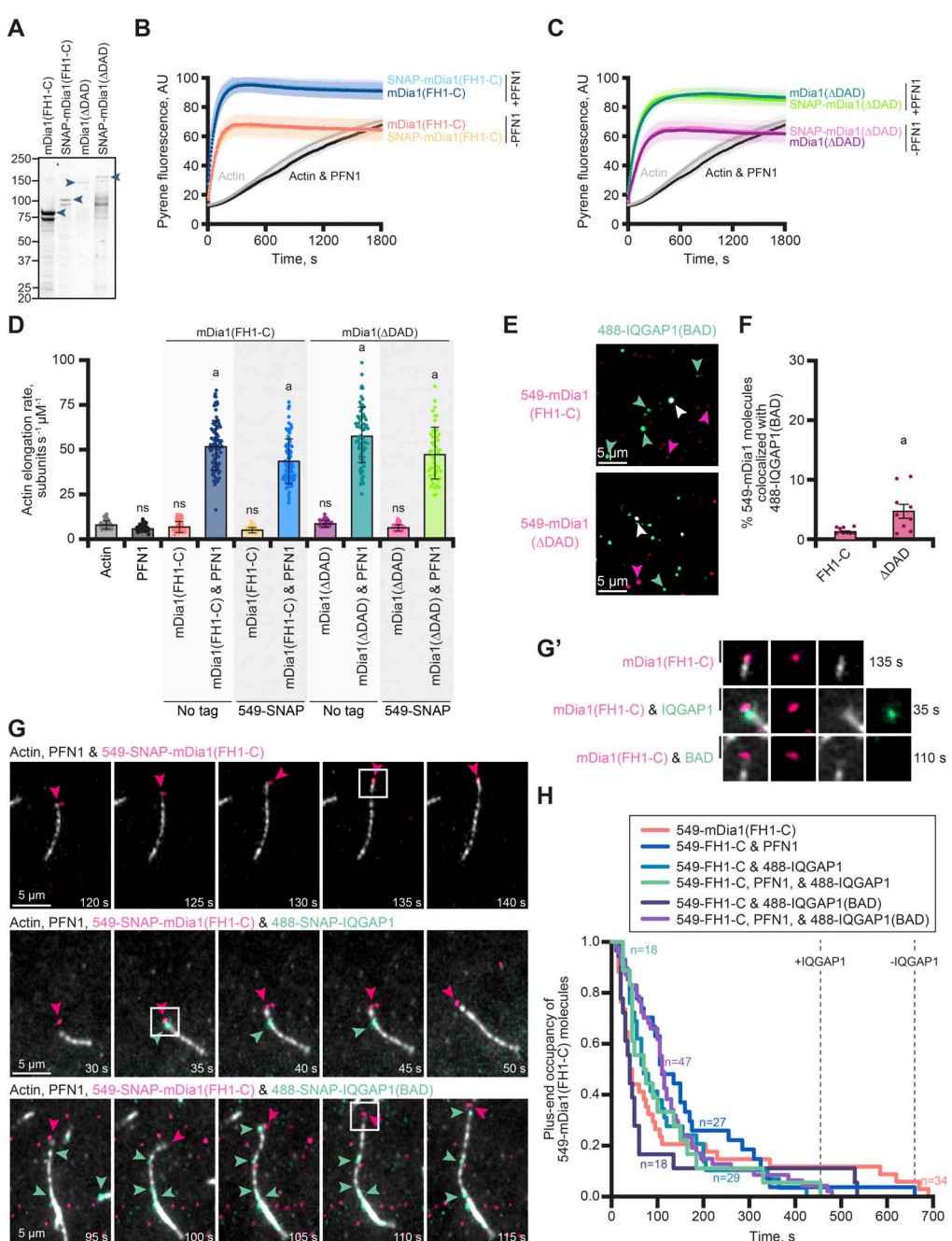

**Figure S3. Comparison of the actin assembly and IQGAP1-related activities of mDia1(FH1-C) and mDia1(ΔDAD). (A)** SDS-PAGE gel of untagged and SNAP-tagged formin proteins. **(B and C)** Bulk actin filament assembly assays comparing tagged and untagged formin constructs. Reactions contain 2 µM actin (5% pyrene-labeled), 5 µM profilin-1 (PFN1), 5 nM mDia1(FH1-C or ΔDAD), as indicated. Shaded values are SD from n = 3 assays. Curves in B and C are directly comparable (comparisons between FH1-C and ΔDAD and controls were performed on the same plates). Thus, the actin alone and actin and profilin control values are the same in each panel. **(D)** Mean actin filament elongation rates from TIRF reactions containing 1 µM actin monomers (10% Alexa 488 labeled) polymerizing alone or in the presence of 5 µM PFN1, 1 nM mDia1(FH1-C or ΔDAD), and 1 nM 549-SNAP-mDia1(FH1-C or ΔDAD) as indicated. Dots represent elongation rates of individual actin filaments (n = 17 filaments per reaction from three independent reactions; 51 total filaments per condition). Error bars, SE. Statistics, ANOVA: (a) P ≤ 0.05 compared with actin alone; ns, P ≥ 0.05 compared with actin alone. There was no significant difference in mean elongation when comparing untagged and SNAP-tagged proteins. **(E)** Images from single-molecule TIRF of 1 nM 549-mDia1 constructs with 1 nM 488-IQGAP1(BAD). Arrows highlight examples of individual molecules of SNAP-IQGAP1(BAD) (green) or SNAP-mDia1 (pink) and complexes of both proteins (white). Scale, 5 µm. **(F)** Quantification of colocalized mDia1-IQGAP1 molecules from reactions in E. Error bars, SE. Dots are percentages calculated for individual FOVs (n = 9 FOVs total, from three replicates). Statistics, Student's t test (two-tailed): (a) P ≤ 0.05 compared with mDia1(FH1-C). **(G)** Time-lapse montages from three-color TIRF reactions containing 1 µM actin (10% Alexa 647 label; gray) polymerizing in the presence of 5 µM PFN1, 1 nM 549-SNAP-mDia1(FH1-C) (pink), and 1 nM 488-SNAP-IQGAP1 or 488-SNAP-IQGAP1(BAD) (green). Scale, 5 µm. Arrows indicate mDia1 (pink) and IQGAP1 (green) localization. **(G')** Insets from G show a zoomed in view of single molecules present on or near filament ends. Scale, 1.5 µm. **(H)** Survival plots of the plus-end occupancy of 549-SNAP-mDia1(FH1-C) molecules in the presence and absence of 488-labeled IQGAP1 proteins from reactions in G. Reactions that contain IQGAP1 are shorter lived than those without IQGAP1 as marked by dotted lines (n = 18–47 molecules per condition [as stated], from n = 3 independent reactions).

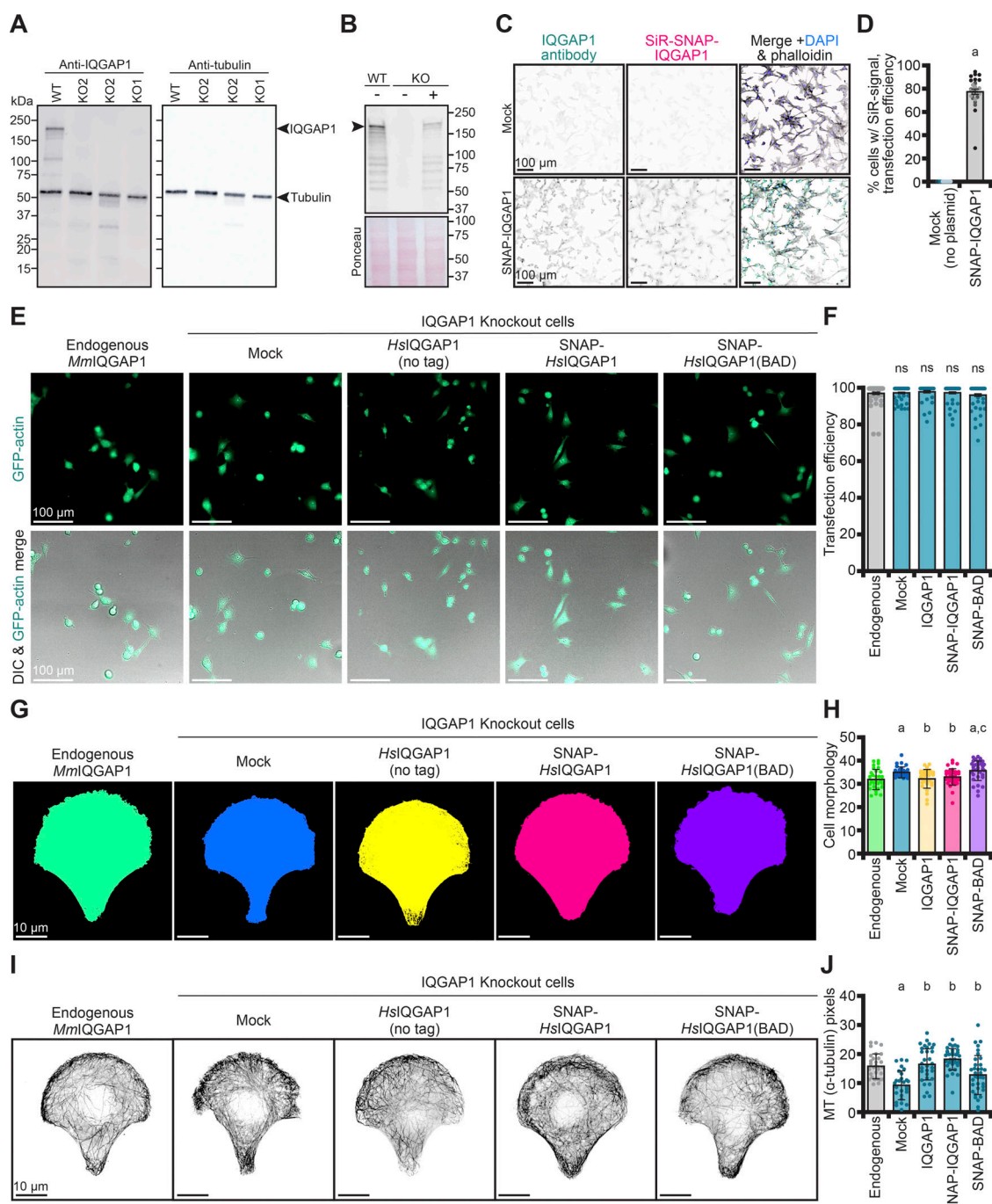

Figure S4. **Characterization of IQGAP1-expressing NIH-3T3 cells. (A)** Blot confirming two clonal IQGAP1 knockout cell lines. Blot was probed for IQGAP1 (1:1,000; BD610612) and α-tubulin (1:2,500; ab18251) as a loading control, with appropriate secondary antibodies (1:5000 IR-dye-680 for IQGAP1 and IR-dye-800 for α-tubulin). **(B)** Blot of NIH-3T3 cell extracts indicating expression of IQGAP1 plasmids (primary: 1:1,000, BD610612; secondary: 1:5,000, IR-dye-680). Ponceau stained membrane used as loading control. −, (mock) no vector. +, cells transfected with IQGAP1 plasmid. **(C)** TIRF images of IQGAP1 knockout cells probed for IQGAP1 (primary: 1:250, BD610612; secondary: 1:1,000, 488- Alexa Fluor 488 donkey anti-mouse) and SiR-SNAP labeling transfected with mock or SNAP-IQGAP1 plasmid. **(D)** Quantification of transfection efficiency from images in C (n = 11 fields of view [FOVs] per condition from three different coverslips). Shaded dots were grouped by each coverslip. Scale, 100 μm. Error bars, SE. Statistics, Student's t test: (a) P ≤ 0.05 compared with mock. **(E)** Images showing GFP and DIC signals to determine the transfection efficiency of GFP-actin plasmids by cells also transfected with plasmids harboring each IQGAP1 (untagged, SNAP-tagged, or SNAP-IQGAP1(BAD)), or with no second plasmid (mock and Endogenous [WT]), as indicated. **(F)** Quantification of transfection efficiency from images in E (n = 60 FOVs [dots] analyzed per condition, pooled from three coverslips. Scale, 100 μm. Error bars, SE. Statistics, ANOVA: ns, P ≥ 0.05 comparing all conditions. **(G and H)** Morphology on micropatterns of NIH-3T3 cells expressing the various IQGAP1 plasmids in E and H associated quantification. Scale, 10 μm. Dots represent measurements from individual cells (n = 27–35 cells per condition). Error bars, SD. Statistics, ANOVA: (a) P ≤ 0.05 compared with endogenous MmIQGAP1; (b) P ≤ 0.05 compared with mock (IQGAP1 knockout cells); (c) P ≤ 0.05 compared with IQGAP1 knockout cells expressing SNAP-IQGAP1 on a plasmid; ns, P ≥ 0.05 comparing treatments under the line. **(I and J)** Maximum intensity images of microtubules from cells in G and J associated quantification (total α-tubulin signal). Statistics, ANOVA: comparisons as in H. Source data are available for this figure: SourceData FS4.

Video 1.   **(Associated with Fig. 1 C). Polymerizing actin filaments in the absence or presence of IQGAP1.** Reactions contain: 1 µM actin monomers (20% Oregon Green [OG] labeled) and noted concentrations of IQGAP1. Playback, 10 frames per second (FPS). Scale, 20 µm.

Video 2.   **(Associated with Fig. 1 D). Example of IQGAP1-mediated pause to actin filament elongation.** Reactions contain 1 µM actin monomers (20% OG labeled) with or without 75 nM IQGAP1. Arrows depict active filament growth (green) or pauses to elongation (pink). Playback, 10 FPS. Scale, 2 µm.

Video 3.   **(Associated with Fig. 2 B). Actin assembly in the presence of different IQGAP1 proteins.** Reactions contain: 1 µM actin monomers (20% OG labeled) and 0 nM (actin alone control) or 75 nM of noted IQGAP1 proteins. Playback, 10 FPS. Scale, 10 µm.

Video 4.   **(Associated with Fig. 3 E). SNAP-IQGAP1(BAD) binds the sides of actin filaments and does not interfere with filament elongation.** Reactions contain 1 µM actin monomers (10% Alexa 647 label) and 75 nM 488-SNAP-IQGAP1, 488-SNAP-IQGAP1(BAD), or 488-SNAP-IQGAP1(160-end). Panels show individual wavelengths and merge. Arrows indicate the absence (purple) or presence of IQGAP1 proteins (green) on filament ends or sides. Playback, 10 FPS. Scale, 2 µm.

Video 5.   **(Associated with Fig. 4, E and E'). IQGAP1 binds to and displaces formin (mDia1) at filament plus ends.** Reactions contain: 1 µM actin monomers (10% Alexa 647 label), 5 µM PFN1, 1 nM 549-SNAP-mDia1(ΔDAD), and 1 nM 488-SNAP-IQGAP1 or 488-SNAP-IQGAP1(BAD). Panels show individual wavelengths and merge. Arrows indicate free plus-ends (gray), individual molecules of IQGAP1 (green) or formin (pink), or IQGAP1–formin complexes (white). Playback, 10 FPS. Scale, 2 µm.

Video 6.   **(Associated with Fig. S3, G and G'). IQGAP1-mediated formin displacement requires IQGAP1-mDia1 interaction.** Reactions contain 1 µM actin monomers (10% Alexa 647 label), 5 µM PFN1, 1 nM 549-SNAP-mDia1(FH1-C), and 1 nM 488-SNAP-IQGAP1 or 488-SNAP-IQGAP1(BAD). Panels show individual wavelengths and merge. Arrows indicate individual molecules of IQGAP1 (green) or formin (pink), or IQGAP1–formin complexes (white). Playback, 10 FPS. Scale, 2 µm.

Video 7.   **(Associated with Fig. 5 D). IQGAP1 promotes the dynamic turnover of actin filament end-binding proteins.** Reactions contain: actin filament seeds (10% Dylight 405 label stabilized with 132 nM Alexa 405 phalloidin), 10 nM 549-SNAP-mDia1(ΔDAD), 10 nM 649-SNAP-Capping Protein (CP) with or without 488-SNAP-IQGAP1. Arrows highlight free actin filament plus ends (purple) and the presence of molecules including IQGAP1 (green), mDia1(ΔDAD) (pink), or CP (yellow), formin-CP decision complexes (orange), or DC with IQGAP1 (turquoise). DC, formin-CP decision complexes. Playback, 10 FPS. Scale, 2 µm.

