## [Peer Review File · The Journal of Cell Biology]

Coordination of actin plus-end dynamics by IQGAP1, formin, and capping protein

Morgan Pimm, Brian Haarer, Alexander Nobles, Laura Haney, Alexandra Marcin, Marcela Alcaide Eligio, and Jessica Henty-Ridilla

Corresponding Author(s): Jessica Henty-Ridilla, SUNY Upstate Medical University

Review Timeline:

Submission Date:	2023-05-16
Editorial Decision:	2023-06-27
Revision Received:	2024-04-01
Editorial Decision:	2024-04-30
Revision Received:	2024-05-06

Monitoring Editor: Pekka Lappalainen

Scientific Editor: Tim Fessenden

Transaction Report:

DOI: <https://doi.org/10.1083/jcb.202305065>

June 27, 2023

Re: JCB manuscript #202305065

Dr. Jessica L. Henty-Ridilla
SUNY Upstate Medical University
Biochemistry and Molecular Biology
750 East Adams Street
4271 Weskotten Hall Addition
Syracuse, NY 13210

Dear Dr. Henty-Ridilla,

Thank you for submitting your manuscript entitled "Coordination of actin plus-end dynamics by IQGAP1, formin, and capping protein". The manuscript has been evaluated by expert reviewers, whose reports are appended below. Unfortunately, after an assessment of the reviewer feedback, our editorial decision is against publication in JCB at this time.

You will see that, while all reviewers were intrigued by the proposal that IQGAP1 can promote the turnover of mDia1 and capping protein from actin filament barbed ends, all raised concerns that experimental methods lacked essential details. These concerns were compounded by data presentation and figure organization that reviewers felt needed improvement. In addition, all reviewers suggested additional experiments and data analyses that would be necessary to confirm the main conclusions of this study. Reviewers 1 and 2 made important suggestions to this end, and especially reviewer 3 raised several important points that would need to be resolved for example on the variable elongation rates in Figures 1-3, multiple concerns related to Figure 4, and quantification of stress fibers in Figure 7. Additionally, we agree with Reviewer 2 that the use of orthogonal methods to support claims made with TIRF imaging would be helpful towards resolving some of the key concerns.

We feel that the requests made by the reviewers are more substantial than can be addressed in a typical revision period. If you wish to expedite publication of the current data, it may be best to pursue publication at another journal. However, given interest in the topic and the JCB's interest in publishing this work, we would be open to resubmission to JCB of a significantly revised manuscript that fully addresses the reviewers' concerns and is subject to further peer-review. Should you wish to pursue publication with a revised manuscript, please provide a plan for revision with point-by-point responses to reviewer concerns. If and when you would like to resubmit this work to JCB, please contact the journal office to discuss an appeal of this decision or you may submit an appeal directly through our manuscript submission system.

Regardless of how you choose to proceed, we hope that the comments below will prove constructive as your work progresses. We would be happy to discuss the reviewer comments further once you've had a chance to consider the points raised in this letter. You can contact the journal office with any questions, cellbio@rockefeller.edu.

Thank you for thinking of JCB as an appropriate place to publish your work.

Sincerely,

Pekka Lappalainen
Monitoring Editor
Journal of Cell Biology

Tim Fessenden
Scientific Editor
Journal of Cell Biology

Reviewer #1 (Comments to the Authors (Required)):

IQGAP1 is a protein that has been previously shown to have several actin-associating activities including F-actin side-binding, capping, bundling, and interacting with formins. However, it is unclear which regions of IQGAP1 are involved in its various activities, as well as how it interacts with other barbed-end associated proteins. In this study, the authors use multicolor TIRF microscopy to characterize IQGAP1's capping activity and isolate its activity to two cysteine residues in the IQ region of the protein. They examine the interactions of IQGAP1, mDia1, and capping protein at the barbed ends of actin filaments. Finally, the

authors also create IQGAP1 CRISPR KO and both wildtype and mutant rescue cell lines to determine the role of IQGAP1 in cell morphology, actin organization, and wound healing.

Overall, this study represents a substantial amount of work and specifically advances our understanding of IQGAP1's capping functions both alone and in conjunction with other proteins at the barbed ends of actin filaments. In general, I think there is a lot of good data in this paper. However, there is also considerable overinterpretation of claims and lack of detail in regards to description of the experiments performed, their quantification, and the underlying rationale. I don't want to demand a bunch more experiments be performed as I think the data generated is sufficient. However, I think a substantial and critical reworking of the manuscript is necessary before its publication.

Major comments:

Figure 1B and Supplemental Figure 1C appear to be very similar. I am assuming that the difference is that figure 1 is His-tagged protein and supp figure 1 is GST-tagged protein? If so, this should be explicitly stated in the figure legend of figure 1 (and perhaps more clearly in supp figure 1 too, I only see it in the title). However, it is striking that the capping activity is much easier to see in the supp figure 1 examples than in figure 1 (potentially partially due to differences in protein concentration?), and the most dramatic effect to my eye in figure 1B is the bundling activity. I would suggest potentially moving Figure 1B to the supplement and instead including several kymographs of growing actin filaments in control vs +IQGAP1 conditions, to more clearly show the elongation stalling, as that is the data that sets up the story. As it stands, Figure 1E isn't super convincing either, and I think kymographs might be an easier way of showing many timepoints to more easily see the elongation effects of IQGAP1.

Figure 2A represents a substantial amount of work that I think deserves to be acknowledged more thoroughly in the text. The results section should state the assays and quantification performed for each construct, just something like "we purified # truncations of IQGAP1, performed TIRF microscopy experiments of actin in the presence of each unlabeled IQGAP1 truncation, and quantified XXX for each truncation." The schematic in Figure 2A itself also needs further clarification. What is represented by the dark shaded purple portions in 745-1450(IQ1), 745-1450(IQ1-2), and 745-1450(IQ3)? In the methods, it seems as though these are different mutation sets, but this should be stated clearly in both the figure legend and the results. In general, the rationale behind these point mutations needs clarification. Why were these tested? Were other mutations tested? What is the general structure of this domain and how are these residues postulated to affect capping abilities?

As shown, I'm not fully convinced that there isn't an issue with dimer formation in the CD mutant. In the methods, the authors state that oligomeric state was determined by photobleaching of complexes adsorbed to the coverglass surface. It's not uncommon to have protein sticking to the coverglass surface during these types of experiments, but it's also not super uncommon to have protein aggregates also adsorbed to the coverglass surface, and as such I do not think this is an appropriate method to determine oligomeric state. I would be more confident if the photobleaching experiments were performed on IQGAP1 proteins associated with the barbed ends of actin filaments, though I realize those experiments might be difficult. Additionally, in the graph in figure 3C, why are different fluorophores used for the WT IQGAP1 and CD mutant? And what is the reason for the differences in the predicted values for a monomer, dimer, trimer in the WT vs CD mutant condition? If this is a difference in protein labeling, the labeling percentages should be clearly stated as well as the math performed to generate these values. In general, much more information is needed in the methods and/or results section to clarify how these experiments were performed, how the predicted values were generated, and how the predicted vs experimental values were compared. To address these concerns, either another method can be used to determine dimerization efficiency, or (2) more details on the methods/results can be provided as shown above, these results can be moved to the supplement, and the claims can be substantially softened to state that it is still possible that dimerization efficiency may play a role in ability to cap actin filament barbed ends.

Similar to the point above, I do not think the experiments performed in Figure 4B are appropriate to determine mDia1-IQGAP1 interactions. I understand the point, which is to determine whether mDia1 and IQGAP1 interact before associating with the actin filament barbed end, but again I find it troublesome to quantify interactions based on colocalization of proteins adsorbed to glass (I'm assuming this is the case as these details are not mentioned in the methods), as aggregates could easily stick to the glass, as is demonstrated by the fact that the mDia1-IQGAP proteins that are theoretically unable to interact still demonstrate a good amount of colocalization via this method. I suggest removing these results completely. Instead, supplemental figure 3B and C could be moved to the main figure. Again, these experiments should also be more clearly described in the results - how are preformed complexes assembled? When are they added? How is the quantification performed? Even if it's done manually, stating "we examined each actin filament barbed end for addition of IQGAP or mDia1. Upon binding, we quantified whether the other protein was also present" would provide clarity. It's hard for me to parse which details are included in the methods, as at times it is hard to tell which part of the methods refer to which experiments, so the methods should be subdivided into the different experiments and details added for clarity.

Figure 5 needs to be explained more thoroughly in the text. The results are jumped right into, but the set up of the experiment should be described first as well as what is defined as a 'blocked end'. In general, I would actually prefer if the experiments were described in terms of elongation. Describing 'blocked ends' seems to me to be jumping into interpretation, whereas elongation is really what is being assessed. In any case, if the 'blocked ends' term is kept, it should be thoroughly described.

In Figure 6A, the single channel images should be shown, as it is very difficult to tell when two vs three components are present on the barbed end. I think an example first image in conjunction with a kymograph may be the best way to go here, as I suggested for Figure 1. I also find Figure 6B hard to follow, I'm not sure at all what is being shown here. In general, I think this figure should be rethought in terms of how to display the data, as these results are potentially very cool, but hard to parse as they are currently shown, and difficult for me to fully evaluate.

Much more detail is needed to explain Figure 7. The results section needs to make clear that these cells are on a micropatterned surface. To my eye, the shape of the cells don't differ that extensively. In what way does the shape change, and what is actually being quantified? Also, I like the color scheme, but all panels of Figure 7C should be shown in grayscale for easier viewing of the actin organization. What is being quantified in figure 7D? The methods state how images were processed, but not what is being quantified. If it's truly just the actin signal, I don't find that to be a quantification as that could just vary based on cell size or numerous other variables, and that doesn't demonstrate that IQGAP1 results in a loss of 'tension-dependent stress fibers', as that specific cytoskeletal network is not examined at all, but rather just the actin in general. Again, much more detail is needed here as it is unclear what these experiments are showing. The wound healing assay is cool, so maybe everything else should be moved to the supplement? Where do the rates come from in Figure 7G? With the data as it is currently in this paper, I would not be comfortable putting rate constants on this diagram.

As mentioned throughout my comments, in general, I think the whole paper could use more detail. The methods section is very sparse - much more detail is needed on how each type of experiment and quantification is performed. It would be easiest to subdivide the methods by experiment and quantification type and make sure adequate detail is provided for each experiment and quantification performed. In addition, I found that the results section often glossed over descriptions of the experiments, and more detail would be useful to the reader. It's totally OK to repeat information in the results, figure legends, and methods, I would rather have all of the information super explicitly stated.

Minor comments:

In general, I find all of the distribution of frequency events figures (Figure 1G, 2F, 6C, Supp Fig 1G, 2C, 2E, 2F) hard to follow, and it's not clear to me why points are only included every 25 seconds (at most). Is this to simplify the plot? It's hard to assess the fit when so few points are shown. I can understand why it's difficult to show all of the points, but if that's the case, the data should be shown in a different way. If this type of diagram is kept, the R² values of the fits should always be listed (they are in Figure 6C, but not in the others).

In this manuscript the terms plus and minus end are used for the ends of actin filaments. However, in figure 4E I found a wayward 'barbed end'. I'm more used to seeing barbed and pointed end as terms for actin, but I'm OK with plus and minus end. However, maybe it would be good to mention both in the intro to appease actin terminology sticklers?

In Figure 2A, in the label for "745-1450(IQ1-2), the I is a 1.

Is IQGAP1 diffusing/sliding on actin filaments ever observed as suggested in the discussion?

Starting with Figure 3, the text states that IQGAP is 649 labeled, while the figure says 549 labeled. I would check all of these numbers in-text, in figure legends, and in the figures themselves. Also a rationale as to when different fluors are used in situations where there is a direct comparison should be added.

Did you ever test whether IQGAP1 is capable of releasing autoinhibition of mDia1? One might hypothesize that based on its binding site within the DID domain. It would be worth mentioning if this experiment was attempted and any potential results (even if negative).

There are multiple sentences in the manuscript that need to be clarified. I've listed several here:

Line 47: "Filament polymerization is further refined by complexes with formin to enhance (e.g., APC, CLIP-170, or spire) or with CP to limit (e.g., twinfilin or CARMIL) assembly¹⁰⁻¹⁹". Is the point that these proteins form complexes with formin or CP? I think this sentence should be reorganized because I was unclear on what it meant.

Line 92: "We used truncation and mutational analyses with 75 nM protein to further delimit the region and residues important for plus end capping because it is sufficient for plus-end binding (kD = 25-35 nM) and filament pauses were obvious with the full-length protein." Should be "because that protein concentration is sufficient for wild-type IQGAP1 plus-end binding".

Line 149: "Notably, stochastic pauses in formin-mediated actin filament growth occurred only when IQGAP1 could directly bind formin i.e., 549-mDia1(Δ DAD), and were rarely observed for reactions performed with 549-mDia1(FH1-C) or IQGAP1(CD) (Figure 4D-G)." The way this sentence is organized, it appears to suggest that IQGAP1(CD) is unable to bind mDia1, but I don't think that is correct right? Should be clarified.

Line 174: "Remarkably, the addition of IQGAP1 to either condition resulted in significantly fewer blocked ends (P {less than or equal to} 0.0001), although significantly fewer free-ends compared to buffer controls (P = 0.897) (Figures 5F-G and S4C; Movie 8)." P=0.897 is significant? I may be confused with how this sentence is constructed or what is being quantified/compared here, or this may not be the correct p value? Please clarify.

Line 231: "formin mutants to show that both IQGAP1 and mDia1 can co-exist on filament plus-ends without binding each other or affecting formin-based activities." Was this shown? To me this suggests an experiment like Figure 4D but with the mDia1(FH1-C), but I don't see this experiment.

Reviewer #2 (Comments to the Authors (Required)):

The manuscript entitled "Coordination of actin plus-end dynamics by IQGAP1, formin and capping protein" by Morgan L. Pimm et al. investigates the role of IQGAP1 in regulating actin dynamics at the barbed ends of filaments. The authors propose that IQGAP1 promotes the turnover of end-binding proteins by reducing their residence time.

The manuscript contains a substantial amount of data and sheds new light on the interaction between IQGAP1, formin and capping protein at the barbed ends of actin filaments. However, I have some concerns about the following points:

First, Figure 6, which is crucial to support the main message of IQGAP1 reducing residence times, needs significant improvement. The figure is currently difficult to comprehend, and it should be revised to ensure clear and understandable representation of the data. In particular, Figure 6C, which displays the lifetime of various complexes, appears excessively small and complex, making it hard to read. The superimposed fits obscure the actual experimental data points, and the low R² values for CP and mDia1-CP raise doubts about the validity of the exponential fit for these data.

Additionally, Figure 6B, illustrating fluorescence intensity profiles, is perplexing, and the method employed to obtain these profiles is unclear. It would be beneficial if the authors considered utilizing alternative analyses, such as kymographs or other appropriate methods, to illustrate the duration of protein presence at filament ends in the presence or absence of IQGAP1.

Second, there is a lack of information regarding the number of times the experiments were repeated. This omission raises concerns about reproducibility and statistical significance. It is essential for the authors to clarify whether the experiments were performed with independent replicates to ensure the reliability of their findings.

Third, although the characterizations in the manuscript rely entirely on fluorescence microscopy (TIRF), it is questionable whether this method alone is sufficient to carry out all the quantifications and draw the conclusions presented. It would have been worthwhile to include bulk experiments using pyrene fluorescence or other appropriate techniques to strengthen the main conclusions of the study.

Finally, some figures are probably not essential as main figures and could be moved as supplementary figures, e.g. the last figure on cell biology contains very little information.

Minor points

1. Supplementary figure 1: why is the dose-dependent response of IQGAP1 a problem? And why choose purified IQGAP1 with a His-tag that does not show a dose-dependent response (because in Hoeprich et al. MBoC 2022, they seem to have this dose-dependent response).
2. Why doesn't 649-IQGAP1 have the same activity as IQGAP1? Have the authors tried to assess the rate of actin elongation in the presence of 649-IQGAP1 without exposing the filaments to the far-red channel? Why aren't the same fluorophores used to compare IQGAP1 and the CD mutant?
4. In general, the plus end is used for microtubules and the barbed end for actin filaments, so the title could be modified accordingly.

Reviewer #3 (Comments to the Authors (Required)):

IQGAP1 is a dimeric, multidomain, scaffolding protein that facilitates the formation of complexes with various ligands to regulate diverse cellular processes such as cell migration, cytoskeletal dynamics, intracellular signaling, and intercellular interactions. In

this work, Pimm and colleagues used single- and multi-color TIRF imaging with IQGAP1 variants in the absence or presence of mDia1 constructs and heterodimeric capping protein (CP) to (i) visualize and characterize the association of IQGAP1 with actin filaments and its consequence on actin assembly and to (ii) study its effect on barbed end decision complexes that either promote or pause filament growth. They found that IQGAP1 had no effect on nucleation, but reduced filament growth at the barbed end by approximately 30%, suggesting transient capping, consistent with previous work (Pelikan-Conchaudron et al., JBC, 2011). Subsequently, they identified two residues in IQGAP1 critical for its association with filament barbed ends. Interestingly, using TIRF imaging, they were also able to visualize dimers of IQGAP1 and mDia1 or CP on filament barbed ends alone and as a multi-component complex bound to filament ends. Notably, IQGAP1 appears to promote the turnover of end-binding proteins, reducing the residence time of CP, mDia1 or mDia1-CP decision complexes by approximately one order of magnitude. All this is potentially quite interesting, but I see a number of issues with the constructs and the TIRF assays (see below). The physiological relevance of these interactions suffers from conceptual design, is not clear from the data presented, and needs to be explored in much more detail. In my view, this work is currently not suitable for publication in JCB.

Specific points:

Unlike full-length IQGAP1, to my knowledge, functional full-length mDia1 has not been purified from *E. coli*. For this reason, Bruce Goode's lab has expressed and purified a functional FL-mDia1 in yeast (Maiti et al., Cytoskeleton, 2012). The Rosen lab also found that solubility and yield of the complete full-length mDia1 was very poor in *E. coli* and therefore turned to expression of mDia1 in Sf9 insect cells (Otomo et al., PlosOne, 2010). It is therefore instrumental that the activity of the new probes, in particular of SNAP-tagged mDia1 Δ DAD, which is even slightly larger compared to FL-mDia1, is first shown to be functional in the TIRF assay. Since mDia1 lacking its C-terminal DAD domain is not expected to be autoinhibited, fluorescent SNAP-tagged mDia1 Δ DAD must be shown to stay associated (surf) with the elongating filament barbed ends and accelerate filament growth in the presence of profilin (PFN) roughly 5-fold, as previously shown for shorter mDia1 constructs containing the FH1-FH2-C region (e.g. Bombardier et al., Ncomms, 2015).

Single- and especially multi-color TIRF imaging is certainly quite challenging, but the quality of the images and movies shown here is not state of the art due to the very high background resulting in a poor signal-to-noise ratio. Unfortunately, the TIRF method section is not described in sufficient detail and lacks information about preparation of coverslips, used laser power, exposure times, filters, technical setup for multicolor imaging etc. As it stands, this study will refer to another study, which will then refer to yet another study. Since TIRF imaging is a central element of this study, all of the necessary details should be described in the Methods section. Moreover, the authors used streptavidin-coated chambers to tether growing filaments (spiked with biotin) to the surface. Depending on the coating conditions, this can lead to physical stalling of the filaments due to altered stress or torsion or even road blocks by proteinaceous deposits on the surface, which makes interpretations of data difficult. An example is Figs. 1E, F and Movie 3, in which IQGAP1 is assumed to cause transient capping in 70% of the filaments. However, in this single-color TIRF experiments using only labelled actin, it is unclear whether reduced growth is in fact caused by capping of filament barbed ends by IQGAP1. These experiments and corresponding analysis of data shown in Figs. 1F and G need to be repeated with fluorescent 649-IQGAP1 to visualize capping of the protein on growing filament barbed ends and correlate it with filament growth. In Movie 6, such an experiment is shown on the left, but frankly, I was not able to detect an obvious pause in filament growth upon IQGAP1 binding to the barbed end. Out of curiosity, I converted the shown mp4 movie 6 into a TIF stack and analyzed rates of filament growth with Fiji. With 6.2 subunits/sec, the filament elongated rather slowly, but more or less constantly over the entire sequence. Moreover, when looking at specific rates from frame to frame, there does not appear to be a direct correlation between binding of IQGAP1 to the barbed end (350 s) and reduced growth. Therefore, in my opinion, the data presented does not support the conclusions drawn.

For this reason, fluorescent IQGAP1 also needs to be used in experiments related to Fig. 2 to clearly assess its potential effects on capping and barbed end growth.

This brings me to the next issue. Why are the elongation rates of spontaneously growing actin control filaments so different in Fig.1 (10.2 subunits/sec) vs. Fig. 2 (7.6 subunits/sec) vs. Fig. 3 (with about 5 subunits/sec). These rates are approximately 25% to 50% lower than published values (for example 10.9 subunits/sec using Oregon-green labelled actin as reported by Kovar et al., 2006 in Cell). So how solid is the statistical data when not even the controls can be accurately reproduced? Is this possibly due to different protein concentrations used? Oddly enough, the authors determined their protein concentrations by densitometry of Coomassie stained gels compared to a BSA standard curve and not by spectroscopy as stated in the method section. Why using such an inaccurate method that is not acceptable, because Coomassie staining is dependent on the size of the proteins as well as on their content of aromatic and basic residues. This measurement is typically only done with impure proteins to get a rough estimate of protein concentration. To exclude this, authors are asked to show a Coomassie-stained gel with all purified proteins in a supplementary figure.

Fig. 3: As mentioned above, better movies (compared to Movie 6) are needed to support the claims of Fig. 3F. Here, the authors show three examples where IQGAP1 apparently enters the filament barbed to cause a pause (red lines), followed by dissociation of the protein from filament ends to resume filament growth. These events should be clearly visible in the movie. The authors should also test an IQGAP1 mutant lacking its CHD F-actin binding sites to assess whether this mutant protein is

still capable of interacting with filament barbed ends in the absence of its side binding activities. This would allow them to distinguish between binding directly to the barbed end or near the barbed end.

In experiments related to Fig. 4 the authors state "Single-molecule assays that contained 549- mDia1(Δ DAD) showed significantly more colocalization with either 649-IQGAP1 or 488- IQGAP1(CD), compared to 549-mDia1(FH1-C) (Figures 4B-C and S3A), as expected." But why should 549-mDia1(FH1-C) co-localize with 488-IQGAP1(CD) at all, given that IQGAP1(CD) lacks barbed end binding capacity and 549-mDia1(FH1-C) the binding site for IQGAP1. In Fig. 4C, however, it appears that there is still a colocalization of about 70% with this combination. How can this be reasonably explained?

Referring to Fig. 4D the authors then state "Molecules of 549-mDia1(Δ DAD) processively tracked the plus-ends of actin filaments (Fig. 4D). Given that only the merged channel is shown, I cannot detect 549-mDia1(Δ DAD) (shown in green) on filament barbed ends. Please provide separate channels in an additional SI figure to support this statement.

This behavior is also not seen in the corresponding movie 7, where I expected to see the formation and disassembly of mDia1(Δ DAD)-IQGAP1 complexes at the barbed ends of the filaments, as described in the text. However, this movie is problematic for a number of reasons. Indeed, an mDia1(Δ DAD)-IQGAP1 complex appears at the barbed end at 355 s and remains at the filament end until the 510 s stage, apparently causing arrest of filament elongation. Then, at 515 seconds, there appears to be extremely rapid filament growth. Notably, calculation of this rate indicates about 250 subunits/sec, which is not possible in this experimental setting. In my view, the only explanation for this is that the end of the pre-existing filament (which continued to grow during the assumed pause) subsequently landed in the TIRF plane from above and became visible again, as is often seen in TIRF imaging. Therefore, the message of this movie implying that IQGAP1-mDia1 complexes arrest barbed end growth needs to be corrected or supported by more convincing video material. Another problem is the inappropriate marking of proteins at the filament end. At 535 sec the arrow head in magenta seems to point at IQGAP1 at the filament end. However, this spot is evidently some immobile IQGAP1-aggregate, which is present throughout the entire movie. Just in one frame, it seems to coincide with the filament end. Moreover, why is mDia1(Δ DAD) also seen within the filaments that do not contain IQGAP1 (e.g. at 415 sec)? The labeling of mDia1 with green arrows seems to correspond more to the authors' idea than to the actual localization of formin. I would strongly encourage the authors to re-examine their data more thoroughly and provide representative and convincing video material.

Fig. 4E: Please plot elongation rate (and not filament length) over time. Since IQGAP1 (red line) does not appear in the plot, the box with the colored lines of the proteins used above seems inappropriate.

Fig. 4F: In contrast to experiments conducted with mDia1(Δ DAD), the filament elongation rates of mDia1(FH1-C) in the presence of PFN1 seems not to be significantly affected by the absence or presence of IQGAP1. Please explain and state whether these measured differences are statistically significant. Once again, filament elongation rates are too slow.

Fig. 4G: Why is the filament elongation rate (in the presence of PFN and mDia1(Δ DAD)) inhibited by the presence of IQGAP1 (CD), which does not bind to the barbed end?

The data in Fig. 5 and associated Movie 8, which is intended to address whether IQGAP1 affects mDia1-CP decision complexes by two-color actin TIRF assays are more or less irrelevant because these experiments were apparently performed without PFN. The interpretation of these experiments is not accurate, given that most, if not all, polymerization-competent actin monomers in cells are thought to be in complex with PFN, which in turn is essential for formin-driven acceleration of filament elongation (Romero et al., 2004, Cell; Kovar et al., 2006, Cell). Moreover, PFN has been shown to increase the affinity of mDia1 to the barbed ends of filaments by bringing the formin FH1 domains into transient contact with the barbed end (Cao et al., Elife, 2017). Therefore, it is essential to investigate the ability of IQGAP1 to displace mDia1-CP decision complexes from filament barbed ends in the presence of PFN.

The data in Fig. 6 and associated Movie 9 were intended to understand how the combination of two capping factors (IQGAP1 and CP) results in more free ends. But why did the authors only analyze non-polymerizing actin filaments in the presence of IQGAP1, mDia1(Δ DAD), and CP, and not polymerizing filaments in the presence of these factors and PFN to better approximate the conditions of the cellular context? I simply do not understand the logic of this experimental approach using static and not growing filaments in the presence of PFN. Please explain.

In Fig. 6A and Movie 9, IQGAP1 in the IQGAP1-mDia1(Δ DAD)-CP complex is not seen in the overlay. Please provide an SI figure and a movie with separate channels to allow localization of individual components on barbed ends. There also seems to be considerable bleaching, which is not that surprising since Alexa-405-labelled actin was used. As mentioned above, the TIRF method section is not described in sufficient detail and lacks information about used laser power, exposure times and preparation of coverslips etc. I also do not fully understand the painted Fig. 6B. Is the drop in fluorescence intensities exclusively caused by dissociation of the proteins from the filament ends or simply by bleaching. Once again, separate channels and detailed information about processing of the images are instrumental to evaluate the data. Instead of showing this painted graph, the authors are encouraged to show measured fluorescence intensity profiles of respective proteins over time.

I also have a conceptual problem with the proposed mechanism. In the Bombardier et al. (2015) study analyzing the decision complex, the authors hypothesized that active mDia1 binds and blocks only one protofilament end, while the other protofilament

end is occupied by one of the two CP subunits. In my opinion, it is therefore very difficult to imagine how this could be mechanistically possible with three components simultaneously binding to the barbed end.

The authors are keeping a low profile in this regard, but this is an issue that needs to be discussed in more depth.

Is the ternary IQGAP1-mDia1-CP complex also seen when mDia1 (FH1-C) is used?

In experiments related to Fig. 7, the authors have attempted to test the physiological significance of these biochemical activities in cells. However in my opinion, these experiments do not come close to addressing the core of this question. How do they define "physiological" when they force cells to spread in confinement on micropatterns? And how do they assess the physiological significance of these complexes in the cellular context, if they exclusively look at potential function of IQGAP1 in morphology and cell migration?

In Fig. 7B, they seem to have measured only area unrelated to the actual fluorescence, but not morphology. Potential changes in cell morphology could be much better analyzed by looking for instance at different parameters such as roundness, ellipticity etc. of unconfined cells. I also doubt that these measurements are statistically valid. The IQGAP1-KO re-expressing Halo-IQGAP1 (red bar) is supposed to be statistically different from the IQGAP1-KO (blue bar), but the values are virtually identical. This also accounts for IQGAP1-KO re-expressing IQGAP1(CD) (pink bar). Furthermore, it appears that statistical analysis selects only those pairs that fit into the picture. Why is NIH 3T3 WT not compared to the KO re-expressing wild type IQGAP1? This would also reveal a statistical difference. Why are error bars show as SEM and not as SD? Expression and localization in transfected cells have not been checked. So how do they know that analyzed cells express respective constructs? Taken together, I cannot make sense of these experiments and recommend removing this data completely.

In Figs. 7 C and D, it is then proposed that the loss of IQGAP1 results in the loss of the ventral stress fibers. Unlike NIH 3T3 cells in which numerous stress fibers are visible, the IQGAP1-KO and reconstituted cells expressing either IQGAP1 or IQGAP1 (CD) all seem to have few stress fibers, albeit quantifications of global phalloidin fluorescence intensities seem to be increased in the latter. Besides the fact that this is not an accurate method for the analysis of (ventral) SFs, these findings are not explained or discussed. Instead of measuring global phalloidin fluorescence, the authors should directly quantify number and phalloidin fluorescence of stress fibers (and possible focal adhesions) of unconfined cells. If they want to make the point that these are contractile stress fibers, then they should also stain the cells with myosin II or alpha-actinin to visualize the striated patterns of these structures.

In Fig. 7E, the authors analyzed the behavior in wound healing assays using the same collection of cells. Important controls are missing. Please provide a blot depicting the expression level of respective constructs. It would also be important to know the fraction of cells expressing these constructs, since these cells were only transiently transfected. But again, where is the connection to the complexes in this experiment?

NIH 3T3 cells (which should not be termed "endogenous") are rarely diploid. They are mostly tri- or tetraploid (Leibiger et al., 2013, Journal of Histochemistry and Cytochemistry). Thus, NIH 3T3 cells should not be classified as (+/+) and IQGAP1 KO (-/-) unless the authors have determined the ploidy of the cells.

The authors mention in the method section that they use artificial intelligence to denoise their images, but the program/algorithm is not mentioned. As this approach removes information from the original images, potentially increases resolution and modifies gamma, I am not sure if this is in line with the current JCB image and video guidelines.

JOURNAL OF CELL BIOLOGY 202305065 REVISION PLAN

We have described how we will address each of the reviewer comments below with the reviewer comments in blue and our responses in black.

EDITORIAL STATEMENT:

You will see that, while all reviewers were intrigued by the proposal that IQGAP1 can promote the turnover of mDia1 and capping protein from actin filament barbed ends, all raised concerns that experimental methods lacked essential details. These concerns were compounded by data presentation and figure organization that reviewers felt needed improvement. In addition, all reviewers suggested additional experiments and data analyses that would be necessary to confirm the main conclusions of this study. Reviewers 1 and 2 made important suggestions to this end, and especially reviewer 3 raised several important points that would need to be resolved for example on the variable elongation rates in Figures 1-3, multiple concerns related to Figure 4, and quantification of stress fibers in Figure 7. Additionally, we agree with Reviewer 2 that the use of orthogonal methods to support claims made with TIRF imaging would be helpful towards resolving some of the key concerns.

We feel that the requests made by the reviewers are more substantial than can be addressed in a typical revision period. If you wish to expedite publication of the current data, it may be best to pursue publication at another journal. However, given interest in the topic and the JCB's interest in publishing this work, we would be open to resubmission to JCB of a significantly revised manuscript that fully addresses the reviewers' concerns and is subject to further peer-review. Should you wish to pursue publication with a revised manuscript, please provide a plan for revision with point-by-point responses to reviewer concerns. If and when you would like to resubmit this work to JCB, please contact the journal office to discuss an appeal of this decision or you may submit an appeal directly through our manuscript submission system.

Overall, we believe that each reviewer provided valid, constructive, and, most importantly, addressable criticisms of our first submission. We have made substantial changes to the organization and presentation of the data throughout the manuscript, added more detailed descriptions to the text addressing experimental logic, methods performed, and limitations of this work. Additionally, we have conducted numerous additional experiments to address critical reviewer suggestions for our experimental analyses. **Due to the substantial feedback from each reviewer, we have briefly summarized how we addressed points made by several reviewers and those of interest to the editor. A point-by-point discussion follows this summary below.**

1. Reviewer concern: Why are the actin filament elongation rates (actin alone controls) in TIRF assays inconsistent?

The TIRF microscopy assays performed in this work were conducted over more than 14 different sessions spanning a period of four years (particularly, **Figure 2**). As pointed out by reviewers, the most likely reason for this discrepancy is differences in actin concentration due to day-to-day measurement errors from our nanodrop. This idea is reinforced by several published articles that report a range of filament elongation rates spanning 5.6-10.9 subunits $\mu\text{M}^{-1} \text{s}^{-1}$ (DOIs: 10.21769/BioProtoc.2146; 10.1529/biophysj.104.047399; 10.1016/j.cell.2005.11.038). Notably, Hoeprich et al., 2022 (DOI: 10.1091/mbc.E21-04-0211), which has a nearly identical experimental setup as ours, published mean filament elongation rates for OG-actin alone controls in a similar range as our values (7.4 ± 1.8 subunits $\text{s}^{-1} \mu\text{M}^{-1}$).

Unfortunately, we could not use actin labeled on Cys₃₇₄ and still visualize mDia1-polymerized actin filaments as the filaments get dimmer in the presence of profilin-1 (DOIs: 10.1074/jbc.RA119.012000; 10.1021/bi980093l; 10.1021/bi9720033). Thus, we compared the effect of different labels and labeling approaches on actin (**Figure S2C-D**). We learned that certain actin labels significantly affected the mean rate of elongation for actin filaments. **Importantly, the labeled actin used within each figure panel and for each experimental comparison is consistent, and appropriate controls (actin alone and actin & FL-IQGAP1) are also included.** We have revised the layouts of each figure and accompanying text to better emphasize these and other details for each experiment.

2. Reviewer concern: The conclusions of this study will be strengthened with an orthogonal (non-microscopy-based) assay to TIRF, specifically bulk “seeded” pyrene assays.

We agree with the reviewers. We now include seeded actin elongation assays as an orthogonal way to show that IQGAP1 but not the Barbed-end Association Deficient, IQGAP1(BAD) (formerly referenced as IQGAP1(CD)), reduces end-based elongation (**Figure 3A**). We further used these assays to compare our SNAP-tagged IQGAP1 proteins with untagged controls and to show that IQGAP1 influences the actin elongation rate in the presence of mDia1-CPz decision complexes (**Figure 5A-C**). Thank you for this suggestion.

3. Reviewer concern: Why are different labels used to compare the association of different SNAP-IQGAP1 and SNAP-mDia1 proteins?

There was not a good rationale for why the original labels were used in the initial version of this manuscript. In the revision, we repurified each of these proteins (SNAP-IQGAP1, SNAP-IQGAP1(BAD), SNAP-mDia1(FH1-C), and SNAP-mDia1(Δ DAD)) and labeled them more intentionally. Each SNAP-IQGAP1 protein was labeled with a 488-dye and each formin protein was labeled with a 549-dye for a more reasonable comparison. We chose these dyes because they do not blink and they have spectral properties that can be effectively separated with our imaging systems. These efforts improved our confidence and image quality but did not change the overall result of the experiment in **Figures 4B-C** and **S3E-F**: either IQGAP1 construct associated significantly more with mDia1(Δ DAD), which retained the IQGAP1-binding region.

4. Reviewer concern: It is difficult to see the presence of various molecules throughout the manuscript.

As requested by multiple reviewers, we have performed additional imaging experiments and substantially rearranged the manuscript figures to include kymographs of individual actin filaments under various conditions, including actin alone and in the presence of IQGAP1 (**Figures 1E, 2C, 3F, and 5D**). For multi-wavelength kymographs and montages, we now include individual panels for each imaging channel and merged views (**Figures 3E-F, 4E', 5D, and S3G'**).

5. Reviewer concern: Are the proteins used in this study pure and functional?

While there is some batch-to-batch variability, we appreciate this valid concern and provide additional Coomassie gels showing the purity of each IQGAP1 protein used in this study (**Figure 1B, S1A-B**), the SNAP-tagged IQGAP1 proteins (**Figure S2A-B**), and untagged and SNAP-tagged versions of mDia1(FH1-C) and mDia1(Δ DAD) (**Figure S3A**). As already noted in the text, we are unable to obtain abundant quantities of these proteins and the protein that gave us the most concern was IQGAP1(745-1024), which was not stable and does not bind the purification column.

One reviewer had concerns about the functionality of mDia1(Δ DAD), expecting to see a larger effect on actin assembly in the presence of profilin-1 in **Figure 4D**. Thus, we tested our proteins

in standard pyrene fluorescence actin filament assembly assays comparing the unlabeled and SNAP-formin proteins. Unsurprisingly, each formin construct (SNAP-tagged or untagged) was active and capable of stimulating actin assembly in the presence of profilin-1 (**Figure S3B-C**). We performed a similar comparison in additional TIRF assays where there were no surprises - either formin, untagged or SNAP-tagged stimulated actin assembly ~5-fold in the presence of profilin-1 (**Figure S3D**). Thus, the concern regarding **Figure 4D** is likely due to differences in actin concentration on the days these data were collected. This may also explain the relatively large spread of datapoints for reactions containing formin and profilin-1.

6. **Reviewer concern:** Are several end-binding proteins bound to the barbed end of the filament or near the end of the filament?

One reviewer questioned whether IQGAP1 was bound to filament ends or *near* filament ends, which we feel we would be unable to unambiguously resolve (to be unambiguous it may require either structural studies or cryo-EM, which is beyond our current capabilities). One accessible way we tried to address this concern was to generate a version of IQGAP1 that lacks the CHD domain and therefore should not bind to filament sides, IQGAP1(160-end). This protein was able to bind filament plus-ends and pause elongation significantly more than control reactions lacking IQGAP1 but also significantly less than the full-length protein (**Figure 2B-D**). We hoped that this protein might allow us to better identify plus-end events by increasing the duration or frequency of pauses, but it did not (**Figure S1C-E**). We generated a SNAP-tagged version of the protein, but the SNAP-tag seemed to interfere with some functions, and we learned that side-binding activities were not completely abrogated (**Figure 3E-K**), in line with another study that suggests side-binding requires the CHD domain and dimerization (DOI: 10.1074/jbc.M111.258772). We further addressed this concern by rearranging the manuscript to include individual channels and merges of various image panels and added a paragraph to the discussion of the manuscript to better describe the problem and why we believe our data and the data from past studies strongly imply IQGAP1 binds plus ends.

7. **Reviewer concern:** What is the transfection efficiency of the 3T3 cells used in this study?

We chose NIH-3T3 cells for our analyses because they require intact actin dynamics for migration and foundational studies of IQGAP1 used them to dissect cytoskeletal-based functions (DOIs: 10.1074/jbc.M304838200; 10.1242/jcs.044644; 10.4161/bioa.21182; 10.1091/mbc.E15-07-0489). In the previous manuscript we used Halo-tagged versions of IQGAP1, but switched to SNAP-tagged versions as we had more success detecting IQGAP1 protein levels. Regardless, we had a hard time showing transfection over 40% by Western blot (**Figure S4B**). We also probed our ability to transfect these cells using immunofluorescence assays, where the efficiency was >70% (**Figure S4C-F**). We believe this discrepancy has to do with the epitopes used to raise the antibody or recognition differences between mouse (endogenous) or human IQGAP1 which we are expressing on plasmids. We repeated all cell-based assays present in this work (**Figures 6 and S4**). The revised figures only include cells where transfection success was visually confirmed.

REVIEWER 1:

IQGAP1 is a protein that has been previously shown to have several actin-associating activities including F-actin side-binding, capping, bundling, and interacting with formins. However, it is unclear which regions of IQGAP1 are involved in its various activities, as well as how it interacts with other barbed-end associated proteins. In this study, the authors use multicolor TIRF microscopy to characterize IQGAP1's capping activity and isolate its activity to two cysteine residues in the IQ region of the protein. They examine the interactions of IQGAP1, mDia1, and

capping protein at the barbed ends of actin filaments. Finally, the authors also create IQGAP1 CRISPR KO and both wildtype and mutant rescue cell lines to determine the role of IQGAP1 in cell morphology, actin organization, and wound healing.

Overall, this study represents a substantial amount of work and specifically advances our understanding of IQGAP1's capping functions both alone and in conjunction with other proteins at the barbed ends of actin filaments. In general, I think there is a lot of good data in this paper. However, there is also considerable overinterpretation of claims and lack of detail in regards to description of the experiments performed, their quantification, and the underlying rationale. I don't want to demand a bunch more experiments be performed as I think the data generated is sufficient. However, I think a substantial and critical reworking of the manuscript is necessary before its publication.

Thank you for recognizing this has been a substantial amount of work. We hope you agree that our commitment to strengthening our claims, adding requested details, and reorganizing the manuscript has been successful.

Major comments:

Figure 1B and Supplemental Figure 1C appear to be very similar. I am assuming that the difference is that figure 1 is His-tagged protein and supp figure 1 is GST-tagged protein? If so, this should be explicitly stated in the figure legend of figure 1 (and perhaps more clearly in supp figure 1 too, I only see it in the title). However, it is striking that the capping activity is much easier to see in the supp figure 1 examples than in figure 1 (potentially partially due to differences in protein concentration?), and the most dramatic effect to my eye in figure 1B is the bundling activity. I would suggest potentially moving Figure 1B to the supplement and instead including several kymographs of growing actin filaments in control vs +IQGAP1 conditions, to more clearly show the elongation stalling, as that is the data that sets up the story. As it stands, Figure 1E isn't super convincing either, and I think kymographs might be an easier way of showing many timepoints to more easily see the elongation effects of IQGAP1.

The reviewer is correct the old manuscript was comparing His versus GST-tags on the full-length (FL) IQGAP1 protein. The reviewer is also correct that the capping activity is dose-dependent with the GST-tagged version, but not with the His-tagged version. The motivation for doing this comparison stemmed from the original two studies (DOIs: 10.1074/jbc.M111.258772 and 10.1091/mbc.E21-04-0211) which mixed and matched tags. Even though we cleave the tags, we chose to use the His-tagged version because we believe this tag has less propensity to spuriously form dimers. We were also concerned that the size of the GST could be blocking filament-side binding activities (inadvertently making more IQGAP1 available to bind ends). Further, to not confuse readers we have removed the GST-related figure from the supplement. We are not the first to show that IQGAP1 pauses occur over a range of concentrations/stoichiometries - in the response to reviewers Hoeprich et al., 2022 also showed that IQGAP1 pauses occur over a range of actin concentrations (IQGAP1 pause duration goes down as actin concentration increases).

We have added the reviewer requested kymographs throughout the text. This was particularly challenging as we use an 'open flow' TIRF system where the actin is present as convoluted filaments rather than nice straight lines in 'constant flow' TIRF systems. We added a paragraph discussing the limitations/advantages of our TIRF system and related considerations to help to make our motivations transparent to readers.

Figure 2A represents a substantial amount of work that I think deserves to be acknowledged more thoroughly in the text. The results section should state the assays and quantification performed for each construct, just something like "we purified # truncations of IQGAP1,

performed TIRF microscopy experiments of actin in the presence of each unlabeled IQGAP1 truncation, and quantified XXX for each truncation." The schematic in Figure 2A itself also needs further clarification. What is represented by the dark shaded purple portions in 745-1450(IQ1), 745-1450(IQ1-2), and 745-1450(IQ3)? In the methods, it seems as though these are different mutation sets, but this should be stated clearly in both the figure legend and the results. In general, the rationale behind these point mutations needs clarification. Why were these tested? Were other mutations tested? What is the general structure of this domain and how are these residues postulated to affect capping abilities?

We would like to thank this reviewer for encouraging us to give more showtime to the data in **Figure 2**. We have clarified the sentences to better highlight the number of proteins purified for this specific analysis and changed how the data were represented to better show the number of replicates performed. We now better explain the coloring of the schematic in both the text and figure legends and importantly, the rationale for how we came to make various truncations and mutants.

As shown, I'm not fully convinced that there isn't an issue with dimer formation in the CD mutant. In the methods, the authors state that oligomeric state was determined by photobleaching of complexes adsorbed to the coverglass surface. It's not uncommon to have protein sticking to the coverglass surface during these types of experiments, but it's also not super uncommon to have protein aggregates also adsorbed to the coverglass surface, and as such I do not think this is an appropriate method to determine oligomeric state. I would be more confident if the photobleaching experiments were performed on IQGAP1 proteins associated with the barbed ends of actin filaments, though I realize those experiments might be difficult. Additionally, in the graph in figure 3C, why are different fluorophores used for the WT IQGAP1 and CD mutant? And what is the reason for the differences in the predicted values for a monomer, dimer, trimer in the WT vs CD mutant condition? If this is a difference in protein labeling, the labeling percentages should be clearly stated as well as the math performed to generate these values. In general, much more information is needed in the methods and/or results section to clarify how these experiments were performed, how the predicted values were generated, and how the predicted vs experimental values were compared. To address these concerns, either another method can be used to determine dimerization efficiency, or (2) more details on the methods/results can be provided as shown above, these results can be moved to the supplement, and the claims can be substantially softened to state that it is still possible that dimerization efficiency may play a role in ability to cap actin filament barbed ends.

Many groups have used this method to assess oligomeric state (<https://doi.org/10.1091/mbc.E20-09-0568>; [10.1016/j.celrep.2019.08.074](https://doi.org/10.1016/j.celrep.2019.08.074); including Hoeprich et al., 2022 for FL-IQGAP1). For the concentrations and amount of protein that we can purify, this was the most suitable approach to begin to probe this question. We wish we could include step-photobleaching on the ends of an actin filament (this would be cool). We tried this experiment, but with the instrument setup we have access to, we were not successful.

The reviewer is correct that the differences in values were due to the percent label of the proteins. We now include details on the methods used for this approach and more detail in the figure (Figure 3B-D) and have repeated this experiment with proteins labeled with the same fluors (i.e., 488-SNAP-IQGAP1, 488-SNAP-IQGAP1(BAD), and 488-SNAP-IQGAP1(160-end). We also were more conservative on specific claims of oligomeric status in the text and point out the limitations and rationale for using this approach as mentioned above.

This comment also references concerns about the details of our TIRF imaging/coverglass coating procedures. The revised text now includes much more detail on these methods.

Similar to the point above, I do not think the experiments performed in Figure 4B are appropriate to determine mDia1-IQGAP1 interactions. I understand the point, which is to determine whether mDia1 and IQGAP1 interact before associating with the actin filament barbed end, but again I find it troublesome to quantify interactions based on colocalization of proteins adsorbed to glass (I'm assuming this is the case as these details are not mentioned in the methods), as aggregates could easily stick to the glass, as is demonstrated by the fact that the mDia1-IQGAP proteins that are theoretically unable to interact still demonstrate a good amount of colocalization via this method. I suggest removing these results completely. Instead, supplemental figure 3B and C could be moved to the main figure. Again, these experiments should also be more clearly described in the results - how are preformed complexes assembled? When are they added? How is the quantification performed? Even if it's done manually, stating "we examined each actin filament barbed end for addition of IQGAP or mDia1. Upon binding, we quantified whether the other protein was also present" would provide clarity. It's hard for me to parse which details are included in the methods, as at times it is hard to tell which part of the methods refer to which experiments, so the methods should be subdivided into the different experiments and details added for clarity.

As this reviewer points out, at face value using this method alone to decipher binding interactions could be troublesome. However, several methods have been used to assess mDia1-IQGAP1 interactions by other groups previously, including reciprocal pull downs from cell extracts and by formin activity in bulk pyrene assays ([10.1074/jbc.RA119.010476](https://doi.org/10.1074/jbc.RA119.010476); <https://doi.org/10.1083/jcb.200612071>). Our goals were: 1) to assess both mDia1 constructs for the sole purpose of seeing if mDia1-IQGAP1 interactions strengthened or diminished activities at filament ends; and 2) to see if mDia1-IQGAP1 association in our hands increased with the protein that retained the IQGAP1 binding site, i.e., mDia1(Δ DAD) (**Figures 4B-C and S3E-F**). We repeated the original experiment with newly purified and more appropriately labeled 488-IQGAP1 and 549-labeled formin proteins. As expected, there was more colocalization with Δ DAD than FH1-C, and a truly matched comparison (i.e., both IQGAPs labeled in the same color compared to both formins the same wavelength, but different than IQGAP1) improved these results.

Figure 5 needs to be explained more thoroughly in the text. The results are jumped right into, but the set up of the experiment should be described first as well as what is defined as a 'blocked end'. In general, I would actually prefer if the experiments were described in terms of elongation. Describing 'blocked ends' seems to me to be jumping into interpretation, whereas elongation is really what is being assessed. In any case, if the 'blocked ends' term is kept, it should be thoroughly described.

Upon suggestions by this reviewer and others we removed this line of experiments from the manuscript as it does not substantially add to the study.

In Figure 6A, the single channel images should be shown, as it is very difficult to tell when two vs three components are present on the barbed end. I think an example first image in conjunction with a kymograph may be the best way to go here, as I suggested for Figure 1. I also find Figure 6B hard to follow, I'm not sure at all what is being shown here. In general, I think this figure should be rethought in terms of how to display the data, as these results are potentially very cool, but hard to parse as they are currently shown, and difficult for me to fully evaluate.

We agree that the original images associated with this figure were not very clear nor did they adequately show which components were present on filament ends. To improve the quality of this figure, we repeated these experiments with our newly purified SNAP-proteins with each molecule captured in a consistent wavelength, i.e., actin was imaged in the 405 wavelength,

IQGAP1 molecules were in the 488 wavelength, formin molecules were in the 549 wavelength, and capping protein (CPz) was in the 647-wavelength (**Figure 5D**). The use of actin filament seeds improved image quality as the actin was prone to photobleaching and there was no additional background fluorescence from unpolymerized monomers.

We also took this reviewer's suggestion to display the data as kymographs that show the individual channels of interest and a merge of all the wavelengths. We think the revised version of **Figure 5** more clearly makes our points and thank the reviewer for their suggestions.

Much more detail is needed to explain Figure 7. The results section needs to make clear that these cells are on a micropatterned surface. To my eye, the shape of the cells don't differ that extensively. In what way does the shape change, and what is actually being quantified? Also, I like the color scheme, but all panels of Figure 7C should be shown in grayscale for easier viewing of the actin organization. What is being quantified in figure 7D? The methods state how images were processed, but not what is being quantified. If it's truly just the actin signal, I don't find that to be a quantification as that could just vary based on cell size or numerous other variables, and that doesn't demonstrate that IQGAP1 results in a loss of 'tension-dependent stress fibers', as that specific cytoskeletal network is not examined at all, but rather just the actin in general. Again, much more detail is needed here as it is unclear what these experiments are showing. The wound healing assay is cool, so maybe everything else should be moved to the supplement? Where do the rates come from in Figure 7G? With the data as it is currently in this paper, I would not be comfortable putting rate constants on this diagram.

We have clarified what is being counted in this figure in both the text and the methods (**Figure 6**).

Briefly, it is the total pixel count of actin or microtubule signal from a binarized image. We converted the cell images in **Figure 6** and most of the images in **Figure S4** to grayscale for easier viewing. In line with other reviewer concerns, we repeated all the assays in **Figures 6 and S4** visually confirming that the cells were appropriately transfected. In addition to these suggestions, we now include circularity measurements as a different readout of cell morphology under unconstrained conditions (**Figure 6A-B**). Cells expressing endogenous (mouse) IQGAP1 were the most circular, whereas IQGAP1 knockout cells were significantly less circular ($P < 0.0001$). Circularity values could not be fully rescued by expressing human IQGAP1 or SNAP-IQGAP1 off plasmids in the knockout cell line, though these rescues were not significantly different from each other and significantly more circular than mock-treated knockout cells. We appreciate the reviewer's suggestion to include this metric in our study.

We have removed the comment on tension specific stress fibers, the reviewer was correct that we did not explore that network. Further, we ended up removing the rate constants from the paper – in the end we did not feel that we were able to count enough molecules of various end binding proteins to feel confident in them.

As mentioned throughout my comments, in general, I think the whole paper could use more detail. The methods section is very sparse - much more detail is needed on how each type of experiment and quantification is performed. It would be easiest to subdivide the methods by experiment and quantification type and make sure adequate detail is provided for each experiment and quantification performed. In addition, I found that the results section often glossed over descriptions of the experiments, and more detail would be useful to the reader. It's totally OK to repeat information in the results, figure legends, and methods, I would rather have all of the information super explicitly stated.

We thank this reviewer for the clarity and encouragement and suggestions on what information would be helpful (even if repeated) and the suggested flow. We have expanded the results and the methods sections to more appropriately detail how we did what we did and why we did it. We believe this information is more transparent and these suggestions make the manuscript easier to read than the previous version.

Minor comments:

In general, I find all of the distribution of frequency events figures (Figure 1G, 2F, 6C, Supp Fig 1G, 2C, 2E, 2F) hard to follow, and it's not clear to me why points are only included every 25 seconds (at most). Is this to simplify the plot? It's hard to assess the fit when so few points are shown. I can understand why it's difficult to show all of the points, but if that's the case, the data should be shown in a different way. If this type of diagram is kept, the R² values of the fits should always be listed (they are in Figure 6C, but not in the others).

The data were collected in 5 s intervals. We applied a few rules for measuring elongation pausing events, mainly that the event needed to occur for at least 3 consecutive frames (15 s). Thus, the smallest pause we can reliably measure is every 20-25 s. We count many pauses on many filaments and in most cases, they were either zero, or in the 20-25 s range – basically emphasizing data outliers rather than the meaningful data (**Figure S1D**). Thus, we used these values to display the frequency of different pause durations as the Gaussians in **Figures 1I, 2F, and S1E-F**. The R² values for plotted Gaussians ranged between 0.9967-1.000, whereas the R² values for non-pausing conditions fit using a non-linear fit ranged between 0.9426-0.9998. This information is now included in the figure legend.

In this manuscript the terms plus and minus end are used for the ends of actin filaments. However, in figure 4E I found a wayward 'barbed end'. I'm more used to seeing barbed and pointed end as terms for actin, but I'm OK with plus and minus end. However, maybe it would be good to mention both in the intro to appease actin terminology sticklers?

We believe the terminology of plus and minus ends is more accessible to the general reader, who may be less familiar with myosin S1 decoration of actin filaments. We chose to continue to use the term "plus-end" throughout this work but took care to point out our terminology for traditionalists.

In Figure 2A, in the label for "745-1450(IQ1-2), the I is a 1.

We have corrected this. Thank you!

Is IQGAP1 diffusing/sliding on actin filaments ever observed as suggested in the discussion?

It would be impossible for us to truly resolve this in our imaging system, but it is an intriguing idea and we believe we have seen some instances where IQGAP1 moves from a plus-end to the side of a filament by this mechanism. We felt that this idea deserved more consideration especially in line with what could be happening with formins that also exhibit this behavior. Thus, we have added a section to the discussion on this topic.

Starting with Figure 3, the text states that IQGAP is 649 labeled, while the figure says 549 labeled. I would check all of these numbers in-text, in figure legends, and in the figures themselves. Also a rationale as to when different fluors are used in situations where there is a direct comparison should be added.

This is a great point. As addressed above we now directly compare the constructs with the same fluor and have verified the labeled proteins.

Did you ever test whether IQGAP1 is capable of releasing autoinhibition of mDia1? One might hypothesize that based on its binding site within the DID domain. It would be worth mentioning if this experiment was attempted and any potential results (even if negative).

This is a great idea! It has already been tested by at least two different research groups (<https://doi.org/10.1083/jcb.200612071> and [10.1074/jbc.RA119.010476](https://doi.org/10.1074/jbc.RA119.010476)). Most recently the Wilde group combined the N-term and C-term fragments of mDia1, which associate and behave like the full-length protein. In bulk pyrene assays it appears that IQGAP1 relieves mDia1 autoinhibition. We did not try to recapitulate this experiment or perform any assays with the full-length mDia1, nor do we have the specific formin constructs used in these studies. We will now mention this finding and use it to explain the logic of experiments in **Figures 4, 5, and S3**.

There are multiple sentences in the manuscript that need to be clarified. I've listed several here: Thank you! We will focus on the clarity of our sentences including all those specifically mentioned below.

Line 47: "Filament polymerization is further refined by complexes with formin to enhance (e.g., APC, CLIP-170, or spire) or with CP to limit (e.g., twinfilin or CARMIL) assembly10-19.". Is the point that these proteins form complexes with formin or CP? I think this sentence should be reorganized because I was unclear on what it meant.

This sentence has been reworded. Thank you for pointing out that its meaning was not clear.

Line 92: "We used truncation and mutational analyses with 75 nM protein to further delimit the region and residues important for plus end capping because it is sufficient for plus-end binding (kD = 25-35 nM) and filament pauses were obvious with the full-length protein." Should be "because that protein concentration is sufficient for wild-type IQGAP1 plus-end binding".

We have reworded this sentence for clarity. The reviewer is correct in understanding our intentions.

Line 149: "Notably, stochastic pauses in formin-mediated actin filament growth occurred only when IQGAP1 could directly bind formin i.e., 549-mDia1(Δ DAD), and were rarely observed for reactions performed with 549-mDia1(FH1-C) or IQGAP1(CD) (Figure 4D-G)." The way this sentence is organized, it appears to suggest that IQGAP1(CD) is unable to bind mDia1, but I don't think that is correct right? Should be clarified.

This sentence has been reworded for clarity. Indeed, we have no evidence that IQGAP1(BAD) and formin are unable to interact. Thank you for pointing this out.

Line 174: "Remarkably, the addition of IQGAP1 to either condition resulted in significantly fewer blocked ends (P {less than or equal to} 0.0001), although significantly fewer free-ends compared to buffer controls ($P = 0.897$) (Figures 5F-G and S4C; Movie 8)." $P=0.897$ is significant? I may be confused with how this sentence is constructed or what is being quantified/compared here, or this may not be the correct p value? Please clarify.

We removed this analysis and text from this version of the manuscript.

Line 231: "formin mutants to show that both IQGAP1 and mDia1 can co-exist on filament plus-ends without binding each other or affecting formin-based activities." Was this shown? To me this suggests an experiment like Figure 4D but with the mDia1(FH1-C), but I don't see this experiment.

In the first version of this manuscript, we did not show this directly. Instead, we came to this interpretation from the mean elongation rates of actin filaments present in the conditions of **Figure 4D**. The reviewer comments inspired us to use 3-color TIRF microscopy to visually confirm that formin and IQGAP1 could co-exist on filament ends. Indeed, in the example now in

Figure 4E and 4E' we show mDia1(Δ DAD) and IQGAP1 together on a plus end. The revised version of the manuscript has been refocused more specifically on the mDia1(Δ DAD) formin for interactions with CPz and IQGAP1. However, we did perform the 3-color analysis with mDia1(FH1-C) and IQGAP1 and observed 29 mDia1(FH1-C)-IQGAP1 complexes (Figure S3H), which are possible if both proteins can simultaneously bind ends, or because each of version of IQGAP1 used still retains the residues in the C-terminal LBR region that are essential for formin-binding.

REVIEWER 2:

The manuscript entitled "Coordination of actin plus-end dynamics by IQGAP1, formin and capping protein" by Morgan L. Pimm et al. investigates the role of IQGAP1 in regulating actin dynamics at the barbed ends of filaments. The authors propose that IQGAP1 promotes the turnover of end-binding proteins by reducing their residence time.

The manuscript contains a substantial amount of data and sheds new light on the interaction between IQGAP1, formin and capping protein at the barbed ends of actin filaments. However, I have some concerns about the following points:

Thank you for acknowledging the amount of data we produced to try to investigate this problem. We believe we have addressed your concerns with new experiments and substantial reorganization. Thank you for your constructive and helpful feedback.

First, Figure 6, which is crucial to support the main message of IQGAP1 reducing residence times, needs significant improvement. The figure is currently difficult to comprehend, and it should be revised to ensure clear and understandable representation of the data. In particular, Figure 6C, which displays the lifetime of various complexes, appears excessively small and complex, making it hard to read. The superimposed fits obscure the actual experimental data points, and the low R² values for CP and mDia1-CP raise doubts about the validity of the exponential fit for these data.

As stated above, we agree that the original presentation of this figure and the data presented therein needed additional explanations and experimental work. Restated here for ease of readability: "To improve the quality of this figure, we repeated these experiments with our newly purified SNAP-proteins with each molecule captured in a consistent wavelength, i.e., actin was imaged in the 405 wavelength, IQGAP1 molecules were in the 488 wavelength, formin molecules were in the 549 wavelength, and capping protein (CPz) was in the 647-wavelength (**Figure 5D**). The use of actin filament seeds improved image quality as the actin was prone to photobleaching and there was no additional background fluorescence from unpolymerized monomers."

We also chose to display the content of this figure as kymographs showing individual and merged channels of interest of all the wavelengths. We present this information as a confirmation that these complexes can form and that they are dynamic but make no claims about the affinities that could be driving these interactions. Even with the additional analysis, as Reviewer 2 points out, there were not enough observations for each condition to determine these values in a reliable or confident manner.

Additionally, Figure 6B, illustrating fluorescence intensity profiles, is perplexing, and the method employed to obtain these profiles is unclear. It would be beneficial if the authors considered utilizing alternative analyses, such as kymographs or other appropriate methods, to illustrate the duration of protein presence at filament ends in the presence or absence of IQGAP1.

We agree that we did not do a great job presenting this data the first time. As suggested, we have expanded the methods with pertinent details, and we now use kymographs in this figure (Figure 5D) and throughout the text to better illustrate the activities of plus-end binding proteins.

Second, there is a lack of information regarding the number of times the experiments were repeated. This omission raises concerns about reproducibility and statistical significance. It is essential for the authors to clarify whether the experiments were performed with independent replicates to ensure the reliability of their findings.

The omission was not meant to obfuscate the data or interpretation of the results and we apologize for frustrating the reviewers and readers of the initial version of this manuscript. A major drawback of TIRF microscopy measurements on actin filaments is that there are often more filaments than can be reasonably counted from a single field of view (FOV). Our efforts to make sure we get adequate coverage of the population of filaments present in each FOV from each condition and replicates gets complicated quickly. We typically count 17 filaments per FOV from three independently performed reactions, totaling 51 filaments. However, our analyses had more conditions than could typically be collected on a single day. In these instances, additional actin alone, actin and IQGAP1, or other relevant controls were also measured. In most cases each of these independent replicates were performed with different batches of proteins. **We now clarify these methods in the text and include the n values in each figure legend. In the most complicated situations, we have added this information directly to the figure panels. We hope that these actions make our efforts more understandable.**

Third, although the characterizations in the manuscript rely entirely on fluorescence microscopy (TIRF), it is questionable whether this method alone is sufficient to carry out all the quantifications and draw the conclusions presented. It would have been worthwhile to include bulk experiments using pyrene fluorescence or other appropriate techniques to strengthen the main conclusions of the study.

As stated above, we agree that complimentary approaches strengthen the overall message of this work. We now include the requested seeded pyrene assays to validate IQGAP1's function (**Figure 3A**) and its function with mDia1-CPz decision complexes (**Figure 5A-C**).

Finally, some figures are probably not essential as main figures and could be moved as supplementary figures, e.g. the last figure on cell biology contains very little information.

We agree and have performed a substantial reorganization of the original figures. We have removed several of the original figures (i.e., the GST-tag comparison, the two-color actin filament assays, and figure panels associated with binding affinities). We disagree with the reviewer about the cell biology content, and therefore this figure remains (**Figure 6**). As stated above, previous studies have shown IQGAP1 plays important roles in actin-based processes in cells. The IQGAP1(BAD) mutant provides an excellent opportunity to test the specific function of IQGAP1's pausing activity in cells, as we demonstrated with multiple direct approaches that it does not have plus-end pausing activity of the normal protein. By expressing this in cells that lack IQGAP1 and assessing cellular activities we use reverse genetics to show the two cysteine residues are essential for several physiological processes! We clarify the logic underlying these experiments in the revised text.

Minor points

1. Supplementary figure 1: why is the dose-dependent response of IQGAP1 a problem? And why choose purified IQGAP1 with a His-tag that does not show a dose-dependent response (because in Hoeprich et al. MBoC 2022, they seem to have this dose-dependent response).

We believe we get the same results as the Hoeprich et al 2022. A main difference between that study and ours is that different constructs of IQGAP were tagged with different tags, whereas

ours makes all comparisons of activity from proteins purified out of the same vector. While there should be no difference in the final cleaved product, there are examples of GST-facilitating the dimerization state of various proteins during the purification process. We were concerned about these ideas, and this motivated our head-to-head test of the full-length protein comparing both tags. As mentioned in our comments to Reviewer 1 above, we chose to use the His-tag over the GST-tag to reduce because it was less likely to produce dimers or block bundling activities, which we reason could facilitate or extend IQGAP1-filament pausing events. We have removed this supplemental figure from the revised manuscript.

2. Why doesn't 649-IQGAP1 have the same activity as IQGAP1? Have the authors tried to assess the rate of actin elongation in the presence of 649-IQGAP1 without exposing the filaments to the far-red channel? Why aren't the same fluorophores used to compare IQGAP1 and the CD mutant?

We agree with the reviewer that the same fluorophores should be used for comparisons. Thus, we repeated these experiments with newly purified and appropriate labeled proteins. At the request of this reviewer, we did not use the far-red label on IQGAP1, instead we used a 488-label. In the revised analysis the SNAP-IQGAP1 and untagged IQGAP1 are not significantly different from each other (**Figure 3J**).

4. In general, the plus end is used for microtubules and the barbed end for actin filaments, so the title could be modified accordingly.

As mentioned above, we are using the terminology “plus end” to make this work accessible to a broader audience.

REVIEWER 3:

IQGAP1 is a dimeric, multidomain, scaffolding protein that facilitates the formation of complexes with various ligands to regulate diverse cellular processes such as cell migration, cytoskeletal dynamics, intracellular signaling, and intercellular interactions. In this work, Pimm and colleagues used single- and multi-color TIRF imaging with IQGAP1 variants in the absence or presence of mDia1 constructs and heterodimeric capping protein (CP) to (i) visualize and characterize the association of IQGAP1 with actin filaments and its consequence on actin assembly and to (ii) study its effect on barbed end decision complexes that either promote or pause filament growth. They found that IQGAP1 had no effect on nucleation, but reduced filament growth at the barbed end by approximately 30%, suggesting transient capping, consistent with previous work (Pelikan-Conchaudron et al., JBC, 2011). Subsequently, they identified two residues in IQGAP1 critical for its association with filament barbed ends. Interestingly, using TIRF imaging, they were also able to visualize dimers of IQGAP1 and mDia1 or CP on filament barbed ends alone and as a multi-component complex bound to filament ends. Notably, IQGAP1 appears to promote the turnover of end-binding proteins, reducing the residence time of CP, mDia1 or mDia1-CP decision complexes by approximately one order of magnitude. All this is potentially quite interesting, but I see a number of issues with the constructs and the TIRF assays (see below). The physiological relevance of these interactions suffers from conceptual design, is not clear from the data presented, and needs to be explored in much more detail. In my view, this work is currently not suitable for publication in JCB.

Thank you for your direct feedback. Your comments have given us a lot to consider.

Specific points:

Unlike full-length IQGAP1, to my knowledge, functional full-length mDia1 has not been purified from *E. coli*. For this reason, Bruce Goode's lab has expressed and purified a functional FL-mDia1 in yeast (Maiti et al., Cytoskeleton, 2012). The Rosen lab also found that solubility and yield of the complete full-length mDia1 was very poor in *E. coli* and therefore turned to expression of mDia1 in Sf9 insect cells (Otomo et al., PlosOne, 2010). It is therefore instrumental that the activity of the new probes, in particular of SNAP-tagged mDia1 Δ DAD, which is even slightly larger compared to FL-mDia1, is first shown to be functional in the TIRF assay. Since mDia1 lacking its C-terminal DAD domain is not expected to be autoinhibited, fluorescent SNAP-tagged mDia1 Δ DAD must be shown to stay associated (surf) with the elongating filament barbed ends and accelerate filament growth in the presence of profilin (PFN) roughly 5-fold, as previously shown for shorter mDia1 constructs containing the FH1-FH2-C region (e.g. Bombardier et al., Ncomms, 2015).

We appreciate this very valid concern. For clarity, we do not purify full-length (FL) mDia1 in this work and are using different expression vectors/systems than the referenced articles. However, relevant to discussions prompted by this reviewer, we do note that an mDia1(Δ DAD) protein is characterized and appears fully functional in the referenced work.

We understand the important point this reviewer is making. In our revision, we characterize these formin constructs more thoroughly. We include Coomassie gels of all IQGAP1 proteins (**Figure S1A-B**), SNAP-IQGAP1 proteins (**Figure S2A-B**), and all formin proteins (**Figure S3A**), specifically untagged and SNAP-tagged versions of mDia1(FH1-C) and mDia1(Δ DAD). We compare the activity of untagged and SNAP-tagged FH1-C and Δ DAD formins in conventional pyrene assays (**Figure S3B-C**) and in TIRF microscopy-based assays (**Figure S3D**). As summarized above, the bulk pyrene fluorescence (**Figure S3B**) or elongation activity (**Figure S3D**) of each formin is stimulated in the presence of profilin-1. Further, the presence of the SNAP-tag does not significantly interfere with these activities. We also performed multi-color TIRF microscopy experiments and confirmed that each SNAP-tagged formin processively elongated or "surf" on actin filaments (**Figures 4E-E', S3G-G', and Movies 5 and 6**). Thus, our formins are not autoinhibited, they "surf" on filament ends, and are appropriate for our analyses.

Single- and especially multi-color TIRF imaging is certainly quite challenging, but the quality of the images and movies shown here is not state of the art due to the very high background resulting in a poor signal-to-noise ratio.

We agree that the original images could be improved. We now include more details about our TIRF imaging system and approach. At this reviewer's request we have repeated all multi-color TIRF microscopy experiments and related analyses in the revised manuscript.

Unfortunately, the TIRF method section is not described in sufficient detail and lacks information about preparation of coverslips, used laser power, exposure times, filters, technical setup for multicolor imaging etc. As it stands, this study will refer to another study, which will then refer to yet another study. Since TIRF imaging is a central element of this study, all of the necessary details should be described in the Methods section.

We agree there is value putting all these details in one easily accessible space. We have expanded and reorganized the methods section at the request of all the reviewers. The revised version contains all the information requested by this reviewer and an additional reference to a video protocol detailing our methods.

Moreover, the authors used streptavidin-coated chambers to tether growing filaments (spiked with biotin) to the surface. Depending on the coating conditions, this can lead to physical stalling

of the filaments due to altered stress or torsion or even road blocks by proteinaceous deposits on the surface, which makes interpretations of data difficult.

We appreciate that each system has advantages and shortcomings, and we thank this reviewer for giving us the opportunity to point out the advantages and limitations of our approach in the discussion section of the revised manuscript. We are confident that the revised methods and inclusion of higher quality images have strengthened our ability to accurately interpret our data.

An example is Figs. 1E, F and Movie 3, in which IQGAP1 is assumed to cause transient capping in 70% of the filaments. However, in this single-color TIRF experiments using only labelled actin, it is unclear whether reduced growth is in fact caused by capping of filament barbed ends by IQGAP1. These experiments and corresponding analysis of data shown in Figs. 1F and G need to be repeated with fluorescent 649-IQGAP1 to visualize capping of the protein on growing filament barbed ends and correlate it with filament growth.

Motivated by this reviewer's suggestions, we have performed the requested experiments with 488-IQGAP1 (**Figure 3E**). We also include kymographs as an additional method for detailing the pauses to filament elongation with unlabeled IQGAP1 (**Figures 1E, 2C, S1C**) and showing fluorescently labeled IQGAP1 proteins bound to ends coincide with pauses in multi-color experiments (**Figure 3F**). We also include pyrene fluorescence assays with actin filament seeds as an orthogonal measure of IQGAP1's end-associated activity (**Figure 3A**).

In Movie 6, such an experiment is shown on the left, but frankly, I was not able to detect an obvious pause in filament growth upon IQGAP1 binding to the barbed end. Out of curiosity, I converted the shown mp4 movie 6 into a TIF stack and analyzed rates of filament growth with Fiji. With 6.2 subunits/sec, the filament elongated rather slowly, but more or less constantly over the entire sequence. Moreover, when looking at specific rates from frame to frame, there does not appear to be a direct correlation between binding of IQGAP1 to the barbed end (350 s) and reduced growth. Therefore, in my opinion, the data presented does not support the conclusions drawn.

We agree that the original data presentation was not representative and have removed it. We took a lot of time to carefully repeat all multi-color TIRF experiments to improve image quality and our analyses. We replaced all previous movies with higher quality ones that were representative of the revised analysis. We also adopted the kymograph suggestions made by Reviewers 1 and 2 to show pauses to filament elongation in a more obvious manner. We thank this reviewer for their insight and efforts, which have helped to improve our work.

For this reason, fluorescent IQGAP1 also needs to be used in experiments related to Fig. 2 to clearly assess its potential effects on capping and barbed end growth.

The requested experiments and analyses are now available in **Figure 3E-F**. Indeed, SNAP-IQGAP1 is associated with filament plus-ends and sides, whereas the Barbed-end Association Deficient IQGAP1(BAD) mutant that replaces two cysteine residues with alanine mostly associates with filament sides. We include montages (**Figure 3E**) and kymographs (**Figure 3F**) of these conditions with individual wavelengths and merged channels to emphasize these points in the revised manuscript.

This brings me to the next issue. Why are the elongation rates of spontaneously growing actin control filaments so different in Fig.1 (10.2 subunits/sec) vs. Fig. 2 (7.6 subunits/sec) vs. Fig. 3 (with about 5 subunits/sec). These rates are approximately 25% to 50% lower than published values (for example 10.9 subunits/sec using Oregon-green labelled actin as reported by Kovar et al., 2006 in Cell). So how solid is the statistical data when not even the controls can be accurately reproduced? Is this possibly due to different protein concentrations used? Oddly enough, the authors determined their protein concentrations by densitometry of Coomassie

stained gels compared to a BSA standard curve and not by spectroscopy as stated in the method section. Why using such an inaccurate method that is not acceptable, because Coomassie staining is dependent on the size of the proteins as well as on their content of aromatic and basic residues. This measurement is typically only done with impure proteins to get a rough estimate of protein concentration. To exclude this, authors are asked to show a Coomassie-stained gel with all purified proteins in a supplementary figure.

We address your concern about the rates with a detailed explanation above. To summarize, many different rates have been observed for actin filament elongation (actin alone) in systems similar and different to ours. All the rates shown are in the range of values published and are reflective of day-to-day differences in actin concentration recorded by our nanodrop. We now comment on this limitation to our approach in the discussion of the revised manuscript.

Concerning protein concentrations: As a measure of quality control, we run quantitative gels to assess the level of purity of each of our proteins. At the request of the reviewer, we include Coomassie gels for each IQGAP1 protein (**Figure S1A-B**), SNAP-IQGAP1 proteins (**Figure S2A-B**), and tagged and untagged formin proteins (**Figure S3A**). We use spectroscopy to determine protein concentration in two specific scenarios: to determine the concentration of dialyzed actin at A_{290} (after seeing the purity on a gel) and to determine % label of SNAP proteins. We revised our methods to clarify these points.

Fig. 3: As mentioned above, better movies (compared to Movie 6) are needed to support the claims of Fig. 3F. Here, the authors show three examples where IQGAP1 apparently enters the filament barbed to cause a pause (red lines), followed by dissociation of the protein from filament ends to resume filament growth. These events should be clearly visible in the movie. The authors should also test an IQGAP1 mutant lacking its CHD F-actin binding sites to assess whether this mutant protein is still capable of interacting with filament barbed ends in the absence of its side binding activities. This would allow them to distinguish between binding directly to the barbed end or near the barbed end.

To address the first point, we include new movies and have added kymographs (**Figure 3F and Movie 3**). We expanded the characterization to include panels of individual image channels and multicolor merges (**Figure 3E-K**).

The comment about the CHD is a great idea and thus at this reviewer's request we generated IQGAP1(160-end) and SNAP-IQGAP1(160-end) proteins. IQGAP1(160-end) can pause filament growth and reduce filament bundling (**Figures 2A-D and 3E-K**). The loss of the CHD did not completely abrogate side-binding activities (**Figure 3H**) but did significantly reduce filament bundling compared to actin and IQGAP1 control (**Figure 3I**) ($P = 0.0059$). The SNAP-tagged version also displayed weaker pausing activity than the full-length protein (**Figure 3K**). Unfortunately, these observations did not aid in extending the length or detectability of IQGAP1 pauses but are consistent with the notion that the high affinity CHD-side binding interactions contribute to slow off-rate of IQGAP1 from filament sides (Pelikan-Conchaudron et., al 2011; Hoperich et., al 2022).

In experiments related to Fig. 4 the authors state "Single-molecule assays that contained 549-mDia1(Δ DAD) showed significantly more colocalization with either 649-IQGAP1 or 488-IQGAP1(CD), compared to 549-mDia1(FH1-C) (Figures 4B-C and S3A), as expected." But why should 549-mDia1(FH1-C) co-localize with 488-IQGAP1(CD) at all, given that IQGAP1(CD) lacks barbed end binding capacity and 549-mDia1(FH1-C) the binding site for IQGAP1. In Fig. 4C, however, it appears that there is still a colocalization of about 70% with this combination. How can this be reasonably explained?

IQGAP1 and mDia1 are known binding partners based on cell and in vitro data by other labs (see comments to Reviewer 1, above). We were as surprised as the reviewer that there was 20% colocalization between FH1-C and IQGAP1 and upon the suggestions from other reviewers that the mismatched fluors could be contributing background to this assay we repeated it with both IQGAP1 proteins in the 488 wavelength and both formin proteins in the 549 wavelength. Indeed, the previous experiment had higher background. However, in the revised experiment there is still some colocalization between mDia1(FH1-C) and either IQGAP1 protein (**Figures 4B-C and S3E-F**). In either case the observed association is significantly less than the observations for mDia1(Δ DAD) ($P = 0.03$ and $P = 0.01$, for SNAP-IQGAP1 or SNAP-IQGAP1(BAD), respectively).

Referring to Fig. 4D the authors then state "Molecules of 549-mDia1(Δ DAD) processively tracked the plus-ends of actin filaments (Fig. 4D). Given that only the merged channel is shown, I cannot detect 549-mDia1(Δ DAD) (shown in green) on filament barbed ends. Please provide separate channels in an additional SI figure to support this statement.

We show the individual channels and merged images of mDia1(Δ DAD) (**Figure 4E-E' and Movie 5**) and mDia1(FH1-C) (**Figure S3G-G' and Movie 6**) in the revised manuscript.

This behavior is also not seen in the corresponding movie 7, where I expected to see the formation and disassembly of mDia1(Δ DAD)-IQGAP1 complexes at the barbed ends of the filaments, as described in the text. However, this movie is problematic for a number of reasons. Indeed, an mDia1(Δ DAD)-IQGAP1 complex appears at the barbed end at 355 s and remains at the filament end until the 510 s stage, apparently causing arrest of filament elongation. Then, at 515 seconds, there appears to be extremely rapid filament growth. Notably, calculation of this rate indicates about 250 subunits/sec, which is not possible in this experimental setting. In my view, the only explanation for this is that the end of the pre-existing filament (which continued to grow during the assumed pause) subsequently landed in the TIRF plane from above and became visible again, as is often seen in TIRF imaging. Therefore, the message of this movie implying that IQGAP1-mDia1 complexes arrest barbed end growth needs to be corrected or supported by more convincing video material. Another problem is the inappropriate marking of proteins at the filament end. At 535 sec the arrow head in magenta seems to point at IQGAP1 at the filament end. However, this spot is evidently some immobile IQGAP1-aggregate, which is present throughout the entire movie. Just in one frame, it seems to coincide with the filament end. Moreover, why is mDia1(Δ DAD) also seen within the filaments that do not contain IQGAP1 (e.g. at 415 sec)? The labeling of mDia1 with green arrows seems to correspond more to the authors' idea than to the actual localization of formin. I would strongly encourage the authors to re-examine their data more thoroughly and provide representative and convincing video material.

We thank the reviewer for pushing us to collect better multicolor TIRF microscopy images. We now provide clearer movies (with lower background) that are more representative.

Fig. 4E: Please plot elongation rate (and not filament length) over time. Since IQGAP1 (red line) does not appear in the plot, the box with the colored lines of the proteins used above seems inappropriate.

We removed this panel from the revised manuscript.

Fig. 4F: In contrast to experiments conducted with mDia1(Δ DAD), the filament elongation rates of mDia1(FH1-C) in the presence of PFN1 seems not to be significantly affected by the absence or presence of IQGAP1. Please explain and state whether these measured differences are statistically significant. Once again, filament elongation rates are too slow.

We have clarified what statistical comparisons were made in the revised text and figure legends. The Δ DAD formin is significantly accelerated compared to profilin-actin controls ($P < 0.0001$) and significantly different from FH1-C with profilin ($P < 0.0001$). This is consistent with our interpretation of a similar comparison in Figure 1 of Gould et al., 2011 (10.1016/j.cub.2011.01.047). We also added clarification to the discussion that addresses the differences in actin rates throughout this work. In sum, we believe that the variability is within the reported literature and likely related to the actin concentration or the specific labels on the actin (Figure S2C-D).

Fig. 4G: Why is the filament elongation rate (in the presence of PFN and mDia1 Δ DAD) inhibited by the presence of IQGAP1 (CD), which does not bind to the barbed end?

We removed this data from the revised manuscript. However, this was likely possible due to direct interactions between IQGAP1(BAD) and mDia1(Δ DAD) through formin's FH3 domain.

The data in Fig. 5 and associated Movie 8, which is intended to address whether IQGAP1 affects mDia1-CP decision complexes by two-color actin TIRF assays are more or less irrelevant because these experiments were apparently performed without PFN. The interpretation of these experiments is not accurate, given that most, if not all, polymerization-competent actin monomers in cells are thought to be in complex with PFN, which in turn is essential for formin-driven acceleration of filament elongation (Romero et al., 2004, Cell; Kovar et al., 2006, Cell). Moreover, PFN has been shown to increase the affinity of mDia1 to the barbed ends of filaments by bringing the formin FH1 domains into transient contact with the barbed end (Cao et al., Elife, 2017). Therefore, it is essential to investigate the ability of IQGAP1 to displace mDia1-CP decision complexes from filament barbed ends in the presence of PFN.

The reviewer is correct that adding profilin increases formin-based actin assembly and that profilin can increase the affinity of formin for barbed ends. Our goal of the four-color experiments was simpler -- can these molecules even be present on ends? We agree that the two-color actin filament assays were not helping us to make this claim and have removed them from the revised manuscript.

The data in Fig. 6 and associated Movie 9 were intended to understand how the combination of two capping factors (IQGAP1 and CP) results in more free ends. But why did the authors only analyze non-polymerizing actin filaments in the presence of IQGAP1, mDia1(Δ DAD), and CP, and not polymerizing filaments in the presence of these factors and PFN to better approximate the conditions of the cellular context? I simply do not understand the logic of this experimental approach using static and not growing filaments in the presence of PFN. Please explain.

In Fig. 6A and Movie 9, IQGAP1 in the IQGAP1-mDia1(Δ DAD)-CP complex is not seen in the overlay. Please provide an SI figure and a movie with separate channels to allow localization of individual components on barbed ends. There also seems to be considerable bleaching, which is not that surprising since Alexa-405-labelled actin was used. As mentioned above, the TIRF method section is not described in sufficient detail and lacks information about used laser power, exposure times and preparation of coverslips etc. I also do not fully understand the painted Fig. 6B. Is the drop in fluorescence intensities exclusively caused by dissociation of the proteins from the filament ends or simply by bleaching. Once again, separate channels and detailed information about processing of the images are instrumental to evaluate the data. Instead of showing this painted graph, the authors are encouraged to show measured fluorescence intensity profiles of respective proteins over time.

The logic for using static F-actin "seeds" is as follows: 1) the 405-actin is very sensitive to photobleaching; 2) the labeled actin monomers not yet incorporated into filaments contribute background noise to the images making plus-ends difficult to see; 3) we tried to visualize actively polymerizing filaments in the presence of formin, but we were unable to reliably detect

the dimmer filaments. We also performed additional 3-color TIRF assays with mDia1(Δ DAD) or (FH1-C), IQGAP1, and actin in the presence or absence of PFN1 (**Figures 4E-E'** and **S3G-H and Movies 5 and 6**). In most scenarios, with the parameters we were able to measure, the maximum plus-end occupancy of formins was less in the presence of profilin. Further, our new seeded pyrene assays performed in the presence of PFN1 may also suggest that the more complicated TIRF assay is not required (**Figure 5A-C**). For these reasons and our original goal of assessing what proteins were on a plus-end, we believe the use of F-actin seeds in **Figure 5** was acceptable.

As requested, and for clarity we will provide individual and merged channels as well as more experimental details in the revised text.

I also have a conceptual problem with the proposed mechanism. In the Bombardier et al. (2015) study analyzing the decision complex, the authors hypothesized that active mDia1 binds and blocks only one protofilament end, while the other protofilament end is occupied by one of the two CP subunits. In my opinion, it is therefore very difficult to imagine how this could be mechanistically possible with three components simultaneously binding to the barbed end. The authors are keeping a low profile in this regard, but this is an issue that needs to be discussed in more depth.

We thank the reviewer for pushing us to think more deeply about our proposed mechanism. We now devote a new paragraph in the discussion to this concept. Importantly, without higher-resolution studies beyond the scope of this work, we cannot determine the exact juxtaposition of molecules in the vicinity of the terminal actin subunits.

Is the ternary IQGAP1-mDia1-CP complex also seen when mDia1 (FH1-C) is used?

This is an interesting idea that we did not try. We are not sure if IQGAP1 and CP are direct binding partners. We did not have enough IQGAP1 protein to assess this in traditional binding assays and could not definitively conclude this from 4-color TIRF. We also do not expect to see significant levels of IQGAP1 on filament ends as it doesn't associate with FH1-C as effectively as Δ DAD. We expect there would be some colocalization because FH1-C and CP can bind each other. However, if it is possible, we expect IQGAP1 interaction in this complex could only be through CP.

In experiments related to Fig. 7, the authors have attempted to test the physiological significance of these biochemical activities in cells. However in my opinion, these experiments do not come close to addressing the core of this question. How do they define "physiological" when they force cells to spread in confinement on micropatterns? And how do they assess the physiological significance of these complexes in the cellular context, if they exclusively look at potential function of IQGAP1 in morphology and cell migration?

We agree that whole additional manuscripts could be dedicated to this question to address it in the detail it deserves! Our original motivation for using the micropatterns was an attempt to "standardize" the shape of mammalian cells. The point on confining the cells by multiple reviewers is well taken and to address this we now include cell circularity measurements of unconfined cells (**Figure 6A-B**). Cells expressing endogenous (mouse IQGAP1) were the most circular, whereas IQGAP1 knockout cells were significantly less circular ($P < 0.0001$). Circularity values could not be rescued by expressing human IQGAP1 or SNAP-IQGAP1 off plasmids in the knockout cell line, though these rescues were significantly more circular than mock-treated knockout cells ($P = 0.0364$ and $P = 0.0151$, respectively).

We also edited the text to state the motivations and logic behind using morphology and migration measurements. As stated above, we chose NIH-3T3 cells for our analyses because

they require intact actin dynamics for migration and foundational studies of IQGAP1 used them to dissect cytoskeletal-based functions (DOIs: 10.1074/jbc.M304838200; 10.1242/jcs.044644; 10.4161/bioa.21182; 10.1091/mbc.E15-07-0489).

In Fig. 7B, they seem to have measured only area unrelated to the actual fluorescence, but not morphology. Potential changes in cell morphology could be much better analyzed by looking for instance at different parameters such as roundness, ellipticity etc. of unconfined cells. I also doubt that these measurements are statistically valid. The IQGAP1-KO re-expressing Halo-IQGAP1 (red bar) is supposed to be statistically different from the IQGAP1-KO (blue bar), but the values are virtually identical. This also accounts for IQGAP1-KO re-expressing IQGAP1(CD) (pink bar). Furthermore, it appears that statistical analysis selects only those pairs that fit into the picture. Why is NIH 3T3 WT not compared to the KO re-expressing wild type IQGAP1? This would also reveal a statistical difference. Why are error bars show as SEM and not as SD? Expression and localization in transfected cells have not been checked. So how do they know that analyzed cells express respective constructs? Taken together, I cannot make sense of these experiments and recommend removing this data completely.

As requested, we include new experiments with unconfined cells and measure the circularity parameter (**Figure 6A-B**). Thank you for this helpful suggestion. Relevant comparisons and P-values are stated in the revised text.

We went to great lengths to quantify the transfection efficiency of our cells, which may help to alleviate this reviewer's concern (**Figure S4**). First, we verified that multiple clonal cell lines were true knockouts that lacked mouse IQGAP1 using western blots (**Figure 1A-B**). We performed the requested Western to verify the levels of human IQGAP1 from our plasmid-expressed system (**Figure S4B**). We were able to detect ~40% transfection efficiency by this method. Next, we used antibodies to check the localization of IQGAP1, which is cytoplasmic and does not appear to change when we visualize IQGAP1 off plasmids with the fluorescence SiR SNAP ligand (**Figure S4C**). We also assessed transfection efficiency of each human IQGAP1 plasmid (including tagged and untagged controls) with a GFP-actin probe due to its more obvious recognizable features (**Figure S4E-F**). To further alleviate this reviewer's concerns, we then repeated every individual cell-based experiment in the paper only including them in the final analysis after visual confirmation that transfection occurred. Thus, we are now confident that we can transfect the various cells used in this work and that the observed cellular phenotypes are associated with IQGAP1 function.

In Figs. 7 C and D, it is then proposed that the loss of IQGAP1 results in the loss of the ventral stress fibers. Unlike NIH 3T3 cells in which numerous stress fibers are visible, the IQGAP1-KO and reconstituted cells expressing either IQGAP1 or IQGAP1 (CD) all seem to have few stress fibers, albeit quantifications of global phalloidin fluorescence intensities seem to be increased in the latter. Besides the fact that this is not an accurate method for the analysis of (ventral) SFs, these findings are not explained or discussed. Instead of measuring global phalloidin fluorescence, the authors should directly quantify number and phalloidin fluorescence of stress fibers (and possible focal adhesions) of unconfined cells. If they want to make the point that these are contractile stress fibers, then they should also stain the cells with myosin II or alpha-actinin to visualize the striated patterns of these structures.

We have removed the statement about specific stress fibers, which as Reviewer 1 also noted, was not accurate.

In Fig. 7E, the authors analyzed the behavior in wound healing assays using the same collection of cells. Important controls are missing. Please provide a blot depicting the expression level of respective constructs. It would also be important to know the fraction of cells expressing

these constructs, since these cells were only transiently transfected. But again, where is the connection to the complexes in this experiment?

As described above we have added many additional experiments and controls including requested Western blots, several measures of transfection efficiency, and further validation of protein localization and probes used (**Figure S4A-F**).

As stated in the response to Reviewer 2, previous studies have shown IQGAP1 plays important roles in actin-based processes in cells. The IQGAP1(BAD) mutant provides an excellent opportunity to test the specific function of IQGAP1's pausing activity in cells, as we demonstrated with multiple direct approaches that it does not have plus-end pausing activity of the normal protein. By expressing this in cells that lack IQGAP1 and assessing cellular activities we use reverse genetics to show the two cysteine residues are essential for several physiological processes. We clarify the logic underlying these experiments in the revised text.

NIH 3T3 cells (which should not be termed "endogenous") are rarely diploid. They are mostly tri- or tetraploid (Leibiger et al., 2013, *Journal of Histochemistry and Cytochemistry*). Thus, NIH 3T3 cells should not be classified as (+/+) and IQGAP1 KO (-/-) unless the authors have determined the ploidy of the cells.

We originally used the +/+, -/- notation to help readers understand when versions of IQGAP1 were present or not. We have removed this notation from the revised text. We use the term "endogenous" to refer to the native levels of *mouse* IQGAP1 present in cells that have not undergone any sort of genetic manipulation. The most relevant comparisons are between mock-treated (all transfection reagents lacking IQGAP1 plasmids) cells lacking IQGAP1 (verified knockouts, **Figure S4A-B**) and these cells transfected with plasmids harboring versions of Human IQGAP1 including a tag-free IQGAP1, SNAP-IQGAP1, and SNAP-IQGAP1(BAD). We use this notation for all cell biology-based figures in the revised manuscript (**Figures 6 and S4**).

The authors mention in the method section that they use artificial intelligence to denoise their images, but the program/algorithm is not mentioned. As this approach removes information from the original images, potentially increases resolution and modifies gamma, I am not sure if this is in line with the current JCB image and video guidelines.

We are sorry for the confusion our terminology caused. We used the terminology that the microscope manufacturer (Nikon) uses to describe this module. All images were acquired and processed in the same manner and are therefore compatible and within the guidelines on JCB's website. We have reworded our description of this module and provided more detail in the revised methods of this manuscript.

April 30, 2024

RE: JCB Manuscript #202305065R-A

Dr. Jessica L. Henty-Ridilla
SUNY Upstate Medical University
Biochemistry and Molecular Biology
750 East Adams Street
4271 Weskotten Hall Addition
Syracuse, NY 13210

Dear Dr. Henty-Ridilla:

Thank you for submitting your revised manuscript entitled "Coordination of actin plus-end dynamics by IQGAP1, formin, and capping protein". We would be happy to publish your paper in JCB pending final revisions necessary to meet our formatting guidelines (see details below).

A. MANUSCRIPT ORGANIZATION AND FORMATTING:

Full guidelines are available on our Instructions for Authors page, <http://jcb.rupress.org/submission-guidelines#revised>. Submission of a paper that does not conform to JCB guidelines will delay the acceptance of your manuscript.

1) Text limits: Character count for Articles is < 40,000, not including spaces. Count includes abstract, introduction, results, discussion, and acknowledgments. Count does not include title page, figure legends, materials and methods, references, tables, or supplemental legends.

2) Figures limits: Articles may have up to 10 main figures and 5 supplemental figures/tables.

3) Figure formatting: Scale bars must be present on all microscopy images, including inset magnifications. Molecular weight or nucleic acid size markers must be included on all gel electrophoresis. Please avoid pairing red and green for images and graphs to ensure legibility for color-blind readers. If red and green are paired for images, please ensure that the particular red and green hues used in micrographs are distinctive with any of the colorblind types. If not, please modify colors accordingly or provide separate images of the individual channels.

** Please include scale bars for Figure 3B, and indicate the scale on the figure panel for kymographs in Supplemental Figure 1C.

4) Statistical analysis: Error bars on graphic representations of numerical data must be clearly described in the figure legend. The number of independent data points (n) represented in a graph must be indicated in the legend. Statistical methods should be explained in full in the materials and methods. For figures presenting pooled data the statistical measure should be defined in the figure legends. Please also be sure to indicate the statistical tests used in each of your experiments (either in the figure legend itself or in a separate methods section) as well as the parameters of the test (for example, if you ran a t-test, please indicate if it was one- or two-sided, etc.). Also, if you used parametric tests, please indicate if the data distribution was tested for normality (and if so, how). If not, you must state something to the effect that "Data distribution was assumed to be normal but this was not formally tested."

5) Abstract and title: The abstract should be no longer than 160 words and should communicate the significance of the paper for a general audience. The title should be less than 100 characters including spaces. Make the title concise but accessible to a general readership.

6) Materials and methods: Should be comprehensive and not simply reference a previous publication for details on how an experiment was performed. Please provide full descriptions in the text for readers who may not have access to referenced manuscripts. We also provide a report from SciScore and an associate score, which we encourage you to use as a means of evaluating and improving the methods section.

** We appreciate that actin binding protein and rabbit muscle actin were purified by different methods that are well described in the references provided. If it is reasonable and accurate, please include a ~brief~ description of these purification methods that encompasses all these proteins. Our aim is to include all protocols without needing to look up references, but we do understand that these are quite well-established in this field.

7) Please be sure to provide the sequences for all of your primers/oligos and RNAi constructs in the materials and methods. You must also indicate in the methods the source, species, and catalog numbers (where appropriate) for all of your antibodies.

Please also indicate the acquisition and quantification methods for immunoblotting/western blots.

8) Microscope image acquisition: The following information must be provided about the acquisition and processing of images:

- a. Make and model of microscope
- b. Type, magnification, and numerical aperture of the objective lenses
- c. Temperature
- d. Imaging medium
- e. Fluorochromes
- f. Camera make and model
- g. Acquisition software
- h. Any software used for image processing subsequent to data acquisition. Please include details and types of operations involved (e.g., type of deconvolution, 3D reconstitutions, surface or volume rendering, gamma adjustments, etc.).

10) Supplemental materials: There are strict limits on the allowable amount of supplemental data. Articles may have up to 5 supplemental figures. Please also note that tables, like figures, should be provided as individual, editable files. A summary of all supplemental material should appear at the end of the Materials and methods section.

13) ORCID IDs: ORCID IDs are unique identifiers allowing researchers to create a record of their various scholarly contributions in a single place. At resubmission of your final files, please provide an ORCID ID for all authors.

15) A data availability statement is required for all research article submissions. The statement should address all data underlying the research presented in the manuscript. Please visit the JCB instructions for authors for guidelines and examples of statements at (<https://rupress.org/jcb/pages/editorial-policies#data-availability-statement>).

Please note that JCB requires authors to submit Source Data used to generate figures containing gels and Western blots with all revised manuscripts. This Source Data consists of fully uncropped and unprocessed images for each gel/blot displayed in the main and supplemental figures. Since your paper includes cropped gel and/or blot images, please be sure to provide one Source Data file for each figure that contains gels and/or blots along with your revised manuscript files. File names for Source Data figures should be alphanumeric without any spaces or special characters (i.e., SourceDataF#, where F# refers to the associated main figure number or SourceDataFS# for those associated with Supplementary figures). The lanes of the gels/blots should be labeled as they are in the associated figure, the place where cropping was applied should be marked (with a box), and molecular weight/size standards should be labeled wherever possible. Source Data files will be directly linked to specific figures in the published article.

B. FINAL FILES:

Thank you for your attention to these final processing requirements. Please revise and format the manuscript and upload materials within 7 days. If you need an extension for whatever reason, please let us know and we can work with you to determine a suitable revision period.

Thank you for this interesting contribution, we look forward to publishing your paper in Journal of Cell Biology.

Sincerely,

Pekka Lappalainen
Monitoring Editor
Journal of Cell Biology

Tim Fessenden
Scientific Editor
Journal of Cell Biology

Reviewer #1 (Comments to the Authors (Required)):

In this manuscript, Pimm and colleagues examine the role of IQGAP1 in mediating 'decision complexes' at the plus ends of actin filaments. The biggest finding is that IQGAP1 is involved in enhancing removal of multiple plus-end actin binding proteins, presumably allowing for more 'dynamic' plus ends that can allow for consistent turnover of different plus end actin binding proteins. These findings have interesting potential relevance for actin network architecture transitions specifically at the leading edge of cells.

I was happy to read the newest version of this paper. The authors clearly put a lot of work into updating the manuscript. Overall, it was much easier to follow, the figures were clear, the data was much more thorough and more convincing, and the claims made more accurately represented the data, and included the appropriate caveats when needed. The inclusion of pyrene data in addition to TIRF substantially increased the rigor and value of the study. I think the authors did a great job with the revision and I recommend it for publication in JCB.

Reviewer #2 (Comments to the Authors (Required)):

The authors have done their best to address the reviewer's detailed comments. I believe that this manuscript contains a substantial amount of new data that improve our mechanistic understanding of the dynamics at the growing end of an actin filament in the presence of regulatory proteins. I therefore recommend publication of this work in The Journal of Cell Biology.

- 1) Text limits: Character count for Articles is < 40,000, not including spaces. Count includes abstract, introduction, results, discussion, and acknowledgments. Count does not include title page, figure legends, materials and methods, references, tables, or supplemental legends.

There are 36,588 characters (no spaces) in the following sections combined: abstract, introduction, results, discussion, and acknowledgements.

- 2) Figures limits: Articles may have up to 10 main figures and 5 supplemental figures/tables.

There are six main figures and four supplemental figures presented.

- 3) Figure formatting: Scale bars must be present on all microscopy images, including inset magnifications. Molecular weight or nucleic acid size markers must be included on all gel electrophoresis. Please avoid pairing red and green for images and graphs to ensure legibility for color-blind readers. If red and green are paired for images, please ensure that the particular red and green hues used in micrographs are distinctive with any of the colorblind types. If not, please modify colors accordingly or provide separate images of the individual channels.

** Please include scale bars for Figure 3B, and indicate the scale on the figure panel for kymographs in Supplemental Figure 1C.

Size markers are present on all gels.

Figure color choices are color-blind friendly according to "Color Oracle" (<https://colororacle.org/>). Differences in most figures can be seen based on shading, except for the 4-color panels, which are presented as individual channels (avoiding red/green combos) and merged views.

Scale bars are now present on all images. We now include them on Figures 3B, 4E' inset, S1C, and S3G' inset.

- 4) Statistical analysis: Error bars on graphic representations of numerical data must be clearly described in the figure legend. The number of independent data points (n) represented in a graph must be indicated in the legend. Statistical methods should be explained in full in the materials and methods. For figures presenting pooled data the statistical measure should be defined in the figure legends. Please also be sure to indicate the statistical tests used in each of your experiments (either in the figure legend itself or in a separate methods section) as well as the parameters of the test (for example, if you ran a t-test, please indicate if it was one- or two-sided, etc.). Also, if you used parametric tests, please indicate if the data distribution was tested for normality (and if so, how). If not, you must state something to the effect that "Data distribution was assumed to be normal but this was not formally tested."

We added the requested statement about the data distribution to the "Analysis, statistics and representation" section of the methods. The missing n = values and statistical test descriptions for Figures 3A, 4C-D and S3F are now stated in the figure legends.

- 5) Abstract and title: The abstract should be no longer than 160 words and should communicate the significance of the paper for a general audience. The title should be less than 100 characters including spaces. Make the title concise but accessible to a general readership.

The abstract is exactly 160 words. The title is 78 characters with spaces.

- 6) Materials and methods: Should be comprehensive and not simply reference a previous publication for details on how an experiment was performed. Please provide full descriptions in the text for readers who may not have access to referenced manuscripts. We also provide a report from SciScore and an associate score, which we encourage you to use as a means of evaluating and improving the methods section.

** We appreciate that actin binding protein and rabbit muscle actin were purified by different methods that are well described in the references provided. If it is reasonable and accurate, please include a ~brief~ description of these purification methods that encompasses all these proteins. Our aim is to include all protocols without needing to look up references, but we do understand that these are quite well-established in this field.

We now include the following brief description of these methods:

RMA was purified from acetone powder previously stored at -80 °C. Acetone powder was ground, rehydrated in G-buffer (3 mM Tris pH 8.0, 0.5 mM DTT, 0.2 mM ATP, 0.1 mM CaCl₂) and cleared via centrifugation. Actin was polymerized at 4 °C overnight and pelleted. The pellet was prepared via dounce homogenization and dialyzed against G-buffer for 2 d, with buffer exchanges every 24 h, cleared via ultracentrifugation, and gel filtered on a 16/60 S200 column (Cytiva). Pyrene-, OG-, and DyLight 405-actin were prepared from RMA pellets (as above), first dialyzed against G-buffer lacking DTT for 4 h, then diluted to 1 mg/mL, polymerized, and labeled with 7-10-fold molar excess dye in 25 mM imidazole (pH 7.5), 100 mM KCl, 0.15 mM ATP, 2 mM MgCl₂, overnight. Actin filaments were pelleted and subjected to the same dialysis and gel filtration treatments as the unlabeled RMA above. Fluorescence RMA labeled on lysine residues was made similarly except the initial dialysis was performed using HEPES-buffered G-buffer (3 mM HEPES (pH 8.2), 0.5 mM DTT, 0.3 mM ATP, 0.1 mM CaCl₂) and labeled on actin filaments in 3 mM HEPES (pH 8.2), 0.5 mM DTT, 0.3 mM ATP, 1 mM MgCl₂, 50 mM KCl. Profilin and CP were purified from previously frozen *E. coli* pellets induced and stored as above. Cells from both pellets were lysed via sonication in the presence of lysozyme and protease inhibitors and precleared via centrifugation. Profilin lysates were subjected to ion exchange chromatography via a 5 mL HiTrap column (Cytiva) over a 30 mL 0-500 mM KCl gradient in 50 mM Tris (pH 8.0), 50 mM KCl, 1 mM DTT. Peak fractions were pooled, and gel filtered on a Superdex 75 Increase (10/300) column (Cytiva) in 50 mM Tris (pH 8.0), 50 mM KCl, 1 mM DTT. CP lysates were loaded onto a different 5 mL HiTrap column (Cytiva) and subjected to a 45 mL salt gradient (0–500 mM KCl) in 20 mM Tris (pH 8.0). Peak fractions were pooled (and labeled as above for SNAP-IQGAP1, below), then gel filtered on a Superdex 75 Increase (10/300) column (Cytiva) into 20 mM Tris-HCl, pH 8.0; 50 mM KCl, 1 mM DTT. Peak fractions for profilin or CP were pooled, flash frozen, and stored at -80 °C.

- 7) Please be sure to provide the sequences for all of your primers/oligos and RNAi constructs in the materials and methods. You must also indicate in the methods the source, species, and catalog numbers (where appropriate) for all of your antibodies. Please also indicate the acquisition and quantification methods for immunoblotting/western blots.

No primers were used in this work.

New plasmids either synthesized/made by Genescript or cloned via sticky ends as already described in the methods.

The source, species, and dilutions of all primary antibodies used for all Western blots were already included in the methods. We now include these details for the two secondary antibodies used for blotting which are as follows: 1) LI-COR IRDye 680RD Goat anti-Mouse IgG (H + L) (Fisher catalog #: NC0252290); 2) LI-COR IRDye 800CW Donkey anti-Rabbit IgG (H + L) (Fisher catalog #: NC9523609). All antibody catalog numbers are now more clearly detailed. No blots were quantified in this work. All blots were acquired using a LI-COR Odyssey Fc system (LI-COR Biosciences, Lincoln, NE).

- 8) Microscope image acquisition: The following information must be provided about the acquisition and processing of images:

- a. Make and model of microscope:
 - i. Several are used in this work. Each are described. Further, each section specifies which scope was used to collect each dataset.
- b. Type, magnification, and numerical aperture of the objective lenses:
 - i. The objectives used and their NA are stated with each microscope setup.
- c. Temperature:
 - i. The temperature for TIRF experiments was already listed in the methods. We have now further specified the temperature in addition to how control was maintained for the wound healing assays. Specifically, temperature was maintained in an OKO labs stage insert (OKO labs, Pozzuoli, Italy) set to 37 °C. Cells were returned to a tissue culture incubator set to 37 °C with 5% CO₂ (Heracell VIOS 160i; Fisher) in between imaging timepoints.
- d. Imaging medium
 - i. The imaging medium is stated for all experiments. For in vitro biochemistry TIRF imaging is performed in: “20 mM imidazole (pH 7.4) 50 mM KCl, 1 mM MgCl₂, 1 mM EGTA, 0.2 mM ATP, 10 mM DTT, 40 mM glucose, and 0.25% methylcellulose (4000 cP)), diluted from a 2× stock, with 1 μL of anti-bleach solution (10 mg/mL glucose oxidase and 2 mg/mL catalase), proteins of interest, and appropriate buffer controls for each assessed protein/combination).”

For wound healing experiments cells were imaged in conventional DMEM (Gibco, Grand Island, NY), supplemented with 10% FBS (Genesee Scientific, San Diego, CA), 200 mM L-glutamine (Gibco), and 45 U/mL penicillin-streptomycin (Gibco), and further buffered with 10 mM HEPES (pH 7.4).
- e. Fluorochromes
 - i. All fluorochromes and fluorescent proteins are listed and described in the methods and in instances of use in each figure.
- f. Camera make and model
 - i. Three different microscope cameras are used in this work. Each one is listed with the description of each microscope system.
- g. Acquisition software
 - i. Leica LAS X was used for the Lecia-based TIRF system. Nikon Elements with additional ai denoise modules was used for Nikon-based TIRF and SoRA systems. Both are listed in the methods.
- h. Any software used for image processing subsequent to data acquisition. Please include details and types of operations involved (e.g., type of deconvolution, 3D reconstitutions, surface or volume rendering, gamma adjustments, etc.).
 - i. Fiji software and additional Nikon ai modules were used to process some of the images in this work. The types and order of image processing are described with each analysis in the methods.

The reference list is in this format. We have updated any Preprints that are now accepted articles.

- 10) Supplemental materials: There are strict limits on the allowable amount of supplemental data. Articles may have up to 5 supplemental figures. Please also note that tables, like figures, should be provided as individual, editable files. A summary of all supplemental material should appear at the end of the Materials and methods section.

There are four supplemental figures and seven videos associated with this work. A detailed description of these files follows:

Online supplemental material

Figure S1 (related to Figure 2), provides schematics, SDS-PAGE gels, and example kymographs and quantification of pauses to elongation for untagged IQGAP1 proteins. Figure S2 (related to Figure 3), provides SDS-PAGE gels of SNAP-tagged IQGAP1 proteins and provides a comparison of the different elongation rates of each fluorescently labeled actin used in this study. Figure S3 (related to Figure 4), contains bulk polymerization assays and the elongation rates of actin filaments from TIRF reactions comparing mDia1(FH1-C) and mDia1(Δ DAD) under different conditions. Additional panels detail multi-color TIRF of 488-IQGAP1, 549-mDia1(FH1-C), 647-actin. Figure S4 (related to Figure 6), provides validation of IQGAP1 knockout lines and transfection efficiency of each IQGAP1 construct in NIH-3T3 cells and details effects of IQGAP1 constructs on cell morphology and microtubule arrays. Video 1 (related to Figure 1C) details actin polymerization in the presence of different IQGAP1 concentrations. Video 2 (related to Figure 1D) is an example of an IQGAP1-mediated pause to actin filament growth. Video 3 (related to Figure 2B) shows TIRF movies of actin filaments polymerizing in the presence of purified IQGAP1 proteins. Video 4 (related to Figure 3E) shows a comparison of 488-SNAP-labeled IQGAP1 proteins with polymerizing 647-actin via two-color TIRF microscopy. Video 5 (related to Figure 4E and 4E') is an example of 488-IQGAP1 displacing 549-mDia1(Δ DAD) from the plus end (647-actin) obtained via three-color TIRF microscopy. Video 6 (related to Figure S3G and S3G') shows a multi-color TIRF video of 488-IQGAP1, 549-mDia1(FH1-C), 647-actin. Video 7 (related to Figure 5D) details the effect of 488-IQGAP1 on 549-mDia1(DAD)-647-Capping Protein decision complexes with 405-actin using four-color TIRF microscopy.

The eTOC Summary was already listed in the manuscript. It is as follows:

IQGAP1 coordinates actin assembly via transient pausing events and by displacing prominent plus-end binding proteins including formin (mDia1), Capping Protein (CP), and mDia1-CP 'decision complexes'.

The declaration of interests is already stated in the work. We have reworded to the exact phrasing used above.

- 13) ORCID IDs: ORCID IDs are unique identifiers allowing researchers to create a record of their various scholarly contributions in a single place. At resubmission of your final files, please provide an ORCID ID for all authors.

All ORCID IDs are linked and already listed in the manuscript file.

We have revised the author contribution section to the CRediT nomenclature as follows:

CRediT Nomenclature:

Author contributions: Conceptualization and methodology – M.L. Pimm, B.K. Haarer, and J.L. Henty-Ridilla. Investigation – M.L. Pimm, B.K. Haarer, A.D. Nobles, A.G. Marcin, and J.L. Henty-Ridilla. Validation – M.L. Pimm, B.K. Haarer, and J.L. Henty-Ridilla. Formal analysis - M.L. Pimm, B.K. Haarer, A.D. Nobles, L.M. Haney, J.L. Henty-Ridilla. Visualization – M.L. Pimm, L.M. Haney, and J.L. Henty-Ridilla. Writing - original draft – M.L. Pimm and J.L. Henty-Ridilla. Writing – review & editing – M.L. Pimm, B.K. Haarer, A.D. Nobles, L.M. Haney, A.G. Marcin, M.A. Eligio, and J.L. Henty-Ridilla. Project administration, supervision, funding acquisition, and resources— J.L. Henty-Ridilla.

- 15) A data availability statement is required for all research article submissions. The statement should address all data underlying the research presented in the manuscript. Please visit the JCB instructions for authors for guidelines and examples of statements at (<https://rupress.org/jcb/pages/editorial-policies#data-availability-statement>).

Datasets for each figure have been uploaded and deposited, and are freely available here <https://doi.org/10.5281/zenodo.10895906>. This information was already present in the manuscript at the end of the methods under the “Data availability” subheading.

Please note that JCB requires authors to submit Source Data used to generate figures containing gels and Western blots with all revised manuscripts. This Source Data consists of fully uncropped and unprocessed images for each gel/blot displayed in the main and supplemental figures. Since your paper includes cropped gel and/or blot images, please be sure to provide one Source Data file for each figure that contains gels and/or blots along with your revised manuscript files. Source Data Figures should be provided as individual PDF files (one file per figure). Authors should endeavor to retain a minimum resolution of 300 dpi or pixels per inch. Please review our instructions for export from Photoshop, Illustrator, and PowerPoint here: <https://rupress.org/jcb/pages/submission-guidelines#revised>

File names for Source Data figures should be alphanumeric without any spaces or special characters (i.e., SourceDataF#, where F# refers to the associated main figure number or SourceDataFS# for those associated with Supplementary figures). The lanes of the gels/blots should be labeled as they are in the associated figure, the place where cropping was applied should be marked (with a box), and molecular weight/size standards should be labeled wherever possible. Source Data files will be directly linked to specific figures in the published article.

We provided a single file containing all the source data for all figures with gels or blots, as detailed above.

B. FINAL FILES:

-- An editable version of the final text (.DOC or .DOCX) is needed for copyediting (no PDFs).
-- High-resolution figure and MP4 video files: See our detailed guidelines for preparing your production-ready images, <https://jcb.rupress.org/fig-vid-guidelines>.

Thank you for your attention to these final processing requirements. Please revise and format the manuscript and upload materials within 7 days. If you need an extension for whatever reason, please let us know and we can work with you to determine a suitable revision period.